# Network of large pedigrees reveals social practices of Avar communities

Guido Alberto Gnecchi-Ruscone[1,21][✉], Zsófia Rácz[2,21], Levente Samu[2], Tamás Szeniczey[3], Norbert Faragó[2], Corina Knipper[4], Ronny Friedrich[4], Denisa Zlámalová[5], Luca Traverso[1], Salvatore Liccardo[6,7], Sandra Wabnitz[6,7], Divyaratan Popli[8], Ke Wang[1,9], Rita Radzeviciute[1], Bence Gulyás[10], István Koncz[2], Csilla Balogh[11], Gabriella M. Lezsák[12], Viktor Mácsai[2], Magdalena M. E. Bunbury[13], Olga Spekker[2,14], Petrus le Roux[15], Anna Szécsényi-Nagy[16], Balázs Gusztáv Mende[16], Heidi Colleran[17,18], Tamás Hajdu[3], Patrick Geary[19], Walter Pohl[6,7], Tivadar Vida[2,20][✉], Johannes Krause[1][✉] & Zuzana Hofmanová[1,5][✉]

From AD 567–568, at the onset of the Avar period, populations from the Eurasian Steppe settled in the Carpathian Basin for approximately 250 years[1]. Extensive sampling for archaeogenomics (424 individuals) and isotopes, combined with archaeological, anthropological and historical contextualization of four Avar-period cemeteries, allowed for a detailed description of the genomic structure of these communities and their kinship and social practices. We present a set of large pedigrees, reconstructed using ancient DNA, spanning nine generations and comprising around 300 individuals. We uncover a strict patrilineal kinship system, in which patrilocality and female exogamy were the norm and multiple reproductive partnering and levirate unions were common. The absence of consanguinity indicates that this society maintained a detailed memory of ancestry over generations. These kinship practices correspond with previous evidence from historical sources and anthropological research on Eurasian Steppe societies[2]. Network analyses of identity-by-descent DNA connections suggest that social cohesion between communities was maintained via female exogamy. Finally, despite the absence of major ancestry shifts, the level of resolution of our analyses allowed us to detect genetic discontinuity caused by the replacement of a community at one of the sites. This was paralleled with changes in the archaeological record and was probably a result of local political realignment.

The kinship practices and social organization of past societies are hard to assess using only the fragmentary archaeological and historical information that has survived to modern times. Biological relatedness does not necessarily correspond to social kinship, but it can nevertheless provide a powerful tool to infer elements of past kinship practices. Ancient DNA has been used for pedigree inference[3–5], but being able to capture the extent of relationships in ancient populations requires a sampling approach that is focused on entire cemeteries of considerable size[6]. Only multiple observations of the same type of relatedness structure can exclude a random occurrence and indicate a reliable pattern. Archaeological contextualization adds social meaning and can disentangle the complex interplay between biological relatedness and human behaviour to help researchers to infer kinship practices on a larger scale.

From the late sixth century AD to the early ninth century, the Avars were the dominant power in eastern central Europe[1,7]. Originating from eastern central Asia, probably from the Rouran khaganate destroyed by the Turks, the Avars' core group of mounted steppe warriors and their families arrived north of the Caucasus in AD 557–558, where further groups joined the march into the Carpathian Basin in 567–568 (refs. 1,7). This region became the centre of the Avar empire, where they settled among a diverse population derived from the previous Roman period followed by the Gepid and Longobard kingdoms[1,8]. After extensive raids into the Byzantine Balkans ended in AD 626, the Avar society changed in many ways. The archaeological record indicates that a sedentary lifestyle in new, stable settlements emerged, with larger cemeteries containing hundreds of graves, and cultural expressions became more

[1]Department of Archaeogenetics, Max Planck Institute for Evolutionary Anthropology, Leipzig, Germany. [2]Institute of Archaeological Sciences, ELTE - Eötvös Loránd University, Budapest, Hungary. [3]Department of Biological Anthropology, ELTE - Eötvös Loránd University, Budapest, Hungary. [4]Curt Engelhorn Center for Archaeometry gGmbH, Mannheim, Germany. [5]Department of Archaeology and Museology, Faculty of Arts, Masaryk University, Brno, Czechia. [6]Department of History, University of Vienna, Vienna, Austria. [7]Institute for Medieval Research, Austrian Academy of Sciences, Vienna, Austria. [8]Department of Genetics, Max Planck Institute for Evolutionary Anthropology, Leipzig, Germany. [9]MOE Key Laboratory of Contemporary Anthropology, Department of Anthropology and Human Genetics, School of Life Sciences, Fudan University, Shanghai, China. [10]Hungarian National Museum, Budapest, Hungary. [11]Department of Art History, Istanbul Medeniyet University, Istanbul, Turkey. [12]Institute of History, HUN-REN Research Centre for the Humanities, Budapest, Hungary. [13]ARC Centre of Excellence for Australian Biodiversity and Heritage, College of Arts, Society and Education, James Cook University, Cairns, Queensland, Australia. [14]Department of Biological Anthropology, University of Szeged, Szeged, Hungary. [15]Department of Geological Sciences, University of Cape Town, Rondebosch, South Africa. [16]Institute of Archaeogenomics, HUN-REN Research Centre for the Humanities, Budapest, Hungary. [17]BirthRites Lise Meitner Research Group, Max Planck Institute for Evolutionary Anthropology, Leipzig, Germany. [18]Department of Human Behavior, Ecology and Culture, Max Planck Institute for Evolutionary Anthropology, Leipzig, Germany. [19]Institute for Advanced Study, Princeton, NJ, USA. [20]Institute of Archaeology, HUN-REN Research Centre for the Humanities, Budapest, Hungary. [21]These authors contributed equally: Guido Alberto Gnecchi-Ruscone, Zsófia Rácz. [✉]e-mail: guido_gnecchi@eva.mpg.de; vida.tivadar@btk.elte.hu; krause@eva.mpg.de; zuzana_hofmanova@eva.mpg.de

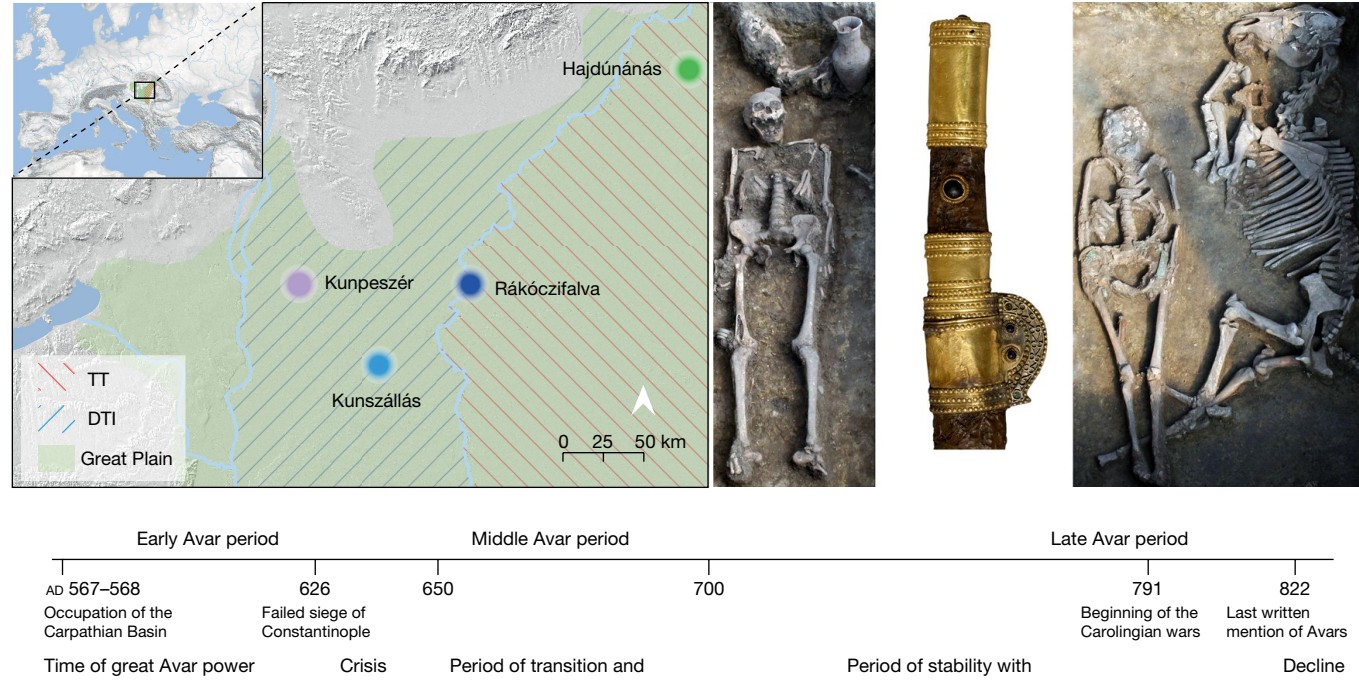

**Early Avar period** — **Middle Avar period** — **Late Avar period**

| AD 567–568 | 626 | 650 | 700 | 791 | 822 |
|---|---|---|---|---|---|
| Occupation of the Carpathian Basin | Failed siege of Constantinople | | | Beginning of the Carolingian wars | Last written mention of Avars |

Time of great Avar power — Crisis — Period of transition and development of large-scale sites — Period of stability with no major changes — Decline

**Fig. 1 | Map of the Great Hungarian Plain showing the locations of the four sites analysed in this study.** Kunpeszér and Kunszállás are located in the DTI; Hajdúnánás and Rákóczifalva are in the TT region. Right, typical archaeological elements that characterize and distinguish between the two main habitation areas of steppe-descent populations of the Avar period: prestigious swords of the DTI elites (KUP) and evidence of burials with horse or animal skin at TT sites (RK). Bottom, timeline of the Avar period in the Carpathian Basin highlighting the three main chronological phases (early, middle and late) showing key historical events. Photo of Kunpeszér sword: Katona József Museum (Kecskemét, Hungary); photo of Rákóczifalva excavation: Sándor Hegedűs. The figure contains modified Copernicus Sentinel data 2024. The map was plotted using R[34].

homogeneous[9]. The Avar realm persisted until it was overcome by the Frankish armies of Charlemagne in around AD 800. Turkic titles of rank (such as khagan, iugurrus, tudun and tarkhan) mentioned in written sources document that the central Asian character of the political structure was maintained until the end of Avar rule[1]. In terms of social structure, patrilineal organization is the norm for Eurasian pastoralist steppe peoples[2], but we were unable to investigate the social practices of the Avars until now owing to a lack of historical sources (Supplementary Information).

By generating new genomic data (Supplementary Table 1) from the exhaustive sampling of four fully excavated cemeteries from present-day Hungary, combined with new isotope data and detailed archaeological and anthropological characterization, we aimed to investigate the population structure, kinship and social organization of these communities at a high level of resolution. We identified 298 biologically closely related individuals that allowed us to reconstruct extensive pedigrees and build networks of distant relatedness across the Great Hungarian Plain. We found striking evidence of recurrent patterns that allowed us to trace kinship and social practices, gain insights into the mobility of men and women and refine the chronology of the sites. In the largest cemetery, we were able to identify a community replacement associated with changes in the archaeological record and dietary habits, suggesting local political realignment. This replacement was not accompanied by an ancestry shift and was detected only by changes in the biological relatedness pattern.

## Analysis of entire cemeteries

The Great Hungarian Plain was the main settlement area for steppe populations during the Avar period. We chose four cemeteries to cover equally its two main regions, divided by the river Tisza: the Transtisza region (TT) east of the river and the Danube–Tisza interfluve region (DTI) to the west (Fig. 1). The DTI was the power centre of the khaganate (the Avar empire), where burials of the highest Avar elite (for example, Kunbábony) were found, and these burials were also investigated in a previous genomic study[10]. From this region, we sampled the site of Kunpeszér (KUP; 33 burials), comprising an early Avar elite cemetery group with rich grave goods, exquisite gold- and silver-decorated swords, belts and jewellery, and a second cemetery group of poor late Avar burials[11]. The site of Kunszállás (KFJ; 63 burials) from the same region, founded in the mid-seventh-century AD, already belongs to the more-uniform material culture of the middle and late Avar period[12]. The TT region is well known for burial customs associated with the steppe, such as the placement of animals, animal skins or horse implements near the deceased. We chose Rákóczifalva (RK) because it is one of the region's largest cemeteries (279 out of 308 graves sampled) that was continuously occupied from around AD 570 to the mid-ninth century[13]. From RK we additionally sequenced 56 individuals from the second to the sixth century to extend the available data on pre-Avar periods and to capture the Avar-period transition on a local scale. The cemetery of Hajdúnánás (HNJ; 18 burials) was selected to cover the northern section of the TT region[14,15] (Supplementary Information; Supplementary Table 2).

After quality controls (Methods; Supplementary Table 1) we obtained genome-wide (around 1,240,000 single nucleotide polymorphisms (SNPs); Methods) data for 424 individuals with an average coverage of 2.6×. Furthermore, we produced new strontium, carbon and nitrogen isotope data ($^{87}Sr/^{86}Sr$; $\delta^{13}C$ and $\delta^{15}N$; Supplementary Table 3) for 154 individuals from RK, KUP and KFJ, and 57 new radiocarbon dates for RK (Supplementary Tables 2 and 6).

## Pedigrees: strict patriliny within sites

To reconstruct the pedigrees, we estimated close biological relatedness using recently published software, KIN (ref. 16), which was designed to

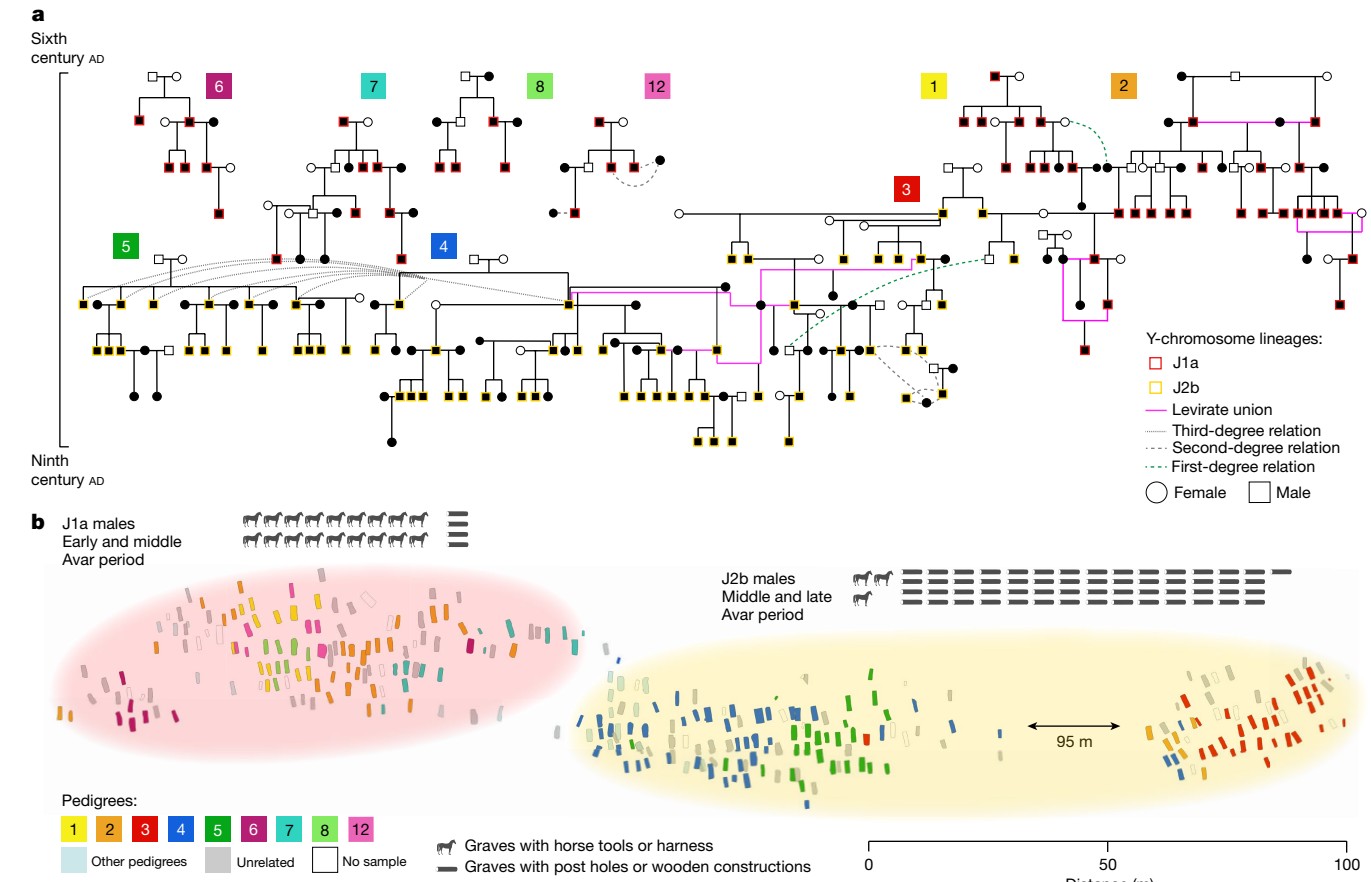

**Fig. 2 | The largest set of interconnected pedigrees reconstructed in RK and the cemetery map highlighting the burial location of related individuals.**
**a**, A large (146 individuals) interconnected set of sub-pedigrees, numbered 1 to 5, and four smaller pedigrees (34 individuals) numbered 6, 7, 8 and 12. Levirate unions are shown with pink lines connecting the individuals involved. The male individuals' Y haplogroups are shown with coloured borders around the individuals. Black symbols refer to individuals whose ancient DNA we have and white ones indicate missing individuals inferred on the basis of the available data. The horizontal axis to the left shows a timeline spanning the whole Avar period, covering the nine or more generations of the pedigrees. **b**, Cemetery map of RK 8, with graves colour coded according to the pedigree shown in **a**. The middle Avar period archaeological transition is exemplified by the different abundance of graves with a horse harness and graves with post holes (each image corresponds to a finding in a grave; silhouette of a horse is from Pixabay). This transition strikingly corresponds to the community shift and spatial organization of the cemetery. The left part is where mostly early-to-middle Avar period individuals and J1a male individuals are found (pink halo), and the right part is where mostly middle-to-late Avar period individuals and J2b male individuals are found (yellow halo).

identify first-, second- or third-degree related individuals (defined as close genetic relatedness) in low-coverage ancient DNA (Methods). We found no close genetic relatedness between sites, but most individuals in each site were closely related, constituting a total of 373 pairs of first-degree (235 parent–child and 138 siblings) and more than 500 pairs of second-degree relatives (Supplementary Table 4). Such a large number, especially of first-degree pairs, allowed us to reconstruct a total of 31 pedigrees of varying sizes, ranging from 2 to 146 individuals (Fig. 2a, Extended Data Figs. 1–3 and Supplementary Figs. 6, 10 and 15). These extended genealogies show a strict patrilineal descent with almost no exceptions. This finding provides compelling evidence for patrilocality and female exogamy, which explains the striking difference in Y-chromosome and mitochondrial DNA (mtDNA) diversity observed among related individuals (Extended Data Fig. 4).

Within RK, 202 individuals had at least one close relative at the site and only 64 were unrelated. Among the related individuals, 146 formed an extended 'macro' pedigree spanning up to nine continuous generations. We divided this into five connected pedigrees (numbered 1 to 5; Methods) descended from 11 founding male individuals. A further 34 individuals formed 4 additional multigenerational pedigrees (numbered 6, 7, 8 and 12) chronologically dated to the early Avar period (Fig. 2a), and the rest formed smaller units (Supplementary Figs. 15 and 16).

Adults (more than 18 years old) represent 83% of the whole RK cemetery, with nearly equal numbers of male and female individuals (Table 1). However, the RK pedigrees contain twice as many male individuals as female ones. This male bias is due to a higher ratio of sons to daughters: we found 102 sons (77 adults and 25 subadults) and 20 daughters, mostly subadults (5 adults and 15 subadults). A strict patrilineality can be observed from the descent structure of the pedigrees. Only one adult daughter (RKC024) has offspring buried in the cemetery, and her son (RKC012) is second-degree related to other members of the pedigree through the missing father (Supplementary Table 4). Consequently, RKC024 and her missing partner are sixth-degree related. All the other mothers lack parents at the site and are considered exogamous partners. Instead, all the fathers are descendants of the founding male individual(s) of their respective pedigree, with no exceptions (Fig. 2a). The founding role in the life of the communities may have been especially important[2]. In several cases, founding male individuals (or, in the case of brothers, one of the male individuals) were buried with valuable grave goods considered status symbols: horse harnesses and belt sets in the early Avar period and belt sets in the middle and late periods (Supplementary Fig. 10).

The KFJ pedigree presents the same pattern of genetic relatedness (Extended Data Fig. 1). Here, of the 45 individuals forming the

**Table 1 | RK Individuals with genetic data and age at death information**

| | All individuals | Individuals in pedigrees |
|---|---|---|
| All individuals | 276 | 202 |
| Adults (more than 18 years old) | 229* | 160 |
| Subadults | 47* | 42 |
| Adult male individuals (more than 18 years old) | 123 | 106 |
| Adult female individuals (more than 18 years old) | 105 | 54 |
| Subadult male individuals | 28 | 26 |
| Subadult female individuals | 18 | 16 |

*Two individuals were of undetermined sex so these are not included in the male and female totals.

second-largest pedigree, 21 are sons (10 of whom are subadults) and 13 are daughters (11 subadults). We observe a similar tendency in the small KUP and HNJ cemeteries, although with fewer individuals (Extended Data Figs. 2 and 3).

These patterns are reflected in the striking difference between female lines (mtDNA haplogroups) and male lines (Y-chromosome haplogroups). Only two Y-chromosome lineages, J1a-Z2317 and J2b-CTS11760 (J1a and J2b hereafter), are found in pedigrees 1–8 and 12 in RK, compared with around 50 different mtDNA haplogroups; only one Y-chromosome lineage, N1a-Y16220 (N1a hereafter) is found in both KFJ and KUP among related and unrelated individuals (compared with around 20 mtDNA haplogroups); and another, Q1a-L715 (Q1a hereafter), is shared between all male individuals of HNJ pedigree 1 (Extended Data Fig. 4).

A comparison of pedigrees with the spatial arrangement of graves and grave groups allows us to assess how much biological and social relatedness correspond, and demonstrates that the concept of descent was central to the organization of the burial site. With few exceptions, all individuals from the same pedigree are found in the same burial cluster (Fig. 2b).

In terms of closer-descent units, we discovered that parents, infants, juveniles and even adult male siblings were buried near each other, forming clusters of close relatives (Supplementary Fig. 9). Within these groups we often find unrelated female individuals. In fact, there is a strong sex bias among the 64 unrelated individuals in RK, because 51 are female and only 13 are male. Most of these female individuals are young adults. Male individuals have a more balanced age distribution, and among related female individuals, older adults are more frequent (Supplementary Fig. 11). The age distribution, position in the pedigree clusters, chronology, burial customs and grave goods suggest that the unrelated female individuals are likely to be exogamous partners of lineage male individuals who had not yet reproduced, or whose children were not found at the site. Therefore, they are not detected as biologically related but could still be part of the social unit.

On the basis of the pedigrees, we can speculate that the beginning of the reproductive age for women was 18–20 years. The youngest mothers were 18–22 years old at death, whereas the youngest fathers were 24–29 years old at death. This is consistent with the observation that juveniles are buried next to their parents (female individuals of 16–19 and male ones of 18–22 years old at death), and lineage female individuals disappear from the pedigrees at late juvenile–early adult age.

## Marriage strategies and levirate unions

Another consistent pattern between sites is that male and female individuals often had multiple reproductive partners. In RK only, we discovered 15 cases involving a male partner and 7 cases involving a female one (Supplementary Information). Male individuals had two partners in ten cases, three partners in four cases, and four partners in one case (RKF042); around 85% of these individuals are older men (aged 35–59). The young ages of female partners at death may indicate serial monogamy (RKC011), but the presence of older female partners in multiple partnerships suggest polygyny (RKF042 and RKF180). Multiple reproductive partners were also discovered in HNJ and KFJ (one and four cases, respectively). That means that polygyny might not have been restricted to the highest stratum of society that is known from the historical sources, but also occurred in the general population[1].

We also identified multiple cases (five in RK and two in KFJ) and, through indirect evidence, another case in KUP, of closely related male individuals having offspring with the same female partner: three pairs of fathers and sons, two pairs of full brothers, one pair of paternal half-brothers and one pair of paternal uncle and nephew (Extended Data Figs. 1 and 2). We assume that these unions were levirate matches (although in some cases concurrent polyandry cannot be excluded; Supplementary Information). Even though the word levirate has a biblical origin[17], in historical and anthropological research the term has a wider application referring to marriages between a widow and an agnate of the deceased. Often found in pastoral societies, which are patrilocal, patrilineal and observe female exogamy, the levirate custom was common in Central Asia and the Caucasus until recent times[18,19]. Although not mentioned for the Avars, levirate partnerships are attested to in contemporary written sources for several steppe peoples[20,21], which suggests that what we find in the pedigrees is probably formal levirate, not extramarital relationships.

According to the sources[22], no levirate union could occur if the deceased's agnate was related to the widow by blood. Indeed, we find no cases of biological consanguinity, based on the absence of long runs of homozygosity (ROH) segments in all analysed individuals (Extended Data Fig. 5). We do not even detect ROH patterns consistent with more-distant consanguineous unions, such as at the level of second-degree cousins, despite a high occurrence of levirate and multipartner unions. Among Eurasian steppe peoples, intermarriage within the paternal line was permitted only after a certain number of generations, which could range between five and nine[20,21]. Such rules would explain the absence of even distant biological consanguinity. It is intriguing that the only case we detected of reproductive partners being related was to the sixth degree (which would still be consistent with such rules) and involves the only non-exogamous female individual in RK. This further suggests the uniqueness of this single case.

All the aforementioned phenomena lead us to assume that the segment of Avar society we investigated had a structure comparable to that of Eurasian pastoralist steppe people[2,21]: the elementary social unit is the patrilineally organized family. Patrilineal genealogies are the constitutive elements of the society and, within them, descent lines are traced and ranked according to the birth order of the male founders. This concept results in a strictly hierarchical structure in the smaller, as well as in the larger, units of society, as evidenced in the archaeological material by various status indicators (Supplementary Fig. 10). We can consider as a contemporary parallel to Avar society the old Turkic kinship system that has been reconstructed on the basis of the Orkhon inscriptions, which date from the eighth century (Supplementary Information).

## Community links through female exogamy

We observe that exogamous female individuals have a central role in connecting the different founding patrilines both within RK and between the sites. One unique case is represented by the female individual RKF140, who is part of two different levirate unions and had a total of four reproductive partners from two different pedigrees, linking the two large patrilineal units of the middle–late Avar period pedigrees (3 and 4–5). In fact, most of the large RK pedigrees are connected

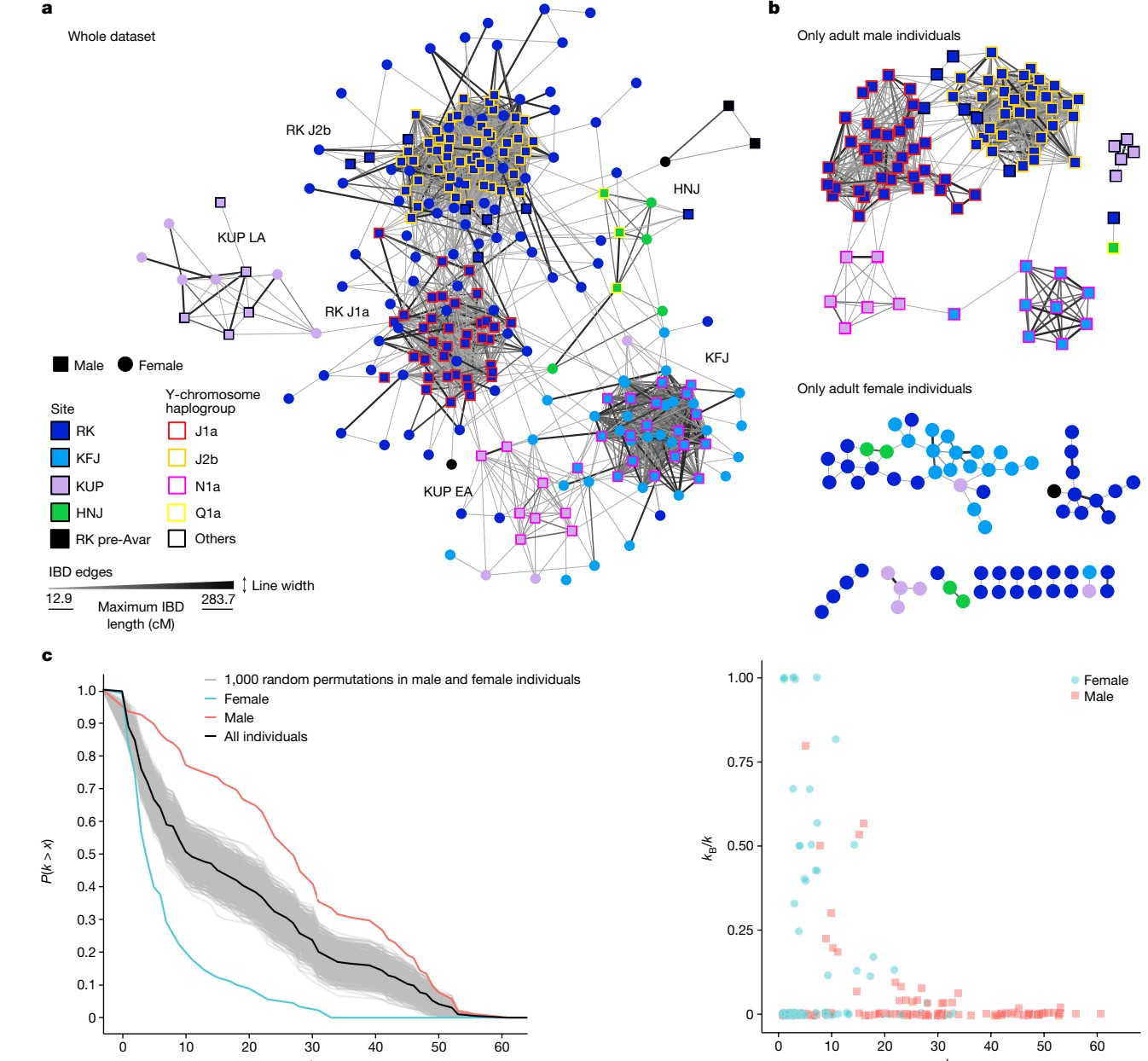

**Fig. 3 | Network analysis of ancIBD haplotype-IBD sharing between Avar-period individuals. a**, Visualization of the network of IBD connections (edges) between individuals (nodes) coloured according to their site: RK, KFJ, KUP and HNJ. The male individuals' Y haplogroups are shown with coloured borders. The strength of the IBD connection is summarized by the maximum IBD length (centimorgans) for each pair of individuals. The distribution of these lengths from the lowest (>12 cM cut-off) to the highest (>280 cM for first-degree relatives) is indicated by the width and colour scale of each edge. **b**, Networks for adult male individuals only (top) and adult female individuals only (bottom). **c**, Network statistics calculated on the adults-only network. Left, degree centrality, $k$ (the number of links held by the node), against the cumulative density function of the degree distribution, defined as the probability that $k$ is more than a value $x$ $P(k > x)$. Right, total $k$ plotted against the ratio of $k$ calculated between site edges ($k_B$) to total $k$.

through female lines: one missing first-degree-related female individual (sister or mother) connects pedigrees 1 and 2, and two maternal second-degree relatives connect pedigrees 2 and 3.

The role of exogamous female individuals becomes even more evident when analysing the patterns of pairwise ancIBD haplotype-IBD (identical-by-descent) sharing within and between individuals from the four Avar sites (Fig. 3). In the network analyses of IBD sharing (Fig. 3a), we can observe tight clusters, reflecting the close genetic relatedness, expectedly within the large pedigrees of RK and KFJ. In the adults-only network (Fig. 3b), we estimate that many female individuals plot outside each site's cluster and have significantly fewer IBD connections than do male individuals, reflected by the lower degree centrality

distribution, corresponding to the number of connections each individual has ($P < 0.05$ using the Kolmogorov–Smirnov test after 1,000 permutations; Fig. 3c). Female individuals instead show significantly higher ratios of connections between sites ($P < 0.05$ using Welch's $t$-test; Fig. 3c). Furthermore, we found seven cases of female individuals (and no male ones) who are unrelated within site presenting IBD connections with another site (Fig. 3a,b). Taken together, our evidence shows the existence of networks of communities centred tightly around a patriline and related to other communities by exogamous female individuals.

We included data from previously published Avar-period sites[10,23] in the IBD network, although there is a bias in sampling strategies between our entire-cemetery-sampling approach and previous sparse-sampling

approaches in which only a few individuals per site were analysed, preventing us from observing the full extent of the connections (Extended Data Fig. 6a).

Nevertheless, it is possible to observe geographic structuring, because we find more intra- than inter-regional connections among the DTI and TT sites. Furthermore, the two nearby sites of Hortobágy-Árkus and HNJ (which are about 50 km apart) are particularly highly connected and share the same Y haplogroup (Extended Data Fig. 6a). We also observe that DTI male individuals with the N1a Y haplogroup tend to cluster together. This lineage is not only common between KUP and KFJ sites but is also shared among the early Avar-period DTI elite sites. Interestingly, the supposedly highest-status individual among these sites (the solitary burial of Kunbábony, which was interpreted as a possible khagan burial on the basis of rich status symbols found in the grave[8]) has the highest number of between-sites IBD connections among all of the new and previously published individuals analysed (Extended Data Fig. 6b).

## Realignment of local power

Archaeologically, RK spans the whole Avar period. By incorporating the relative chronological framework provided by the generations of the pedigrees in the Bayesian modelling of [14]C dates, which reduces the uncertainties of the dates by up to 60% (ref. 24), we were able to refine the start and end events of three large pedigrees (with a maximum span of around 300 years) and place them in a relative order (Supplementary Fig. 43, Supplementary Table 6 and Supplementary Information). Integrating chronology and pedigrees allows us to observe a shift in the local community in the second half of the seventh century. First, ten smaller pedigrees are found in the early Avar period, but only three connected ones dominate in the middle and late phases (Fig. 2). Strikingly, the J1a male lineage is mostly found in the early pedigrees, whereas the J2b haplogroup appears and becomes the predominant male lineage in the later ones (pedigrees 3, 4 and 5 all carry haplogroup J2b). We can clearly pinpoint when this shift occurs: going from pedigree 2 to pedigree 3, through the connection between the two maternal half-brothers, with one from pedigree 2 carrying J1a and one from pedigree 3 carrying J2b. In fact, pedigree 2 is the only one spanning from the early to the late phase, continuing the only two remaining J1a descent lineages after this shift. The haplotype IBD network in RK shows an even clearer pattern, indicating the community shift, because all of the different J1a-carrying pedigrees and the J2b pedigrees 3, 4 and 5 share more IBDs within them than between them, forming two clearly distinct clusters separating J1a and J2b male individuals (Fig. 3a).

Interestingly, one generation above (generation 4, dating to the middle Avar period), in pedigree 2 there are 12 related male individuals, of whom only three had children buried at the site. All the remaining male individuals except two juveniles (aged 18–22 and 15–17) were adults with no children found in the cemetery. This evidence further supports the replacement of the patriline in the community buried in RK. No associated skeletal traumas were observed in these individuals, so the shift in the male lineage cannot be clearly attributed to an act of violence. In fact, all of them are buried close to unrelated female individuals, who were potentially their exogamous partners, suggesting that the change of community occurred in the following generation of their children not buried at site. Given the strong patrilineality observed in all the sites we analysed, this change must have had strong social implications.

This shift mirrors the archaeological evidence (Fig. 2b and Extended Data Fig. 7). First, the western grave group of the RK site is made up of the large pedigree 2 and the several smaller pedigrees from the early Avar period. This part of the burial place was abandoned, and it is conceivable that many of the descendants of pedigree 2 left at that time. Except for a few scattered early burials, the central part of the cemetery was then established in the middle Avar period by the founders of pedigrees 4–5, along with the eastern part, which is mostly composed of

pedigree 3 and the latest group of individuals from pedigree 2, who are the ones biologically related to pedigree 3. New burial customs, such as wooden grave constructions, distinguish the graves of newly settled families, whereas the old ones, such as burial with a horse harness or a pot next to the deceased person's head, were phased out (Fig. 2b and Extended Data Fig. 7).

In RK, we also found significantly higher $\delta^{13}C$ and lower $\delta^{15}N$ values in the early Avar period than in the subsequent phases (Supplementary Fig. 25). Especially during the early phase, the carbon isotope data revealed a gradual change in dietary composition from substantial contributions of a $C_4$ component, which was probably millet, a primary staple crop in Eastern Asia[25], to the predominance of $C_3$ plants. Although millet was also consumed in the subsequent phases, individuals with outstandingly high $\delta^{13}C$ values are lacking in the middle and late Avar periods. The higher $\delta^{15}N$ values in the later phases indicate an increase in the consumption of meat and dairy products. However, largely overlapping ranges indicate that this affected only some of the individuals (Supplementary Information and Supplementary Figs. 25 and 26). Starting in the early phase, but especially in the middle and late phases, we observe a number of burials of male individuals with outstandingly high $\delta^{15}N$ values (Supplementary Figs. 25 and 26).

Taken together, these findings indicate that there was a replacement of the community buried, and thus likely to be living, in RK during the middle Avar period. Although the ancestry of the individuals and the descent system before and after the shift did not change (Fig. 4), the succeeding community differed in its burial customs and dietary habits. It is noteworthy that the HNJ and KFJ cemeteries were established in this later period, implying that larger transformations occurred in the Carpathian Basin in the second half of the seventh century[9,26].

## Steppe descent communities in Europe

Population-genomic analyses (Fig. 4, Extended Data Fig. 8, Supplementary Information and Supplementary Figs. 12–14) confirm that the four cemeteries belonged to communities with steppe descent. Most (88%) of the individuals carry portions of a northeast-Asian ancestry profile that is ultimately traceable to the eastern Eurasian Steppe[10,23,27] with varying degrees of admixture with western Eurasian sources. The northeast-Asian ancestry ranges from a median of about 100% in the DTI site of KUP to just 32% in the TT site of RK (Extended Data Fig. 8c and Supplementary Information). Independent evidence from the admixture modelling of qpWave/qpAdm and admixture dating of DATES reveals a process of continuous admixture between western and eastern sources over centuries that largely predates the Avar period, and therefore the arrival of these populations in the Carpathian Basin, and presumably took place in the steppe (Extended Data Figs. 8 and 9 and Supplementary Information). These analyses suggest that the post-arrival admixture with the local contemporary (post-sixth century) Carpathian Basin population was around 20% (Extended Data Figs. 8 and 9 and Supplementary Information).

Despite these clear patterns confirming a recent steppe origin for these populations, the strontium-isotope compositions ($^{87}Sr/^{86}Sr$) were largely similar, with values consistent with local and regional variations[28] (Supplementary Fig. 23). The datasets from KUP and KFJ were isotopically indistinguishable from one another, whereas the burials at RK yielded substantially more radiogenic strontium, indicating a small variation of the local baseline values (Supplementary Information). This homogeneity indicates that although local mobility (within the Great Hungarian Plain, for example) is plausible, migration between distant areas (such as across the Eurasian Steppe) is unlikely because they would always need to be isotopically indistinguishable. This implies that, with one potential exception (Supplementary Information and Supplementary Fig. 23), the first generation of migrants was not buried at the cemetery, and that there was high regional continuity across the Avar period.

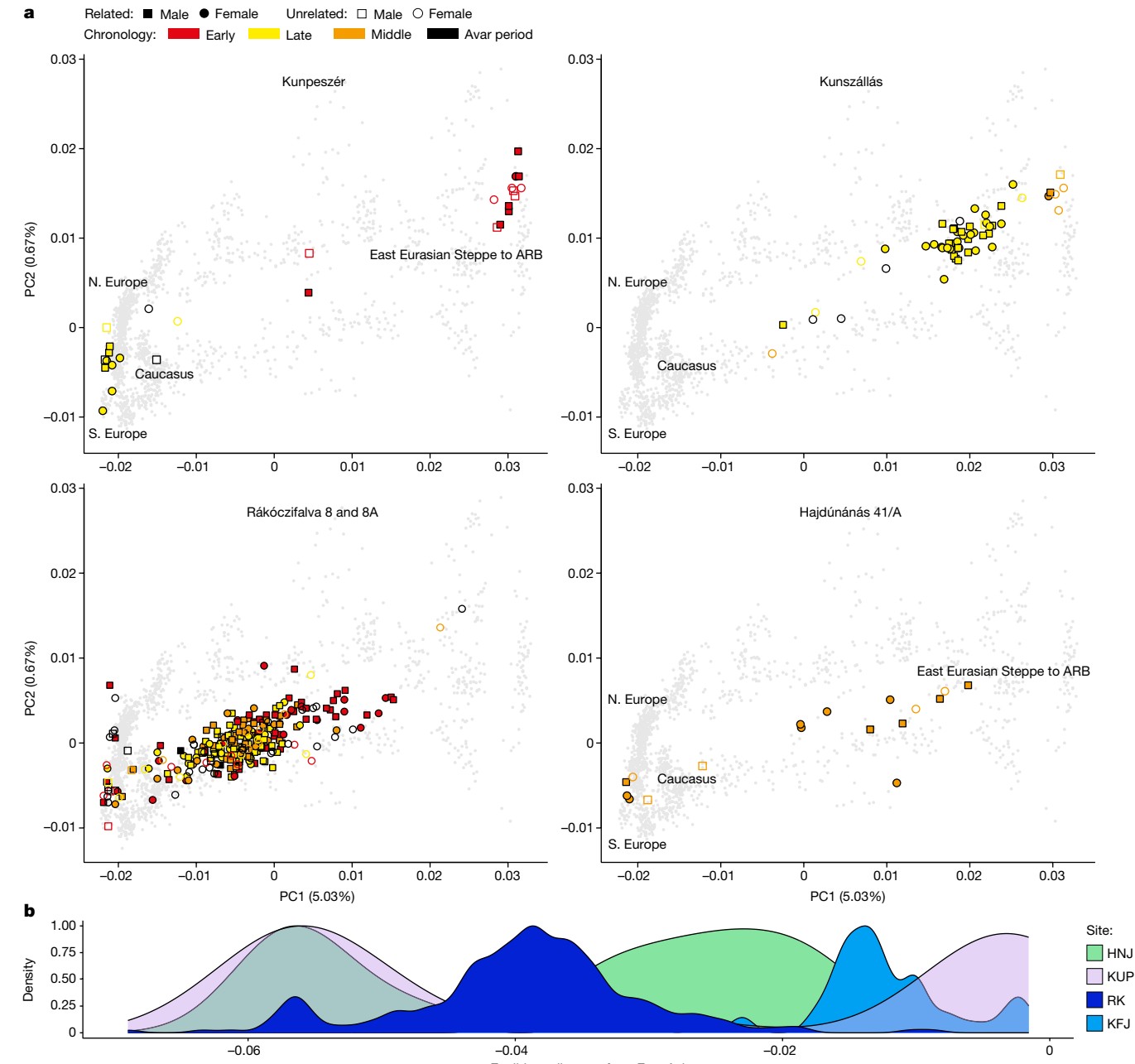

**Fig. 4 | PCA for newly sequenced Avar-period individuals. a**, 'Eurasian PCA' (principal component analysis; see Methods) for each of the four sites. Modern individuals used to calculate the PCA are shown as grey dots. The variance explained by the first two principal components (PC1 and PC2) is shown in brackets. The approximate geographical locations of the most-relevant modern individuals are shown: northern (N.) and southern (S.) Europe, the Caucasus region and the eastern Eurasian Steppe (EES) to the Amur River

Basin (ARB). Ancient individuals are highlighted by symbols coloured by period (early, middle and late) and black when dated generally to the Avar period: filled coloured symbols represent individuals who have at least one close genetic relative at the site (first or second degree), and empty symbols indicate unrelated individuals. **b**, Site-based density plot of Eurasian PCA Euclidean distance of the first three PCs of each individual to the PC coordinates of the Rouran genome, used as a proxy for a non-admixed EES ancestry.

The density of sampling allowed us to uncover a pattern of geographic structuring of genomic ancestry that went unnoticed in previous studies. This is most evident between the DTI sites (KUP and KFJ) and the TT site of RK, which cover the whole Avar period. However, their admixture profiles remain largely distinguishable and non-overlapping (Fig. 4b). In fact, although KFJ has high amounts of northeast-Asian ancestry in the late Avar period, RK individuals, even in the early phase, carry admixed ancestries, 95% of which were dated pre-arrival (Extended Data Fig. 9). This is in line with the observation of higher IBD sharing within DTI and TT than between the two regions (Extended Data Fig. 6a). These

differences mirror cultural differences found in the archaeological record. Several features of the TT sites, especially at RK (Supplementary Information), are strikingly similar to those of nomadic burials of the sixth to seventh century in the Pontic Steppe, known as the Sivashovka horizon[29] (Supplementary Fig. 3). Instead, some cultural elements of the elite of the DTI area, which includes the early graves of the KUP site, can be traced back to the eastern Eurasian Steppe[30–32].

In conclusion, we confirm the arrival and establishment in the Carpathian Basin of entire communities of steppe descent. We reveal that genetically and culturally distinct steppe communities settled

in the area and, despite some admixture with the local population, remained distinct during the course of the Avar period. This substantial post-arrival genetic continuity, together with striking isotopic homogeneity over time, poses a challenge to the long-lasting archaeological hypothesis[1,33] that there were successive large-scale migrations from the steppe, indicating instead a pattern of local, small-distance mobility once settled.

## Conclusions

The reconstruction of extended multigenerational pedigrees from four Avar-period sites indicates a consistent reproductive strategy based on patrilineal descent, patrilocality, female exogamy, strict avoidance of consanguinity, and, in several cases, multiple reproductive partners and the practice of what seems to have been levirate unions. We found indications that social and biological relatedness overlapped to a large degree, because patterns of biological relatedness corresponded to the spatial distribution of the graves and grave goods. These social practices survived political changes, shifts in lifestyle reflected in material culture, dietary changes, and interactions with the local population from the late sixth century to the early ninth century AD. Descent units were strictly organized around patrilines but on a larger scale were connected by exogamous female individuals, and these connections may have been one of the main cohesive elements of Avar society. Mostly small pedigrees, of two to four generations, were found in the early phase, and larger ones, of four to seven generations, started in the mid-seventh century. This change reflects the increasing size of cemeteries and settlements since the middle Avar period and the development of the early medieval settlement system in the Carpathian Basin. The largest site we analysed (RK) experienced a community shift in the second half of the seventh century, which was probably caused by a realignment of local power, but it had no effect on the social organization or general ancestry patterns. Detecting this shift required the reconstruction of a biological-relatedness network of the entire cemetery and shows that genetic continuity at the level of ancestry might still conceal the replacement of whole communities.

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

## Methods

### Ancient-DNA laboratory analyses

For the archaeogenetic investigations, petrous bones and teeth were preferentially sampled (Supplementary Table 1). Samples were prepared in dedicated ancient-DNA laboratory facilities at the HUN-REN RCH Institute of Archaeogenomics in Budapest. Sample surfaces were decontaminated using UVC light and cleaned by mechanical removal. About 25–50 mg bone powder was obtained by drilling or powdering and transferred to MPI-EVA in Leipzig, Germany. DNA extraction and subsequent laboratory steps were done in the Ancient DNA Core Unit of the MPI-EVA. DNA was extracted from between 25 mg and 52 mg of powdered sample material using a silica-based method optimized for the recovery of short DNA fragments[35]. Briefly, lysates were prepared by adding 1 ml extraction buffer (0.45 M EDTA, pH 8.0, 0.25 mg ml$^{-1}$ proteinase K, 0.05% Tween-20) to the sample material in 2.0-ml Eppendorf LoBind tubes and rotating the tubes at 37 °C for approximately 16 h[35,36]. Using an automated liquid-handling system (Bravo NGS Workstation B, Agilent Technologies), DNA was purified from 150 µl lysate using silica-coated magnetic beads and binding buffer D, as described previously[36]. Elution volume was 30 µl. Extraction blanks without sample material were carried alongside the samples during DNA extraction.

DNA libraries were prepared from 30 µl extract using an automated version of single-stranded DNA-library preparation[37] described in detail previously[38]. *Escherichia coli* uracil–DNA–glycosylase (UDG) and *E. coli* endonuclease VIII were added during library preparation to remove uracils from the interior of molecules. Libraries were prepared from both the sample DNA extracts and the extraction blanks, and further negative controls (library blanks) were added. Library yields and efficiency of library preparation were determined using two quantitative PCR assays[38]. Libraries were tagged with pairs of sample-specific indices by PCR extension using AccuPrime *Pfx* DNA polymerase as described previously[38]. Indexed libraries were amplified and purified using SPRI (solid-phase reversible immobilization) technology[39] as described previously[38].

Sample and control libraries were enriched in solution for 1,237,207 informative SNPs (a method commonly used in the field and known as 1240k capture[40]) targeting 394,577 SNPs first reported in ref. 41 (390k panel) and 842,630 SNPs first reported in ref. 42 (840k panel). Two consecutive rounds of 1240k capture were performed using the Bravo NGS workstation B. Up to 20 libraries were pooled together and sequenced single-read or pair-read on a HiSeq4000 sequencing platform (Illumina Technology). In total, 440 1240k-enriched libraries were sequenced and an average coverage of 2.6× (median 2.25×) for the 1,237,207 sites in the genome, corresponding to a median of 708,514 1240k SNPs covered at least once (Supplementary Table 1).

### Ancient-DNA data process and quality controls

The raw sequenced read data (fastq files) were processed through a nf-core/eager v.2.3.2 pipeline[43] (https://nf-co.re/eager). To remove adaptors and short reads of less than 30 base pairs, AdapterRemoval v.2.3.1 was used[44]. The reads were then mapped to the Human Reference Genome Hs37d5 using the bwa v0.7.17 aln/samse alignment algorithm[45] with the parameters -n and -l set to 0.01 and 1,024, respectively. The reads with phred mapping quality of less than 30 were then discarded using -q (q30-reads) in Samtools v1.9 (ref. 46). We then used the Picard tools MarkDuplicates function (https://github.com/broadinstitute/picard) to remove PCR duplicates. To estimate the amount of cytosine-to-thymine taphonomic deamination at the ends of the mapped fragments, we used mapDamage v.2.0 (ref. 47) run on a subset of 100,000 q30 reads. Exogenous human autosomal DNA contamination was estimated in male individuals by assessing X-chromosome heterozygosity levels using ANGSD v.0.910 (ref. 48) and mtDNA contamination in male and female individuals was estimated using Schmutzi[49]. Schmutzi was also used to reconstruct the consensus mitochondrial genome sequence of each individual used as input for HaploGrep2 (ref. 50) to assign mitochondrial haplogroups. For the purpose of graphical representation in Extended Data Fig. 4, all the mitochondrial haplogroups were pruned to the first three characters. If two individuals had, respectively, a two- and three-characters resolution, both of their haplogroups were trimmed to the first two characters. Individuals with only a one-character resolution were excluded from the plot.

Y-chromosome haplogroups were inferred using two different methodologies and the results compared. The Y-chromosome variants were called from in the bam files from samples whose genetic sex was estimated to be male or unassigned using the Samtools v1.9[46] mpileup and PileupCaller (https://github.com/stschiff/sequenceTools) using the mode --majorityCall; Y-chromosome haplogroup assignment was performed using the software yHaplo (https://github.com/23andMe/yhaplo), with ISOGG panel v.11.349 as a reference (https://isogg.org/tree/; date of access: 2 February 2023). Y-chromosome haplogroups were also defined using the Y-Lineage-Tracker subcommand 'classify'[51], using as a reference panel the ISOGG Y-haplogroup tree v.15.73 (https://isogg.org/tree/); in this case the input files were genotypes from each individual, estimated using the allelePresence method from the ATLAS (https://bitbucket.org/wegmannlab/atlas/)[52] call tool, accounting for post-mortem damage patterns and base-score recalibration patterns, estimated respectively with the ATLAS tools PMD and recal.

The results from the two methodologies were then compared, taking into account the differences between the two reference panels. In cases where the two methodologies yielded deeply diverging results (that is, to the first two ISOGG alphanumeric classification symbols) or were discordant with the estimated reciprocal genetic relatedness between individuals (described in the Biological relatedness section), the haplogroup assessment was further investigated using the software pathPhynder[53] with default options, using as reference the BigTree Y-chromosome dataset and the reference phylogenetic tree for sample placement provided by GitHub with the software and as input files the bam files filtered for phred mapping quality more than 30. In any other case, the conservative results from Y-LineageTracker (the column Key haplogroup) were considered reliable, given the more-stringent estimation of the genotypes and the updated ISOGG Y-chromosome phylogenetic tree version.

The results of the whole procedure can be found in Supplementary Table 1. PileupCaller (https://github.com/stschiff/sequenceTools) was used to carry out genotype calling from the q30 reads with the --randomHaploid flag that calls haploid genotypes by randomly choosing one high-quality base (phred base quality score ≥30) on the 1240k panel (pseudodiploid calls). We also used the --singleStrandMode, which removes only real cytosine-to-thymine deamination observed with single-stranded DNA libraries by ignoring cytosine–thymine polymorphisms at reads aligning to the forward strand and guanine–adenine polymorphisms at reads aligning to the reverse strand.

To produce the Y-chromosome haplogroup plots in Extended Data Fig. 4, all the haplogroup nomenclature was pruned to the first three characters; haplogroups with less than three characters of ISOGG notation were excluded from the plots. Complete Y-chromosome haplogroups can be found in Supplementary Table 1.

We found low mitochondrial contamination estimates (Supplementary Table 1). Most were less than 5% and only five samples had values between 5% and 10%. Of these we excluded one female individual (RKF048) with 7% contamination and one individual (KFJ019) with 5% contamination and ambiguous sex determination (an indirect sign of possible contamination); the remaining male individuals had low nuclear contamination and were therefore kept for nuclear genomic analyses. We also found low nuclear contamination estimates among the male individuals. We excluded four further individuals with values of more than 7%; RKF094 (15% contamination) was still counted among the related as showing high likelihood of close genetic relatedness with other individuals (Supplementary Table 4). We also excluded

individuals with particularly low coverage (more than 20,000 SNPs) because they were not practically usable for further analyses (additional filtering for higher coverage thresholds is detailed for specific analyses in the following sections); these include two individuals also excluded for contamination and another 15 individuals still included as showing high likelihoods of close relatedness (RKF225, HNJ005, HNJ009 and RKF128). We kept 419 individuals for further analyses, 413 excluding one pair among the identical pairs found, and 424 including the previously published individuals from the KUP and KFJ sites[10] (Supplementary Table 1). We then merged them with a reference genome-wide panel of 2,280 modern individuals genotyped with microarray technology using the commercial HumanOrigins chip[54–56] and previously published ancient-individuals' genotypes sequenced with the same 1240k capture method or a 1240k SNPs subset from data obtained using whole-genome shotgun sequencing[10,27,54,55,57–74] downloaded from Poseidon (https://poseidon-framework.github.io). We produced two datasets, one including the modern data and the SNPs overlap between the 1240k sites and the HumanOrigins SNP chip (1240KHO dataset, around 600,000 SNPs), and one with ancient data and the whole 1240k panel (the 1240k dataset).

## Genomic ancestry modelling with PCA, qpWave/qpAdm, DATES

We used principal component analysis (PCA) with smartpca v.16000 in the EIGENSOFT v.6.0.1 package[75] on the 1240KHO dataset using the lsqproject and the autoshrink parameters to project the genotypes of the ancient individuals (containing variable amounts of missing data) on top of the principal components calculated on the set of modern worldwide populations. For one PCA (Fig. 4a) we used a subset of Eurasian populations (the Eurasian PCA) as originally in reference[54] adapted as in reference[27], and for another PCA (Extended Data Fig. 8b) we used a standard subset of only west Eurasian populations (the west Eurasian PCA), as originally reported[76] and then adapted[10].

We used the software qpWave/qpAdm (v.1520) of the ADMIXTOOLS package[56] to run the $f_4$-statistics-based ancestry analyses on the 1240k dataset[41,77]. Standard errors for the computed $f$-statistics were estimated using a block jack-knife with a 5-cM block. We used the default allsnps: NO parameter, thereby calculating all the underlying $f_4$-statistics using the SNP overlap between all the groups for each test. We used a set of outgroups (or right populations) that are similar to those of a previous study[10] that included representatives of ancient Eurasian lineages (European Mesolithic hunter-gatherers, European/Anatolia Neolithic, Levant Neolithic, Iranian Neolithic for western Eurasia, and ancient North Eurasian lineage (ANE[76]), ANA, ancient Siberian and southern East Asia for eastern Eurasia, and key non-Eurasian ones (African, South Asian, Native American) when available, otherwise hte present-day proxies Mbuti.DG, Levant_N, Onge.DG, Iran_N, Iron_Gates_HG, EHG, Mixe.DG, Anatolia_N, DevilsCave_N.SG, Tarim_EMBA1, Kolyma_M.SG and YR_LN. The only difference with respect to ref. 10 is that we used Tarim_EMBA1 (ref. 72) instead of the three Russia_Bolshoy individuals[65], which is a higher-coverage dataset of 12 individuals and a better representative of the ANE lineage[73] than any other high-ANE ancestry group available in the literature.

To select the sources (or left populations) to model the admixed ancestry of our newly sequenced individuals (the targets), we followed the following rationale. Among the data available from previous studies, we selected only ancient populations (of more than two individuals) that are either approximately contemporaneous or temporally preceding but are as close as possible to the time period of our target individuals, as suggested previously[78]. In our selection, we also considered the findings from a previous genomic study of the Avar period[10], as well as populations that are geographically, historically and archaeologically relevant. This led to a selection of 13 different source groups falling in 3 categories. (1) Sources representative of the east Eurasian Steppe ancestry that include ancient populations and cultures available from preceding time periods in the east Eurasian

Steppe and surrounding areas in east Asia. (2) 'Pre-Avar' populations that are found in the Carpathian Basin in the first centuries AD, before the Avar period. (3) Relevant temporally preceding (first millennia BC and AD) populations available from across the Pontic- and central Asian Steppe (the 'steppe' sources).

Two- and three-way combinations of these sources led to a total of 190 different combinations being tested, all with qpWave $P$-values of much less than 0.05, which means that the sources are sufficiently differentiated with respect to the set of outgroups. They are therefore suitable sources to be tested[76] (Supplementary Table 5), applying the following rationale, which is the same as that used in a previous study[10] based on suggestions discussed previously[78]. We first tested two-way admixing sources using all combinations of eastern Eurasian Steppe groups plus the pre-Avar and steppe sources. If we could reject one but not the other, between the pre-Avar and steppe source models (if one had $P < 0.05$ we can reject; if the other had $P > 0.05$ we cannot reject), we considered the one we cannot reject ($P > 0.05$) as valid. If the two-way models did not significantly reject one or the other between the pre-Avar and steppe sources (both with $P > 0.05$) or produced no fitting results at all (both with $P < 0.05$), we proceeded by testing three-way competitive models, including the eastern Eurasian populations and contrasting directly the pre-Avar plus steppe sources as well as pre-Avar plus pre-Avar, accounting for the variability in ancestry and time period between the pre-Avar populations.

If the three-way models resulted in one of the two contrasting sources between pre-Avar plus steppe resetting the other (bringing its estimated admixture proportion to 0%), we considered these models. If the contrasting sources had intermediate admixture proportions, we considered as successful only those tests that could reject one of the two scenarios between either pre-Avar plus steppe or pre-Avar plus pre-Avar. The individuals who still had unresolved or non-fitting models between a pre-Avar or a steppe source were considered unsolved or failed and were not used for further meta-analyses or interpretations.

For the sake of simplicity and consistency, we chose one eastern Eurasian source to include in our plots and summary statistics: the genetically easternmost group of individuals from the early Avar period in the DTI region (DTI_EA_East; Fig. 4 and previously published[10]), to which we added data from unrelated individuals at the early-period site of KUP that presented the same genomic profile (Supplementary Fig. 12). We always used this eastern proxy, except in the few instances in which it did not produce fitting models, in favour of another one, suggesting an existing heterogeneity in the eastern component although much reduced with respect to the variability in the western sources (Supplementary Table 5). Nevertheless, it is important to note that although DTI_EA_East is the source that overall produced more fitting models, several other eastern sources (including lateXiongnu, AR_Xianbei_P_2c) resulted in many equally fitting models as well (Supplementary Table 5).

We used DATES v.753 (https://github.com/priyamoorjani/DATES) to date the average time of the east–west Eurasian ancestry admixture estimated for most of the Avar period individuals from the four sites analysed. This method is based on the same principle as many admixture dating methods[70,79]; it assumes an admixture event between two admixing source populations, an east Asian and a west Eurasian one; in our case we used the unadmixed and high-SNP-covered LBA/IA group of the Ulaanzuukh_SlabGrave in Mongolia[63] or the same DTI_EA_East group used in the main qpAdm models as an ANA proxy and the pre-Avar Carpathian Basin ancient sources, Sarmatian[10] and Longobard period[58] individuals, as a west Eurasian ancestry proxy. DATES calculates the decay of ancestry covariance coefficients between every pair of available overlapping SNPs between the test individuals and the source populations over increasing-genetic-distance windows[70]. Population-genetic theory suggests that if admixture happens, an exponential function can be fitted to the decay of weighted ancestry covariance, and the number

of generations since admixture can be derived from the parameters of such functions[79]. The higher age limit of admixture events that would still produce detectable decays is theoretically considered to be around 4,000 years[80]. In practice, recent admixture events (about one to three generations ago) are not properly detected because chromosomal recombination had insufficient generation time to start producing the expected decay pattern[81,82]. To estimate the goodness of a fit, DATES calculates standard errors and $Z$-scores using a jack-knife approach, dropping a chromosome at a time. We set a maximum distance parameter of 0.5 cM, a bin size of 0.001 and a starting genetic distance of 0.45 cM. The integrated least-square function was used to estimate the number of generations since admixture parameter. If the raw data show no decay, the exponential function either cannot be fitted or is fitted with low $Z$-scores, much less than 2, and unreasonable dating estimates with negative values, or large numbers over the theoretical maximum of 4,000 years back in time. All samples showing such values were also inferred as non-admixed by PCA and qpAdm and were excluded from our inferences. For Extended Data Fig. 9, we also included dates with $Z$-scores of less than 2 (shown with a transparency factor) because in part they reflect the recent (for example, first or second generation) admixture events that we can observe directly in the pedigrees. These DATES estimates are mostly not significant because there is no decay pattern yet to fit an exponential function, but some still provide qualitatively correct recent admixture dates (Supplementary Table 1). We used a standard of 29 years per generation[70] to convert the generation times in years since admixture, and used the Avar-period chronological phase of the individuals as the date at death.

## Biological relatedness

We used KIN[16] as the primary method to assess biological relatedness between each pair of individuals from the four sites we investigated, although we validated the relatedness estimates with the independent methods of haplotype-IBD (detailed below) and BREAD (https://github.com/jonotuke/BREADR) (Supplementary Information). Given that single-stranded UDG-half treated libraries still preserve a roughly 10–30% proportion of C-to-T deamination at the last two base pairs of the mapped fragments, for this analyses we masked two base pairs at both ends of the q30 reads using the trimBam module of bamUtil v.1.0.13 (ref. 83) and used these masked bam files as input data. KIN can confidently identify first- and second-degree relations while differentiating between parent–child and sibling relations[16]. Although the method does not explicitly differentiate relationships within the second degree, it outputs information about IBD sharing that can help to differentiate between avuncular and grandparent–grandchild relationships. We simulated avuncular, half-sibling and grandparent–grandchild pairs (Supplementary Information) to show that the length of IBD segments and the number of IBD segments can be used to differentiate between avuncular and grandparent–grandchild relationships, while half siblings overlap with both cases. Furthermore, KIN provides indications about third-degree relationships (with around 70% accuracy at 4× sequence coverage). Although these analyses are not sufficient to confidently identify within second-degree relationships, and may lack the power to identify third-degree relatives, they can be crucial when combined with other information, such as pedigree information from different pairs as well as from information about the skeletal age at death, the sex and the uniparental haplogroups (Y chromosome and mtDNA). Therefore, all this information was considered when building and cross-checking the pedigrees of biological relatedness (Supplementary Information). For clarity, we numbered the pedigrees that we found and we define one pedigree as a group of individuals who can be directly connected with close genetic relatedness and for whom a line of descent can be traced. In the case of the largest pedigree we reconstructed (146 individuals from RK), we divided it into five pedigrees descending from five different groups of 11 'founder male individuals' (including multiple brothers as co-founders).

## Simulations on second-degree relationships

We followed the methods section for KIN[16] and simulated eight diploid individuals using msprime[84] with default parameters for the mutation rate ($1 \times 10^{-8}$ per base per generation), the recombination rate $r$ ($1 \times 10^{-8}$ per base per generation) and an effective population size of 3,000. For each individual, we simulated 22 chromosomes with the same lengths as the GRCh38.p14 genome. To form a pedigree, we first simulated a recombined set of chromosomes for each parent and combined them to create the progeny. We obtained recombination points for each chromosome from the software Ped-sim[85]. We matched the genotype density and the coverage of reads to that of our samples. We simulated 60 such pedigrees (see figure S9 in ref. 16 and Supplementary Figs. 17 and 18).

## Consanguinity test (ROHs)

Consanguinity can be tested genetically by a straightforward approach: counting the length and number of long stretches of homozygous portions along the genome of an individual. This analysis is usually defined as ROHs. To estimate ROH, we applied a method called hapROH[86] that was designed to infer them on pseudo-haploid, lower-coverage and higher-missing-data ancient DNA samples; the method has also been shown empirically to be highly consistent with independent ROH estimates calculated on the same ancient imputed diploid genomes[10]. Specific patterns of long ROH (more than 4 cM) along the genome of an individual are typical of consanguineous unions between some of its recent ancestors (up to second-degree cousins[86]). In Extended Data Fig. 5 we plotted ROH using the python package implemented in hapROH (https://pypi.org/project/hapROH/).

## Genotype likelihood calls and imputation/phasing

Haplotype-based analyses (such as IBD described below) require information of the phase for each pair of paternal and maternal chromosomes of an individual, and this in turn requires there to be virtually no missing data along the genome. Obtaining such data from ancient genomes has been shown by recent studies[87,88] to be reliable in other similar contexts for coverage of more than 0.5–0.7×, and it has also been applied to 1240k capture data[10,89] through simultaneous statistical imputation and phasing. We used the ancient-DNA-specific genotype caller MLE function of ATLAS (https://bitbucket.org/wegmannlab/atlas/)[52] to call genotype likelihoods. ATLAS can also calculate the base-quality recalibration (the recal function) that we performed in batches among libraries sequenced in the same sequencing run, accounting for specific sequencing errors. ATLAS recalibration also corrects the base qualities accounting for the empirical ancient DNA-damage pattern observed from the data and reduces the effect of reference bias introduced by genome mapping by relying on a list of 10 million highly conserved genomic positions across 88 mammal species downloaded from ensembl (https://grch37.ensembl.org/). We called genotype likelihoods on the whole 1,000-genomes SNPs panel of around 20 million SNPs and used these calls as input data for imputation with GLIMPSE[90], for which we used the phased 1,000 genomes phase-3 release data as reference haplotypes[91]. We ran GLIMPSE with the default parameters using sex-averaged genetic maps from HapMap, as suggested previously[88]. The function GLIMPSE_phase was used to perform simultaneous imputation and phasing on genomic chunks of 2,000,000 base pairs with a buffer of 200,000 base pairs. We then used the integrated GLIMPSE_ligate and GLIMPSE_sample functions and bcftools v1.3 (refs. 88,92) to obtain the final phase/imputed vcf files with the genotypes posterior probabilities at every 1240k position.

## Haplotype IBD sharing analysis

We performed haplotype IBD analysis with ancIBD, a recently developed method that accounts for the high phasing errors of ancient DNA[93]. This analysis searches for long haploid blocks along the genomes

of two individuals that are identical by descent (IBD), meaning they have been inherited by a common ancestor at some time in the past. Therefore, it can detect close genetic relatives (first to third degrees of relation) as KIN does, but it can also detect more-distant relations, up to sixth degree, within ranges of biological stochasticity[85]. However, it requires a much higher threshold of coverage, reducing the number of individuals analysed relative to KIN. We used imputed or phased data, including only those individuals with more than 450,000 SNPs obtained with our pseudo-haploid calls and SNPs with genotype posterior probabilities greater than 0.99 after imputation. We used the HapBLOCK function of ancIBD to perform the pairwise estimation with default parameters and only shared blocks of more than 8 cM containing more than 220 SNPs per centimorgan were considered. To further filter for possible false-positive hits, we considered only shared IBD segments longer than 12 cM, and if a pair of individuals had segments of less than 16 cM, we included them only if they had more than one such segment (Supplementary Table 4). We used Cytoscape v.3.9.1 (ref. 94) to plot the networks of pairwise IBD relations.

### Network analysis

For the IBD network analysis, only the Avar-period individuals were included. Because subadult individuals might be a confounding factor when assessing the sex-specific patterns of mobility and connectedness, we made an additional network that included only adults. The threshold of adulthood was set at 18 years of age, based on the lower limit of the estimated age of the youngest parent. The entire network consisted of 257 nodes, of which 195 represented adults (105 male and 90 female individuals) and 62 subadults (35 male and 27 female individuals) from four archaeological sites. The links of the network are represented by the IBD connections, which number 2,658 if the entire network is considered and 1,211 if only the adults are selected (Supplementary Table 4). In our analysis, we considered both unweighted and weighted networks. The unweighted network represents a configuration in which the found IBD relations define the presence or absence of links irrespective of their values. However, in the weighted network, the links are weighted by the maximum IBD values of the analysis, allowing the magnitude of relatedness to be evaluated. Both networks are undirected because sharing of IBD segments between two individuals has no directionality.

Degree centrality ($k$) is defined as the number of links held by the node. The average degree $\langle k \rangle$ of the Avar-period adults' network is 18.07. Considering the assigned weights on the links, which in our case is the sum of the weights (max_IBD) of the links attached to each node, the mean strength $\langle w \rangle$ is 1,620.54. When sex is considered as a node attribute, the degree and the strength distributions are significantly different between male and female individuals (Fig. 3c and Supplementary Figs. 47 and 48). For male individuals, $\langle k \rangle$ is 27.39 and $\langle w \rangle$ is 2,392.37, whereas for female individuals, $\langle k \rangle$ is 7.21 and $\langle w \rangle$ is 720.08. The two-sample Kolmogorov–Smirnov test revealed significant differences between the male and female individuals' degree and strength distribution ($P < 0.05$).

The degree centrality of a node can be partitioned into within-module ($k_W$) and between-module ($k_B$) links by considering the archaeological site of the burial as a module. The $k_B/k$ ratio represents the ratio of between-module connections over the total connections, which can range between 0 and 1, with 0 indicating that related individuals are buried solely at the same site and 1 indicating that related individuals are buried only at a different site. To evaluate this ratio, the value of degree centrality must also be considered because individuals with small degree centrality may have a higher $k_B/k$ ratio. The other results of the analysis are explained in Supplementary Figs. 44–48. The analysis was performed using R and the node measurements were calculated using customized R scripts with the igraph package[95].

### Isotope analysis and [14]C dating

[14]C dating and isotope analysis ($\delta^{13}C$, $\delta^{15}N$) was performed in the same bone material in the isotope and radiocarbon laboratories at the Curt Engelhorn Centre Archaeometry in Mannheim, Germany. Bone samples were cleaned, chemically treated and collagen extracted using a modified Longin method[96]. For stable isotope analysis of carbon and nitrogen, triplicates of the resulting collagen were combusted in an elemental analyser (PYROcube, Elementar) and isotopic ratios were measured by isotope ratio mass spectrometry (precisION, Elementar). The same collagen extract was used for [14]C dating. After ultrafiltration to remove short-chained macromolecules, the collagen was reduced to graphite using either a commercially available system (AGE3, IonPlus) or a custom-made system. A MICADAS-type accelerator mass spectrometer (IonPlus) was used to determine the conventional [14]C ages[97]. [14]C dates were modelled in the software Oxcal v.4.4.4 (ref. 98) and terrestrial samples were calibrated using IntCal20 (ref. 99). Bayesian modelling of [14]C dates include prior information of relative chronological information provided by pedigrees following methods outlined previously[24]. Model results and detailed explanations are given in Supplementary Tables 2 and 3 and Supplementary Information.

For all strontium measurements, the tooth enamel was extracted in a laboratory at the Institute of Archaeogenomics in Budapest. The surface of the teeth was cleaned by a Dremel tool with an abrasion tip, then, after a ten-minute ultrasonic bath, the enamel was carefully powdered with a diamond-coated dental drill bit attached to the Dremel tool, until 25–50 mg was obtained. Strontium separation chemistry for all samples followed a previous method[100]. Analyses were performed on a Nu Instruments NuPlasma HR at the MC-ICP-MS facility in the Department of Geological Sciences at the University of Cape Town in Rondebosch, South Africa, and followed the procedure and referencing values (SRM987 $^{87}Sr/^{86}Sr$ of 0.710255) described previously[101]. Past 4.11 software[102] was used for the statistical analysis of the isotope data.

### Reporting summary

Further information on research design is available in the Nature Portfolio Reporting Summary linked to this article.

### Data availability

The sequence data have been deposited in the European Nucleotide Archive (ENA) with the accession number PRJEB72021. The haploid genotype data are available through the Poseidon framework via GitHub at https://github.com/poseidon-framework/community-archive/tree/master/2024_GnecchiRuscone_CarpathianBasinAvarPedigrees (ref. 103). Geographic maps were plotted with R[34].

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

**Acknowledgements** We thank G. Csíky for work on the archaeological dataset; K. Sebők for help with the excavation documentation of the Rákóczifalva cemetery; A. Ben Rohrlach for advice on network statistical analyses; H. Ringbauer for sharing data, feedback and discussion on IBD analyses; and I. Rainer, P. Hofman and A. Plonka for graphical support. Data were produced by the Ancient DNA Core Unit of the Max Planck Institute for Evolutionary Anthropology, which is funded by the Max Planck Society. M. Spross, H. Beigzad, L. Schwarz, S. Lindauer, E. Dimitrakopoulos, E. Podolskaja, M. Hänisch and J. Wintel contributed to stable isotope analyses and radiocarbon dating at CEZA. S. Gábriel, D. Pokker, V. Bódis and K. Kerestély contributed to the DNA sample preparation in the HUN-REN RCH Institute of Archaeogenomics in Budapest. We thank the Hungarian Natural History Museum and the Department of Biological Anthropology of the University of Szeged for access to samples. This project received funding from the European Research Council under the European Union's Horizon 2020 research and innovation programme (grant 856453 ERC-2019-SyG), the Czech Grant Agency (GACR 21-17092X), the Czech Ministry of Education, Youth and Sports (CZ.02.01.01/00/22_008/0004593) and the Max Planck Society. T.S. was supported by the ÚNKP-22-4 New National Excellence Program of the Ministry for Culture and Innovation from the National Research, Development and Innovation Fund. The analysis of the pre-Avar radiocarbon data was supported by the Hungarian National Research, Development and Innovation Fund project 128035 led by Z.R.

**Author contributions** Conceived and led by: Z.H., J.K., T.V., W.P., P.G., G.A.G.-R., and Z.R. Formal analyses: G.A.G.-R., Z.R., L.S., T.S., N.F., C.K., R.F., L.T., D.Z., D.P., K.W. Sample preparation and laboratory work: R.R., B.G., I.K., C.B., G.M.L., V.M., O.S., M.M.E.B., P.R., A.S.-N., B.G.M., T.H. Visualization: G.A.G.-R., L.S., Z.R., T.S., N.F., C.K., R.F., L.T., D.Z., D.P. Writing, original draft: G.A.G.-R., Z.R., Z.H., W.P., L.S., S.L., S.W., T.S., N.F., C.K., R.F., L.T. Writing, reviewing and editing: J.K., T.V., P.G., H.C. with contributions from all authors.

**Funding** Open access funding provided by Max Planck Society.

**Competing interests** The authors declare no competing interests.

**Additional information**
**Correspondence and requests for materials** should be addressed to Guido Alberto Gnecchi-Ruscone, Tivadar Vida, Johannes Krause or Zuzana Hofmanová.

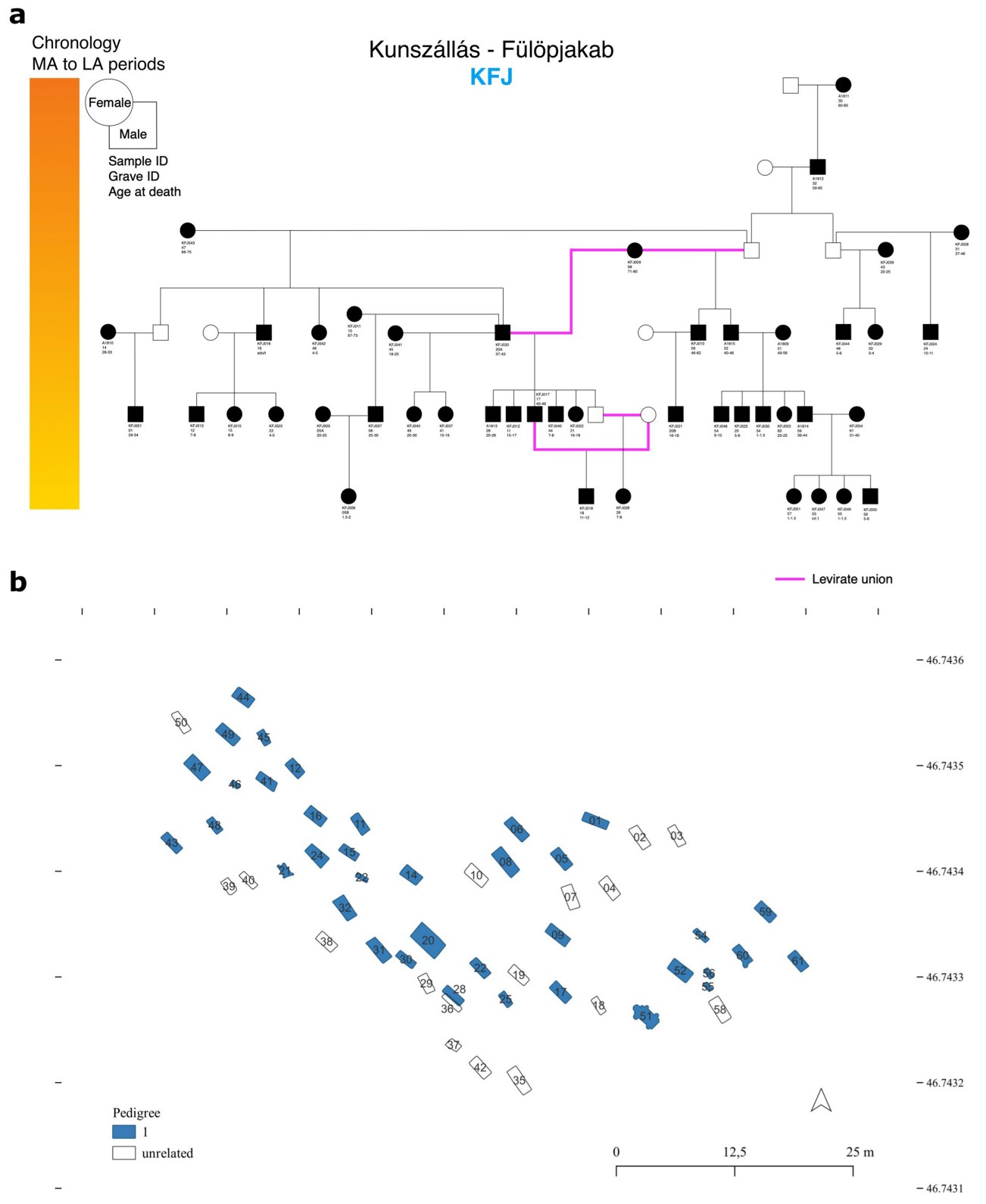

**Extended Data Fig. 1 | Pedigree and cemetery map of the individuals analyzed in the site of Kunszállás (KFJ). a**) pedigree highlighting the father-son levirate union discovered. **b**) cemetery map showing the burial location of the related and unrelated individuals in Kunszállás.

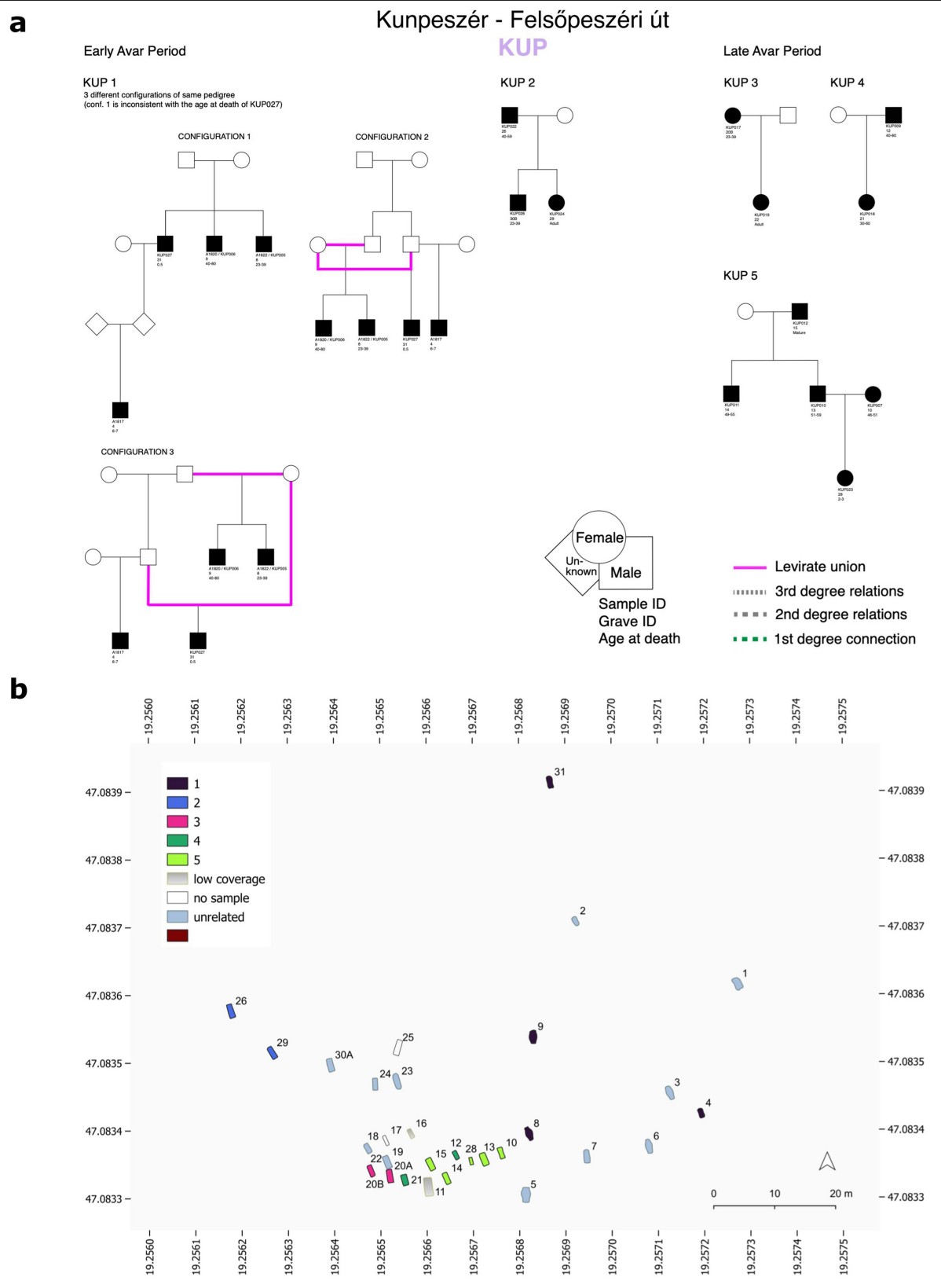

**Extended Data Fig. 2 | Pedigrees and cemetery map of the individuals analyzed in the site of Kunpeszér (KUP). a**) the unconnected early (left) and late (right) Avar period pedigrees highlighting the possible levirate union reconstructed for pedigree 1. **b**) cemetery map showing the burial location of the related and unrelated individuals in Kunpeszér.

# Hajdúnánás - Fürj-halom-járás 41/A
## HNJ

**a**

HNJ 1

HNJ 2

**b**

**Extended Data Fig. 3 | Pedigrees and cemetery map of the individuals analyzed in the site of Hajdúnánás (HNJ). a)** the unconnected admixed ancestry pedigree 1 and European ancestry pedigree 2. **b)** cemetery map showing the burial location of the related and unrelated individuals in Hajdúnánás.

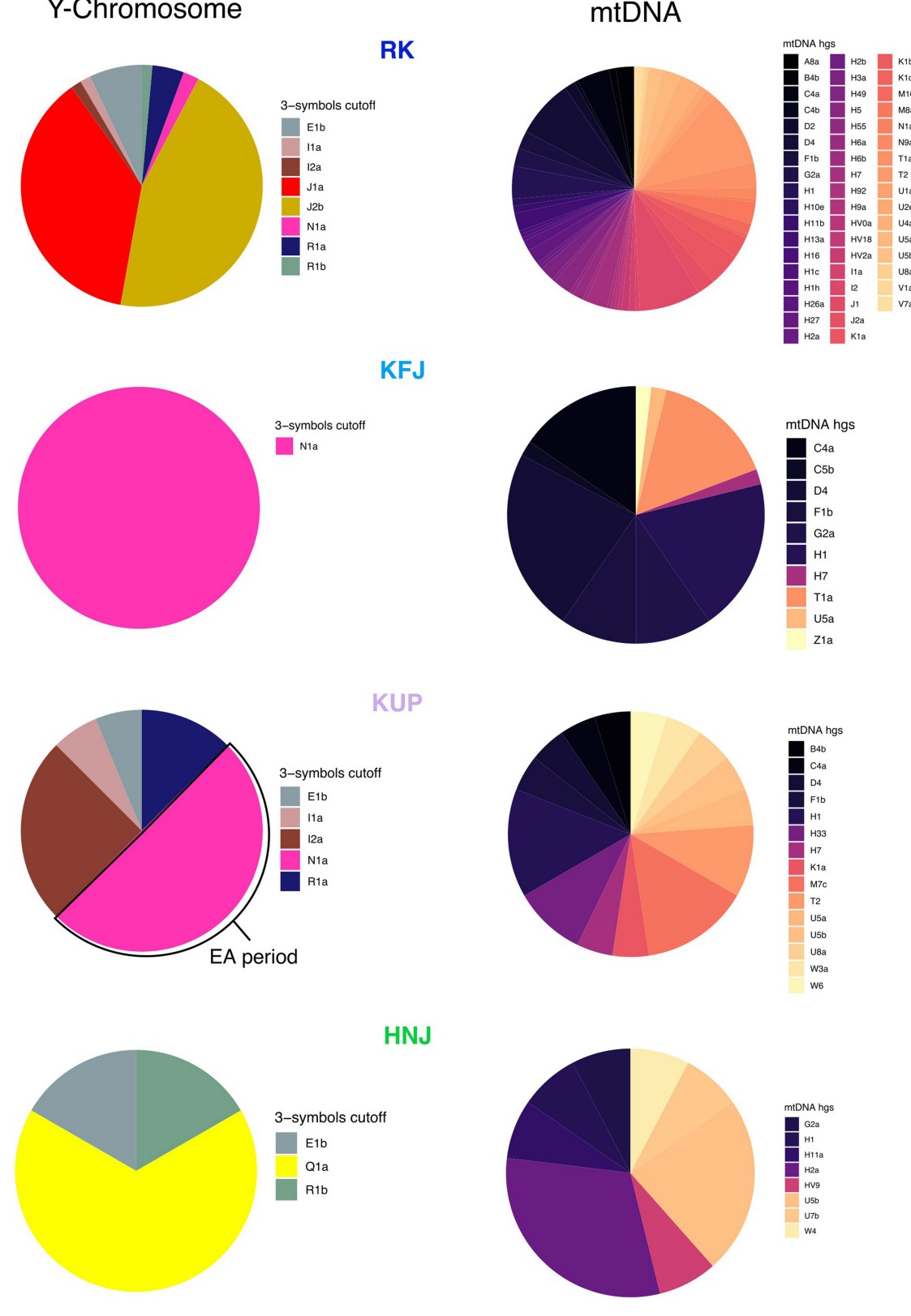

**Extended Data Fig. 4 | Pie-charts showing the frequency of the Y-chromosome and mtDNA haplogroups in the four Avar-period sites.** The four sites are dominated by one predominant Y-chromosome lineage (or two in case of RK) and the remaining ones are mostly restricted to outlier, unrelated individuals or smaller pedigrees not genetically related to the main ones whose patterns are analyzed in this article. While the mtDN-haplogroup diversity is much higher and more uniformly distributed.

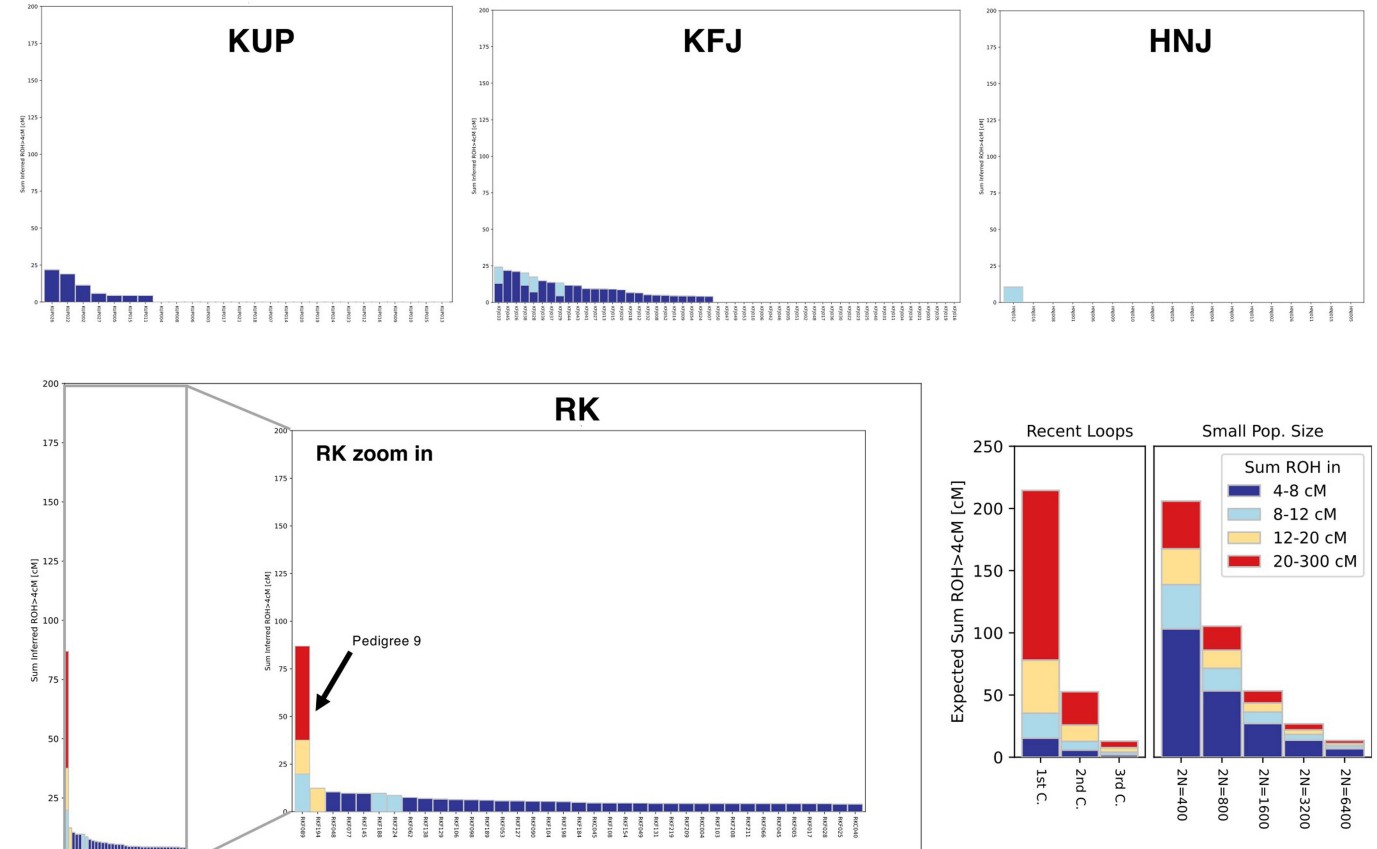

**Extended Data Fig. 5 | Runs of homozygosity (ROH), test for consanguinity.** all the Avar period individuals from the 4 sites are shown in 4 panels. The only individual that shows a pattern of long ROH consistent with its parents being relatively close relatives (possible 1st cousins) is a European ancestry individual found in the RK site, unrelated and unconnected through IBDs to the main extended pedigrees described in the article.

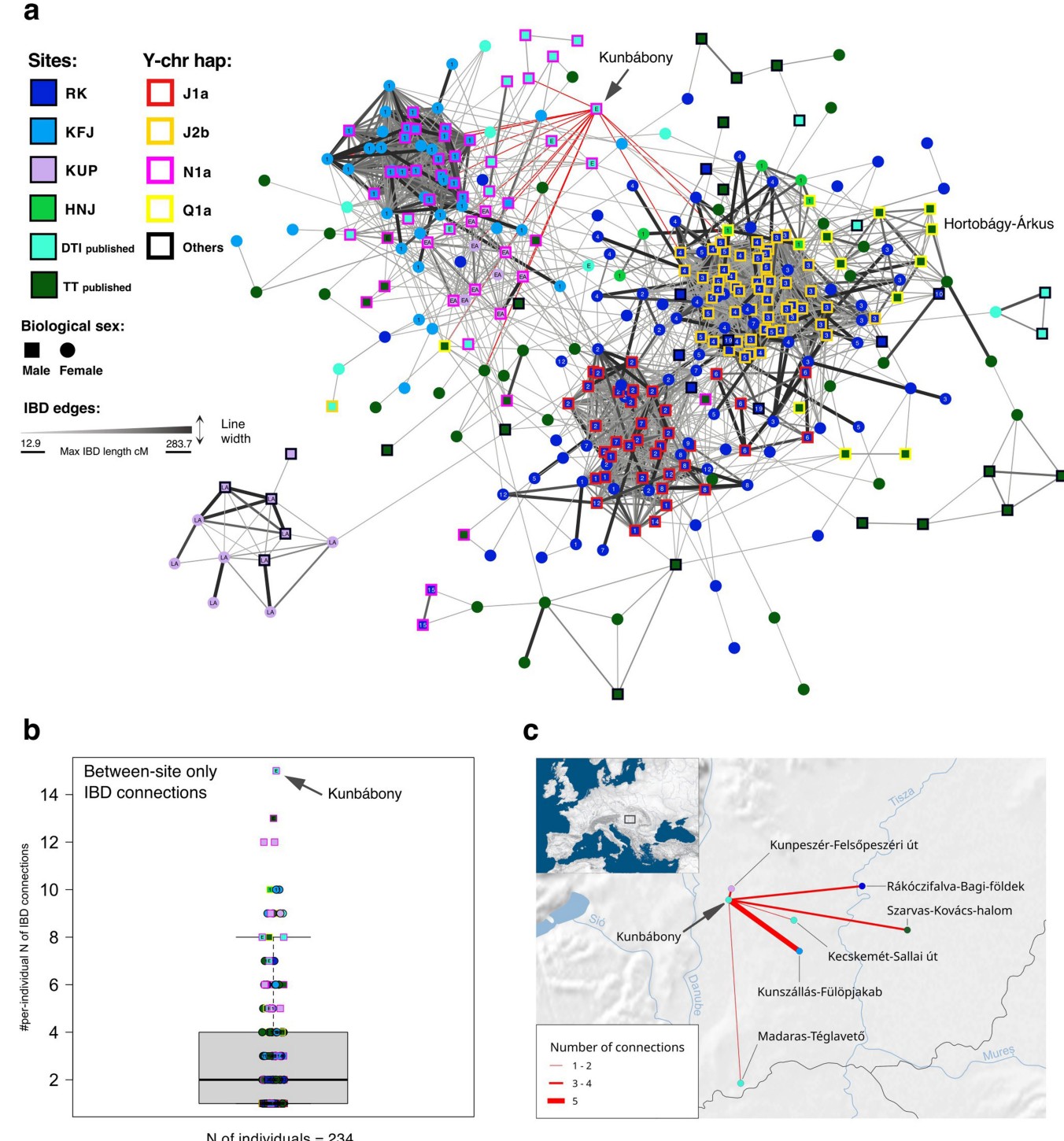

**Extended Data Fig. 6 | Network analysis of the ancIBD haplotype-IBD sharing between the Avar-period individuals analyzed in the study and previously published. a)** Visualization of the network of IBD connections (edges) between the ancient individuals (nodes) colored coded as in Fig. 3. The published individuals were retrieved from a number of different sites and are color coded to the regions where the site is located, DTI or TT. **b)** box plot showing the number of IBD connections for all the individuals from the network in a) having at least one IBD link with individuals from a site other than its own (237 individuals meeting this requirement). The plot shows all individual data points as well as the median (black line), upper and lower quartiles (the contour of the box) and the whiskers are the minimum and maximum values as calculated by the standard *boxplot()* function in R. **c)** map showing the sites with individuals connected to Kunbábony though IBDs. The thickness of the lines corresponds to the number of individuals connected. The base geographic map is from https://www.naturalearthdata.com and plotted with R[34].

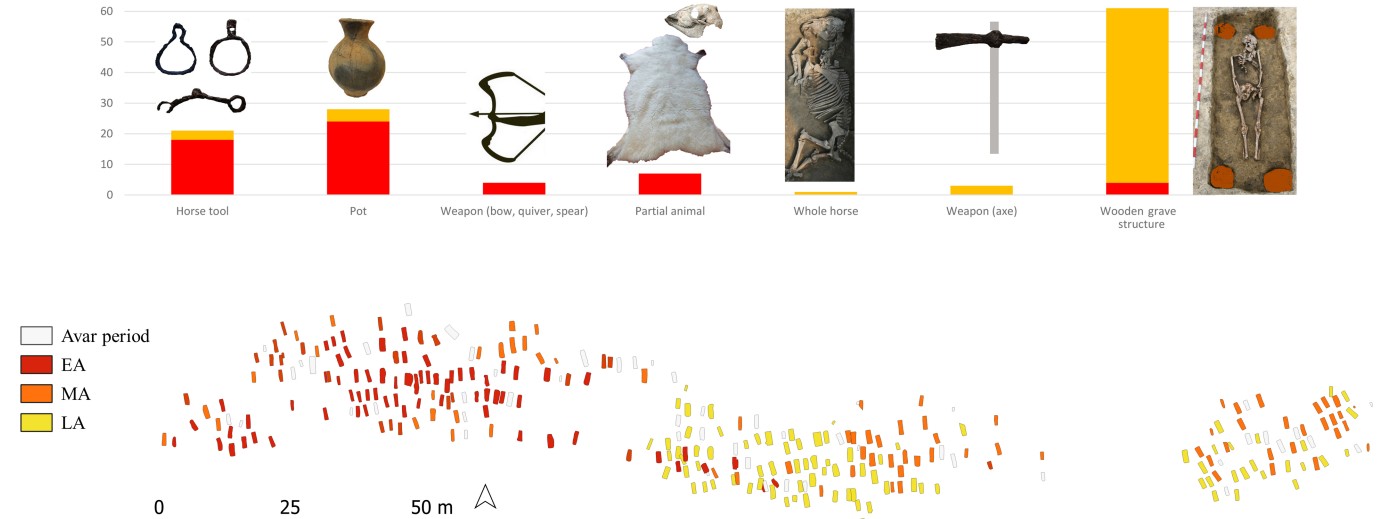

**Extended Data Fig. 7 | Summary of main burial customs and cemetery map of Rákóczifalva 8 and 8 A.** On the top, the main characteristics of burial customs in the early to middle and the middle to late Avar period in the Rákóczifalva (RK) cemetery. In the early phase, several people were buried with horse tools, pots, and animal skin (partial animal), while in the late phase, wooden grave structures were the dominant feature of burial customs. The changes in burial customs correspond to the community shift and the spatial organization of the cemetery. On the bottom, the cemetery map shows the distribution of early, middle and late Avar-period graves. The left part of the cemetery is where early to middle Avar period graves are found, while the middle and right part contain predominantly middle to late Avar-period graves.

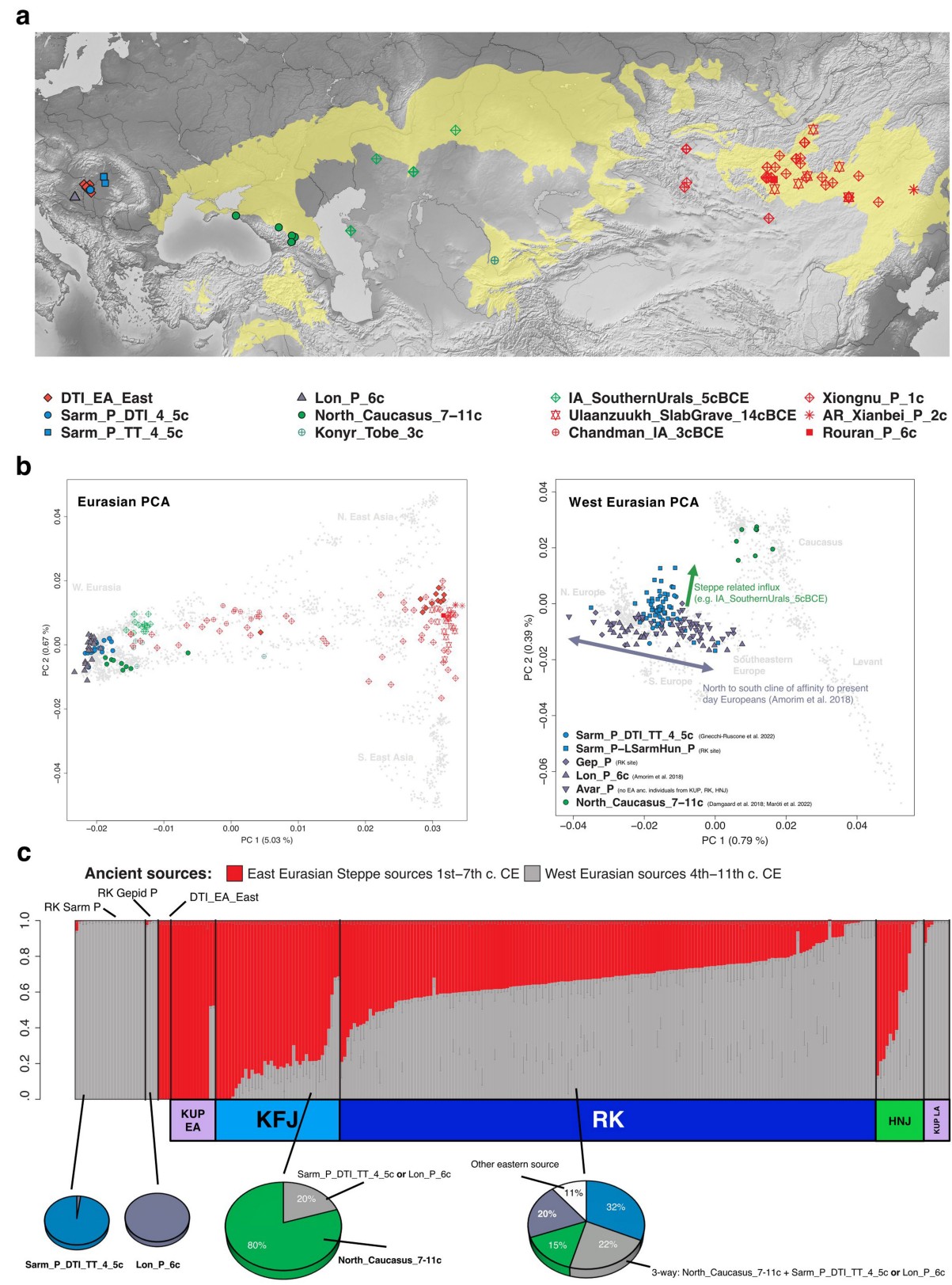

**Extended Data Fig. 8 | Evaluation of the sources of admixture with Western or Central Eurasian ancestries. a)** Map showing the location of the published data used as reference for our ancestry modeling. **b)** "Eurasian PCA" (left) and "West Eurasian PCA" (right), showing the ancient genomes from the sites in a) and unpublished pre-Avar individuals from the RK site. **c)** Summary of the best working *qpAdm* models for the newly sequenced individuals from the 4 sites.

Gray color represents unspecified West Eurasian sources and the pie charts at the bottom show the sites' average for the specific West Eurasian sources tested (full data in Supplementary Table 5). Gray slices in the pie charts represent models with unresolved West Eurasian sources. The base geographic map is from https://www.naturalearthdata.com.

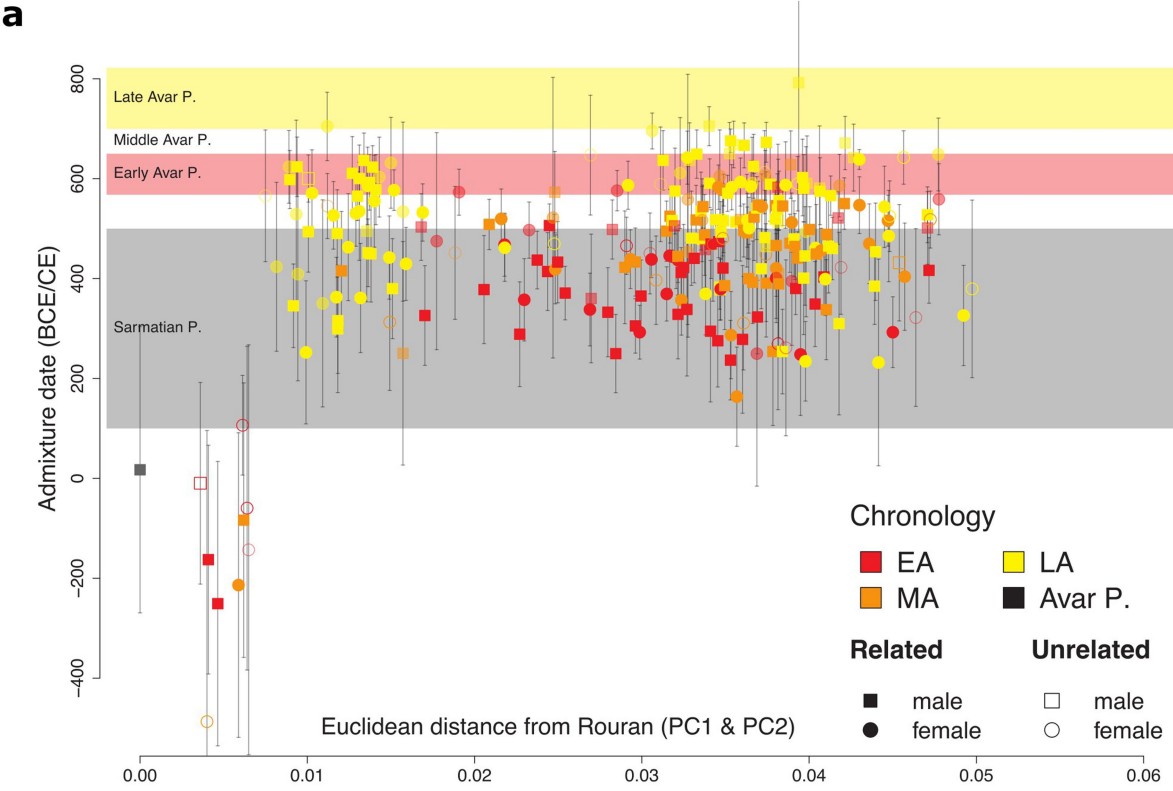

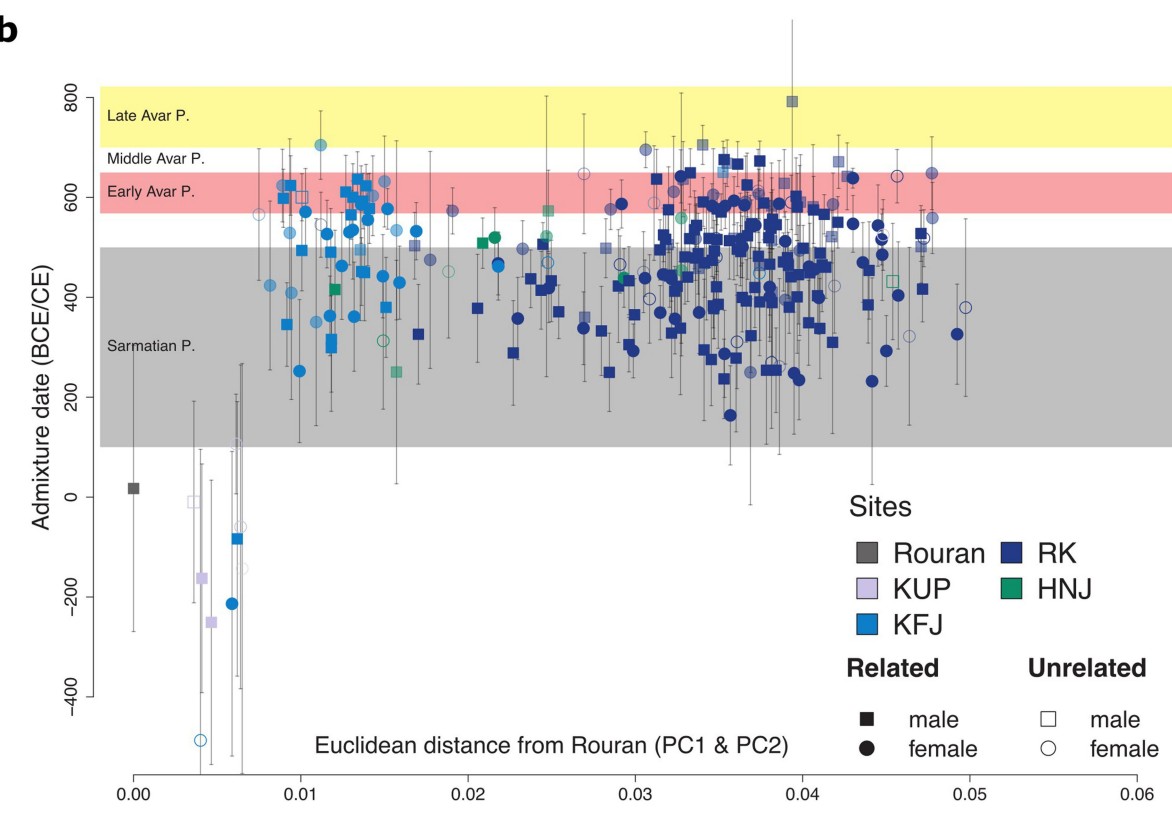

**Extended Data Fig. 9 | Results of DATES admixture dates for all the newly sequenced individuals. a**) Individuals are colored according to their chronological category and **b**) colored according to their site. On the x axis are reported the Eurasian PCA Euclidean distances of each individual to the Rouran genome, used as a proxy for a non-admixed East Eurasian Steppe ancestry[10]. A transparency factor is added to the admixture dates with Z-score <2. Standard errors (SE) and Z-scores are obtained using a standard jackknife approach of 23 independent runs, dropping a chromosome at the time (Methods).

# Reporting Summary

## Statistics

For all statistical analyses, confirm that the following items are present in the figure legend, table legend, main text, or Methods section.

| n/a | Confirmed | |
|---|---|---|
| ☐ | ☒ | The exact sample size ($n$) for each experimental group/condition, given as a discrete number and unit of measurement |
| ☐ | ☒ | A statement on whether measurements were taken from distinct samples or whether the same sample was measured repeatedly |
| ☐ | ☒ | The statistical test(s) used AND whether they are one- or two-sided<br>*Only common tests should be described solely by name; describe more complex techniques in the Methods section.* |
| ☒ | ☐ | A description of all covariates tested |
| ☒ | ☐ | A description of any assumptions or corrections, such as tests of normality and adjustment for multiple comparisons |
| ☐ | ☒ | A full description of the statistical parameters including central tendency (e.g. means) or other basic estimates (e.g. regression coefficient) AND variation (e.g. standard deviation) or associated estimates of uncertainty (e.g. confidence intervals) |
| ☐ | ☒ | For null hypothesis testing, the test statistic (e.g. $F$, $t$, $r$) with confidence intervals, effect sizes, degrees of freedom and $P$ value noted<br>*Give P values as exact values whenever suitable.* |
| ☐ | ☒ | For Bayesian analysis, information on the choice of priors and Markov chain Monte Carlo settings |
| ☒ | ☐ | For hierarchical and complex designs, identification of the appropriate level for tests and full reporting of outcomes |
| ☒ | ☐ | Estimates of effect sizes (e.g. Cohen's $d$, Pearson's $r$), indicating how they were calculated |

*Our web collection on statistics for biologists contains articles on many of the points above.*

## Software and code

Policy information about availability of computer code

| Data collection | No specific software was used for sample collection. Genotype data were generated from sequencing reads. All software used in this study is listed below. |
|---|---|
| Data analysis | The following freely available software were used for data analyses, the corresponding citations are provided in the Methods section: nf-core/eager v2.3.2 (https://nf-co.re/eager); AdapterRemoval (v2.3.1), BWA (v0.7.17), MarkDuplicates v2.22.9 (https://github.com/broadinstitute/picard) , mapDamage (v2.0), BamUtil (v1.0.13), ANGSD (v0.910), Schmutzi (v0.7.12), pileupCaller (https://github.com/stschiff/sequenceTools), samtools (v1.9), HaploGrep2 (v2.4.0), smartpca (v16000; EIGENSOFT v6.0.1), qpWave/qpAdm (v1520), yHaplo (v11.249) & Y-LineageTracker using the ISOGG SNP index (v.11.349 and v15.73), ATLAS (v0.9), R(v 4.05), DATES (v 753), AncIBD v0.5 (https://pypi.org/project/ancIBD/), hapROH v0.64 (https://pypi.org/project/hapROH/), KIN v3.1.3 (https://github.com/DivyaratanPopli/Kinship_Inference), BREADR (https://github.com/jonotuke/BREADR), Cytoscape (v3.9.1), Oxcal (v.4.4.4). |

For manuscripts utilizing custom algorithms or software that are central to the research but not yet described in published literature, software must be made available to editors and reviewers. We strongly encourage code deposition in a community repository (e.g. GitHub). See the Nature Portfolio guidelines for submitting code & software for further information.

## Data

Policy information about availability of data

All manuscripts must include a data availability statement. This statement should provide the following information, where applicable:
- Accession codes, unique identifiers, or web links for publicly available datasets
- A description of any restrictions on data availability
- For clinical datasets or third party data, please ensure that the statement adheres to our policy

> The newly produced sequence data is deposited in the European Nucleotide Archive (ENA) with the following accession number: PRJEB72021. The new haploid genotype data is available through the Poseidon framework (https://github.com/poseidon-framework/community-archive) under https://github.com/poseidon-framework/community-archive/tree/master/2024_GnecchiRuscone_HungaryAvarPedigrees.

# Field-specific reporting

Please select the one below that is the best fit for your research. If you are not sure, read the appropriate sections before making your selection.

☒ Life sciences ☐ Behavioural & social sciences ☐ Ecological, evolutionary & environmental sciences

For a reference copy of the document with all sections, see nature.com/documents/nr-reporting-summary-flat.pdf

# Life sciences study design

All studies must disclose on these points even when the disclosure is negative.

| | |
|---|---|
| Sample size | We did not rely on statistical methods to predetermine sample sizes. Sample sizes for ancient populations depended solely on the availability of archaeological material and on ancient DNA preservation. We applied a "whole cemetery sampling" approach, meaning that we exhaustively sampled all the individuals excavated from the four cemeteries under investigation |
| Data exclusions | Data from specimens that showed insufficient levels of ancient DNA content or high levels of DNA contamination were excluded from further analyses. |
| Replication | We studied unique entities (past and present populations) and did not perform experiments or study various treatments, so replication is not applicable. But we note that samples from the same population carry similar genetic signatures. Moreover, genome-wide data allows for the analysis of multiple realisations of the sample history, by studying hundreds of thousands of SNP sites. |
| Randomization | We studied unique entities (past and present populations) and did not perform experiments or study various treatments, so randomization is not applicable. |
| Blinding | We studied unique entities (past and present populations) and did not perform experiments or study various treatments, so blinding is not applicable. |

# Reporting for specific materials, systems and methods

We require information from authors about some types of materials, experimental systems and methods used in many studies. Here, indicate whether each material, system or method listed is relevant to your study. If you are not sure if a list item applies to your research, read the appropriate section before selecting a response.

| Materials & experimental systems | | | Methods | |
|---|---|---|---|---|
| n/a | Involved in the study | | n/a | Involved in the study |
| ☒ | ☐ Antibodies | | ☒ | ☐ ChIP-seq |
| ☒ | ☐ Eukaryotic cell lines | | ☒ | ☐ Flow cytometry |
| ☐ | ☒ Palaeontology and archaeology | | ☒ | ☐ MRI-based neuroimaging |
| ☒ | ☐ Animals and other organisms | | | |
| ☒ | ☐ Human research participants | | | |
| ☒ | ☐ Clinical data | | | |
| ☒ | ☐ Dual use research of concern | | | |

## Palaeontology and Archaeology

| | |
|---|---|
| Specimen provenance | The Rákóczifalva (RK) cemetery was excavated in 2005-2006 by the Institute of Archaeological Sciences of the Eötvös Loránd University (Katalin Sebők, Gábor Szabó, Katalin Kovács, and Gábor Váczi). The Hajdúnánás-Fürj-halom-járás site (M3 motorway Site 41A) was a rescue excavation in 2005 led by Gábor Szabó from the Institute of Archaeological Sciences of the Eötvös Loránd University. The cemetery of Kunszállás-Fülöpjakab was excavated by Elvira H. Tóth between 1967 and 1979. The site of Kunpeszér, |

Felsőpeszéri út (Bács-Kiskun county, Hungary) was retrieve during a rescue excavation, between 1982 and 1984 conducted by Elvira H. Tóth from the Katona József Museum (Kecskemét). Both KFJ and KUP skeletal specimens are conserved at the Department of Biological Anthropology, University of Szeged (Hungary).

Specimen deposition

Specimens were returned to the owning institutions after laboratory analyses. Left over bone powder samples from the specimens are conserved at the Institute of Archaeogenomics, Research Centre for the Humanities, Eötvös Loránd Research Network, Budapest (Hungary).

Dating methods

New AMS 14C dates were obtained from ultra-filtrated collagen. Collagen extraction and 14C measurements were carried out at the Curt-Engelhorn-Center Archaeometry gGmbH (CECA), Manheim, Germany.

☒ Tick this box to confirm that the raw and calibrated dates are available in the paper or in Supplementary Information.

Ethics oversight

No ethical oversight was required strictly. However, we confirm that all analyses followed established ethical guidelines for archaeogenetic research, as detailed in Wagner et al., AJHG, 2020 and Alpaslan-Roodenberg, Nature, 2021. Ester Banfy, the Ethics Advisor of the ERC funded project HistoGenes (grant agreement number 856453 ERC-2019-SyG), approved the study

Note that full information on the approval of the study protocol must also be provided in the manuscript.

