## [Peer Review File · Nature]

Manuscript Title: Network of large pedigrees reveals social practices of Avar communities

Reviewer Comments & Author Rebuttals

Reviewer Reports on the Initial Version:

Referees' comments:

Referee #1 (Remarks to the Author):

The article presents the results of the genomic analysis of whole Avar period cemeteries. They provide data that allow for the reconstruction of extensive multigenerational pedigrees, which unravels the intricate social dynamics of the Avar Khaganate. The data were taken from 424 individuals from four cemeteries: Kunpeszér (KUP) and Kunszállás (KFJ) from the core region of the Danube-Tisza Interfluve (DTI), and Hajdúnánás (HNJ) and Rákóczifalva (RK) from the neighboring Transtisza (TT) region. The article concludes that the data reveal a consistent reproductive strategy, in which patrilineal descent, patrilocality, female exogamy, and avoidance of consanguinity play a major role. Some cases suggest the existence of multiple reproductive partners and so-called "levirate unions".

The article presents the results of recent research projects within the framework of the mentioned funding institutions. This kind of research is really new and groundbreaking. It provides fascinating results uncovering important aspects of the social organization under the Avar realm. While previous research was limited to archaeological data or written sources, the results from this study as well as related projects and other publications present a novel approach by utilizing a larger amount of ancient DNA from entire cemeteries. This is what this study could generate and analyze. It provides a fresh perspective on the Avar society at a micro level and contributes to our understanding of its family structures.

The researchers utilize high-quality data sets to construct the pedigree structures. The presentation of the data contains clear visualizations and explanations that enhance the reader's understanding of the topic. The authors address uncertainties inherent in their research by acknowledging limitations in the interpretative potential of their data. They provide transparency in their methodology, allowing readers to evaluate the robustness of the analysis and the resulting conclusions.

The conclusions drawn in the article demonstrate robustness, validity, and reliability. The researchers carefully interpret the data, ensuring that their conclusions align with the observed patterns. While the methodology requires a profound knowledge of how DNA analyses work, the results and their interpretation can easily be followed. The findings contribute to the broader field of history, archeology, and social anthropology. The authors also suggest avenues for further research. The article is comprehensive and well-written. There is, however, always room for further exploration. The audience of this article is mainly interested in the DNA-analysis and their interpretation. For historians and archaeologists, a closer connection with the archaeological data from the mentioned cemeteries would be useful. This, however, is certainly done in other publications and does not have to be necessarily included in this study.

The article appropriately credits previous works, acknowledging the contributions of earlier

research. The authors provide a solid foundation by building upon the existing body of knowledge. The article provides sufficient clarity and context. The abstract and summary concisely capture the key findings and research objectives, making the article easily accessible to a wide range of readers. The introduction provides the necessary background information, establishing the context for the study. The conclusions effectively summarize the main insights while highlighting their significance. The passage on the methodology provides useful details for those readers whose main interest is how the results were generated.

The researchers' meticulous methodology, appropriate interpretation of their data, and convincing conclusions contribute to the validity and reliability of their findings. While acknowledging the limitations of the data, the article contributes significantly to our understanding of the Avars' social organization. The results of this project provide new opportunities. We are just at the beginning. This is a very promising path and in a few years, it will be an integral part of any future research. DNA analysis is a new and extremely valuable resource for historians, archaeologists, and social anthropologists. It will broaden our scope and change the way we will conduct our research. This study, which provides data of larger quantities taken from entire cemeteries, is an important step toward this new path. It should be published and will be a valuable reading for everyone who does research on the Avars and similar topics or for anyone who is interested in the potential of DNA analysis for historical research.

Daniel Ziemann

Referee #2 (Remarks to the Author):

Gnecchi-Ruscione and colleagues have produced a wonderful body of work on social organisation among the Pannonian Avars. The main novelties lie in (1) the size of the pedigree reconstructed, (2) the demonstration that the society practiced levirate unions, (3) capturing a likely realignment of political power and a replacement of the main kin group at the RK cemetery. This final discovery is in my view the most significant for the field and a real testament to the exhaustive genomic and isotopic sampling strategy carried out by the authors. The discovery of patrilineality, patrilocality and female exogamy is also of interest, as is the lack of consanguinity, but less impactful as these phenomena have been observed in the ancient genomic record previously and are not surprising for this type of society.

I have no criticism of the methods used – the authors are innovative and should be commended for getting the absolute most out of their data. The IBD analysis is particularly impressive. They've set an excellent example for ancient DNA surveys of cemetery sites.

I believe these results have the potential to attract broad interest from the readers of Nature, but not in the manner they are currently presented. As it stands, the manuscript reads like a promising first draft. However, it is long-winded and overly detailed in many places, making it hard for a coherent narrative to emerge. I think a non-specialist would struggle to grasp the significance of the results presented. The authors tend to point out interesting observations (e.g. sex bias among unrelated individuals, high status individual is hub in IBD network) but provide no further interpretation or context. Lines of reasoning are not always clear (e.g. lack of admixture in Carpathian Basin) and results are not presented in order of importance.

For example, it took me until line 357 to find out about the most captivating result – the replacement of the patriline at RK. This is a really exciting application of ancient DNA that, to my knowledge, has never been demonstrated before – capturing changes in the political landscape and providing quite precise dates on these with pedigree data. This is exciting stuff and the authors should be whacking the reader over the head with this upfront. Instead, their short abstract is extremely vague, only mentioning “insights into kinship”. The abstract is not much better and doesn't really mention this finding.

The reader also has to trawl through the supplementals to find important contextual information on steppe social organisation, while the main text provides unnecessary detail on methods and sampling strategies. New and relevant findings are also relegated to the supplementary (e.g. marriage age estimates), while older findings from previous studies are revisited at length.

My recommendation is that the authors carry out a substantial restructuring of the manuscript to make it more concise and accessible to the Nature readership. The key results should be made clear up front and their broader significance highlighted. Extraneous information and excessive detail should be avoided, as should long reaffirmations of the results from Gnecchi-Ruscione et al. 2022. The focus should be on the novel. The authors' should reconsider what to display in the main figures – the first two don't contain any significant new findings. Legibility of the figures could also be improved – they are quite busy and some text is very small, preventing the most important trends

from jumping out. Grammar checks also needed throughout – there are many incoherent sentences.

I provide some more detailed comments section-by-section below. I provide multiple suggestions on how the text may be restructured, but I stress these are suggestions. The authors may have better ideas on how to streamline the manuscript. I have no real issues with interpretation, except for the section on admixture, which I also outline below.

-> Introduction and Conclusions

I think both these sections need to readjust their focus. It is well understood that kinship does not always map to biological relatedness, but this is not a key finding of this study so it is a bit odd that both the introduction and conclusion open on this point. Ancient DNA has been used many times to study kinship, but how many times has it been used to study local realignments of power? It would be more useful to discuss how social and political structures can be built upon kinship and descent systems and theories surrounding how Avar societies were organised (e.g. more general discussion of steppe societies).

Also, the list of questions given in the introduction fail to anticipate the major findings of this work. The authors' never really revisit which steppe traditions were maintained, at least not in the main text (are they implying patriliney, levirate unions and avoidance of consanguineous marriage were customs imported from the steppe?) The interactions between Avar and local Carpathian populations are not properly resolved and aren't a key selling point of this study.

-> Interdisciplinary analysis of entire cemeteries

This section reads more like methods (with some introductory archaeological context), rather than results and discussion. It is overly-detailed and could be shortened considerably. There is also repetition with the final paragraph of the introduction, which already tells us that isotope and genomic data was retrieved from four fully excavated cemeteries.

I think the archaeological context for each site would be better introduced when site specific results are being discussed (e.g. in the pedigree section), rather than overloading the reader at the beginning. For example, the relevance of RK being the largest burial ground and occupied continuously across the Avar period is not clear until the nine generation pedigree is presented.

This section is also very heavy on numbers. For example, it tells us exactly how many individuals from each site were sampled for isotopes and RC dating, as well as the sampling strategy. I'm not sure this information is really necessary in the main text, especially as we have to wait another 65 lines before detailed isotope results are given. It is an impressive amount of sampling, but this will be lost on the non-specialist reader.

Also, for the general reader, being introduced to nine different acronyms in the first two paragraphs is a bit excessive. I understand acronyms sometimes can't be avoided, but when both geographical regions and archaeological sites have acronyms, text becomes harder to follow.

-> Genomic ancestry, cultural connections and mobility of Avar period communities with steppe

descent

I think this section could also be shortened considerably and the novel results provided in a more concise and clear fashion. Main criticisms:

1. Novelty: My main issue with this section is that it reads as a follow up on a previous publication with some new data added (Gnecchi-Ruscione et al. 2022, Cell). This is not the level of novelty a reader would expect for Nature, so I'm unclear why these results are being placed front and centre. For example, the final line of the first paragraph provides the same conclusion as the 2022 paper –

Current manuscript: "only more aDNA data from these undersampled steppe areas would allow to refine the sources of admixture"

2022 paper: "Instead, it rather matches the steppes north of the Caucasus, although the scarceness of comparative data from the steppe in the 1st millennium CE calls for a future investigation of possible better alternative sources."

2. Context: A related issue is that it is very hard for the reader to identify what the new findings in this section actually are. You really would have to be an expert on Avar ancient genomics to understand how these new data have significantly changed our understanding of this population's migratory or admixture history. There are a lot of confirmations of Gnecchi-Ruscione et al. 2022, but I believe there are also rejections/refinements of some interpretations made in that paper? This is not made clear to the reader. For example, the final line of the section says the evidence presented "poses a challenge to the hypothesis that there were successive large-scale migrations from the steppe". However, no citation is given for this hypothesis. Is this a rejection of the interpretation from Gnecchi-Ruscione et al. 2022 that:

"This non-local admixture together with the retention of a high level of eastern ancestry in the late elites and the absence of genetic inbreeding may point to continued migration from the steppes after the initial arrival of the Avars in the Carpathian Basin".

I think the authors' conclusion that admixture dates support most admixture occurring prior to Avar arrival in the Carpathian also departs from that of the 2022 paper?

3. Clarity and Conciseness: Again related to the above, there is no clear summary given for the current model for the formation of Avar populations in the Carpathian Basin. The title of this section is very long and vague and does not provide any overview of the main results within. The abstract also provides no one-line summary of this section. There is also no concise conclusion at the end of the section. In fact, the "in conclusion" paragraph is longer than the rest of the section and introduces new results, including strontium isotopes. This poor structuring immediately makes the paper feel inaccessible to the non-specialist reader.

4. Interpretation: The authors seem to conclude that only a minimal amount of admixture took place within the Carpathian basin during the Avar period, but no exact estimate is given. The argument for

this is not made very coherently. I'm not unconvinced, but I find the logic hard to follow.

Firstly, it is not explained how Sarmatian ancestry differs from "main European" ancestry in the pre-Avar Carpathian basin or if Sarmatian-related groups persisted into the Avar period. For example, in Fig. 2B it is not obvious what population is represented by the grey "Pre-Avar Carpathian Basin" and the piecharts in Extended Data Fig 1C don't help matters as their relationship to the 3-way qpAdm is unclear. I also don't follow lines 164-169. Do 20% of individuals in RK have some "main European" ancestry? If so, which source populations and how much? It's also not clear how many mothers you would be expecting to be unadmixed to give rise to this 20% estimate, especially as the authors point out later they may not have sampled the earliest generation of migrants.

Fig 2B shows most individuals are majority grey ancestry, so it seems the conclusion that minimal admixture took place in the Carpathian Basin rests solely on admixture dates and a hypothetical Sarmatian-related population further east (rather than "multiple lines of evidence")? The authors also say that there is "regional genetic homogeneity throughout time" (Line 190), but Figure 2 shows clear temporal shifts in ancestry towards west Eurasian populations for every site where there is multi-period sampling. Admixture dates for many LA samples also fall within the EA period (Extended Data Figure 2). The fact that LA samples have later dates than EA samples also suggests continuous admixture.

The arguments made from isotopic data also appear weak. The authors also concede that distant areas may be isotopically indistinguishable, but then go on to argue for mobility over short distances. They presume they have not managed to sample the first generation of migrants, even though sites were sampled exhaustively. Is it a possibility that the first generation migrants have indeed been sampled, but come from a distant but isotopically indistinguishable area?

I'm also not convinced that various steppe descent groups remain genetically distinguishable in different parts of the Carpathian Basin, given the heterogeneity seen in each site (lines 169-172). I don't think I could confidently differentiate between a KFJ and HNJ sample on the basis on ancestry.

-> Kinship sections

1. Pedigree and Kinship patterns
2. Polygyny, levirate reproductive unions and consanguinity
3. Exogamous females and connections between communities
4. Shifts in local communities from the Early to the Middle/Late Avar periods

These four sections contain the most interesting results, but their structure could be much improved. Related sets of results are scattered throughout multiple sections (which I'll label 1-4 for convenience). For example, we're told in line 255 that only one Y lineage (N1a) is found in both KFJ and KUP, but have to wait until line 344 to find out these males cluster together in IBD analysis.

RK is the main case study. There is a huge amount going on at this site and it would be easier to digest if most of the site-specific results from RK were presented together in a single section, after the more general trends across sites were highlighted. As it stands, the reader is drip-fed key findings: In section 1 we find out two Y lineages dominate RK, associated with certain pedigrees; in

section 2 we find many females at RK have multiple reproductive partners; in section 3 find these females have a key role connecting these different patriline and in section 4 we're told one patriline appears to replace other, the connection being between two maternal half-brothers. This is a very long-winded way of telling a truly fascinating story.

Currently, sections 1 and 4 are focussed mainly on RK, while sections 2 and 3 are more evenly split between RK and more general trends seen across sites. My suggestion is that the authors try put the more general findings first, followed by a deep dive into RK and the incredible details they have gleaned from that site. For example:

1. Connections between communities: patrilocality and female exogamy
2. Pedigree Reconstruction: Strict Patriline and Levirate customs within sites
3. Rákóczifalva: capturing shifts in local power structures

The initial case for patrilocality and female exogamy can be made clearly with patterns of IBD sharing among adult males and females, alongside Y chromosome and mtDNA diversity. The IBD sharing data also gives the reader a nice first overview of the pedigrees/kin groupings present at different sites. Pedigree data from across sites can then be used to further demonstrate patriline, female exogamy as well as polygyny/levirate unions/inbreeding. The stage is then set for the RK case study, including integration of spatial and grave good data, refinements of site chronology, isotopes, all of which is needed to fully contextualise the replacement of one patriline with the other.

Small comments:

1. The Y chromosome haplogroups given in the main text are very basal (e.g. J1a and J2b). It could be made clearer that the haplotype resolution is much higher for most individuals and still is consistent with a small number of dominant patriline (e.g. J1a2a1a2d2b2b2 and J2b2a1a1a1a1a1e).

Some grammatical issues:

Lines 110-111 two "ands" in a row without comma, makes it hard to know if the core region is "Danube-Tisza Interfluvium and Hajdúnánás" or the two cemeteries from the neighbouring region are "Hajdúnánás and Rákóczifalva".

Lines 129-130: This is hard to understand. Unclear if there are cultural/social differences between different regions on the same side of the Tisza (and we see this on both sides of Tisza).

Lines 152-14. Again too many "ands" makes the sentence harder to follow. Could be made more concise.

Lines 157-159. This sentence is hard to understand. There shouldn't be a comma before "is very limited".

Lines 157-159. To simplify and keep consistent with terms used higher up in the paragraph, I would change "working with groups representing the main European genomic profile found in the 5th-6th

CE and during the course of the Avar period in the Carpathian basin” to “working with western Eurasian ancestries associated with the Carpathian basin in the Avar and preceding periods”.

Lines 162-163: “would allow to refine the sources of admixture” -> “will allow the sources of admixture to be better refined”

Lines 186-188. “On the one side” is not an easily understood phrase. It implies these two conclusions are meant to be contradictory, which they aren’t. The first generation of migrants may not have been sampled AND there may have been high regional continuity across the period.

Line 206-207: “was closely related” -> “were closely related”

Line 255-256: “is found between” -> “is found in both”

Line 270-272: Hard to understand. In the vast majority of cases?

340-343: This sentence doesn’t make sense and is incorrectly capitalised.

Line 359-361: Hard to follow – needs to be better phrased. IBDs -> IBD segments.

Line 378: “the burial with horse harness” -> “such as burial with a horse harness”

Line 391-393: Hard to follow. “while overall ancestry” -> “while the overall ancestry”

Figure issues

Fig 2b. Text is different sizes and too small for final group. Why are RK individuals separated by relatedness? If there is a difference here it is not highlighted in the main text.

Fig 3. Timeline needs date intervals marked. Why are the Y lineages for unsampled males not shown, when they can be inferred from their sampled sons? In pedigree 4 it looks like two brothers are also 3rd degree relatives?

Referee #3 (Remarks to the Author):

Gnecchi-Ruscone, Rácz et al. present results of an analysis involving the reconstruction of multigenerational pedigrees using genomes from Avar-period cemeteries. This analysis extends recent work by Gnecchi-Ruscone et al. Cell 2022 to investigate cultural interactions following the formation of the Avar groups.

The paper is interesting and well written, although several parts of it are outside of my expertise. This review focuses on statistical and genealogical aspects of this work, rather than on the archeological and anthropological findings, which I hope others will be better positioned to comment on.

Major comments:

- Genetic maps: My understanding is that the authors are using the same genetic map to determine the length of IBD segments for all pairs of individuals (although I could not find details on this). This genetic map is likely computed as an average between male and female recombination rates, which vary substantially (see e.g., Kong et al. 2010 Nature, Caballero et al 2019 Plos Genetics). This affects the calculation of IBD segment lengths in cM, used in several steps of the analysis. For instance, $\langle k \rangle$ and $\langle w \rangle$ observed in females are lower but this may reflect the fact that females transmit segments that appear to be shorter if a sex-averaged map is used. Two questions arise: (1) are the analyses based on IBD segments reported here, for instance on male vs female IBD graph connectivity, robust to the use of a sex-averaged map? This could be assessed using simulations where sex-specific maps are used to generate the data, but sex-averaged maps are assumed during analysis. It would also be useful for this manuscript to have a supplementary figure showing the distribution of number, length, and sum of IBD segments as a function of relationships and number of males/females in the genealogical path between two individuals. (2) Can the authors take this into account to refine their estimated IBD segment lengths and downstream analyses? (This would be nice if achievable, but a discussion would be sufficient.)

- Phasing, imputation, and IBD detection: L726: "Obtaining such data with ancient genomes has been proven reliable..." This seems a bit too strong. Surely switch error and genotyping error rates remain quite high, so "reliable" entirely depends on specific applications. Relevant to this work, accurate IBD detection requires high quality phasing and genotyping and the ancIBD approach that underpins several analyses in this work is still under review. I would therefore recommend that the authors perform independent validation of the precision and recall of ancIBD using simulated pedigree data to rule out potential biases (also see my other comment on more realistic simulations, which would also apply to these simulations).

- Pedigree simulations: the simulations of Supp Fig 16 and 17 are a bit too simplistic. Most importantly, they use 22 chromosomes of 100 cM each. This may be fine if the goal is to observe general trends, but my understanding is that these simulations are used to establish a null model, which is then used to reject relationships in real data, which has a different total cM length. Unless I misunderstood, this analysis needs to be revised. More broadly, I would recommend making the simulations of L702 more realistic by (1) using the correct chromosome sizes; (2) using a genetic map

(possibly sex-specific); (3) matching the genotype density of real data; (4) modeling low coverage data, genotyping errors, phasing errors. (It may be also relevant to consider varying the demographic model, as homozygosity in the pedigree founders due to a small effective population size may affect IBD detection accuracy.)

- Admixture analyses: Please clarify statements such as “resulted in a significant rejection” (L623) and “did not significantly reject” (L625). What constitutes a significant rejection? L620 reports 0.05 as a significance threshold. Were significance levels properly adjusted for multiple testing? This also applies to other parts of the manuscript, e.g., (I believe, but see my comment on Supp Table labeling) Supp Table 4.

Additional comments:

- I don't understand the choice of only filtering out pairs of individuals sharing a single segment within the range of 12 to 16 cM. Why not apply this to >16 cM? Close relationships should also result in multiple shared segments. The authors should provide more support for this choice, which appears arbitrary. Also, ancIBD recommends only considering segments >8cM. The authors first state they considered segments >5 cM (L754) and then >12 cM (L756). Is the >5 cM filter not superseded by the >12 cM filter? Please clarify.

- “Maximum IBD value” in L771 is unclear. But based on the values reported in the caption of Figure 4 I suspect it is the largest segment found in the pair. I do not understand the choice of this statistic as it is more noisy than other natural choices such as the sum of the lengths of all segments found in the pair (or equivalently the fraction of genome shared). The authors should provide support for what appears to be an odd choice or consider using the sum instead.

- Admixture analyses: I am generally a bit uneasy about analyses that compare several parametric models where the space of possible combinations is extremely large, which are affected by the choice of models being compared and the availability of appropriate reference populations. The authors do a good job at describing the rationale of their choices, but I would recommend they add more methodological context by providing references to other papers that used model selection strategies like the one adopted here.

- L164: How was the 20% admixture with European populations estimated? Please clarify the methodology and make admixing populations more explicit.

- “The majority of these are older men (35-59)”, clarify “majority”.

- “Pedigree building strategy”: additional support (e.g., simulations) for the approach used to resolve inconsistencies in the case of levirate unions would be good.

- Supp Fig 16 and 17. It is hard to tell circles and squares apart.

- L144-145 should this be fig 2b? Also, it took me a few readings to understand what highest/lowest median refers to.

- I found it hard to locate supplementary tables as tables in the excel files do not always have a caption.

- EDF7 caption "Aver" → "Avar"?

- "Population genetic theory says" → "suggests"?

- L192: The numbers in parentheses are not defined in the caption. Is it the Explained Variance Ratio?

- L768 : (typo) if only the adults (are?) selected

Author Rebuttals to Initial Comments:

We thank the reviewers for the constructive feedback. We have addressed all the reviewers' comments, a detailed point-by-point response to these comments can be found below (marked "R", original comments are marked with "C"). All the changes, new additions and re-formatting adjustments are marked in red in the manuscript for the convenience of the peer review process.

Referee #1 (Remarks to the Author):

C: The article presents the results of the genomic analysis of whole Avar period cemeteries. They provide data that allow for the reconstruction of extensive multigenerational pedigrees, which unravels the intricate social dynamics of the Avar Khaganate. The data were taken from 424 individuals from four cemeteries: Kunpeszér (KUP) and Kunszállás (KFJ) from the core region of the Danube-Tisza Interfluve (DTI), and Hajdúnánás (HNJ) and Rákóczi falva (RK) from the neighboring Transisza (TT) region. The article concludes that the data reveal a consistent reproductive strategy, in which patrilineal descent, patrilocality, female exogamy, and avoidance of consanguinity play a major role. Some cases suggest the existence of multiple reproductive partners and so-called "levirate unions".

The article presents the results of recent research projects within the framework of the mentioned funding institutions. This kind of research is really new and groundbreaking. It provides fascinating results uncovering important aspects of the social organization under the Avar realm. While previous research was limited to archaeological data or written sources, the results from this study as well as related projects and other publications present a novel approach by utilizing a larger amount of ancient DNA from entire cemeteries. This is what this study could generate and analyze. It provides a fresh perspective on the Avar society at a micro level and contributes to our understanding of its family structures. The researchers utilize high-quality data sets to construct the pedigree structures. The presentation of the data contains clear visualizations and explanations that enhance the reader's understanding of the topic. The authors address uncertainties inherent in their research by acknowledging limitations in the interpretative potential of their data. They provide transparency in their methodology, allowing readers to evaluate the robustness of the analysis and the resulting conclusions.

The conclusions drawn in the article demonstrate robustness, validity, and reliability. The researchers carefully interpret the data, ensuring that their conclusions align with the observed patterns. While the methodology requires a profound knowledge of how DNA analyses work,

the results and their interpretation can easily be followed. The findings contribute to the broader field of history, archeology, and social anthropology. The authors also suggest avenues for further research.

R: We thank the reviewer for positive evaluation of our work. We are happy that the reviewer appreciates our analysis and interpretation, especially from the perspective of humanities. We have strived for integration of the disciplines to achieve such understanding.

C: The article is comprehensive and well-written. There is, however, always room for further exploration. The audience of this article is mainly interested in the DNA-analysis and their interpretation. For historians and archaeologists, a closer connection with the archaeological data from the mentioned cemeteries would be useful. This, however, is certainly done in other publications and does not have to be necessarily included in this study.

R: Indeed, we are planning further publications on the Rákóczi falva cemetery: a monograph including a detailed archaeological and anthropological analysis of the site, and an article on archaeological chronology and radiocarbon dating for a specialist audience. In the current article, we added more archaeological information from the supplements to lines 92, 97 and 101-103. Furthermore, we created a new Extended Data Figure 7 that shows changes in the burial customs from the early to the late phase in the Rákóczi falva community, and can thus be well correlated with the population change we have genetically detected. Some of the most important features that depict the archaeological characteristics of the early and middle-late Avar populations/cemetery sections, as well as the differences between them, have been added to the main Figure 2. In addition, Figure 1 has been supplemented with some excavation and artifact photographs of the sites investigated.

C: The article appropriately credits previous works, acknowledging the contributions of earlier research. The authors provide a solid foundation by building upon the existing body of knowledge. The article provides sufficient clarity and context. The abstract and summary concisely capture the key findings and research objectives, making the article easily accessible to a wide range of readers. The introduction provides the necessary background information, establishing the context for the study. The conclusions effectively summarize the main insights while highlighting their significance. The passage on the methodology provides useful details for those readers whose main interest is how the results were generated.

R: We thank the reviewer for evaluation of each section of the manuscript and for putting our study in the wider context. We very much agree with the reviewer that more studies of this kind are and should be coming and we hope that our study will contribute to a direction where historical research would be enriched.

C: The researchers' meticulous methodology, appropriate interpretation of their data, and convincing conclusions contribute to the validity and reliability of their findings. While

acknowledging the limitations of the data, the article contributes significantly to our understanding of the Avars' social organization. The results of this project provide new opportunities. We are just at the beginning. This is a very promising path and in a few years, it will be an integral part of any future research. DNA analysis is a new and extremely valuable resource for historians, archaeologists, and social anthropologists. It will broaden our scope and change the way we will conduct our research. This study, which provides data of larger quantities taken from entire cemeteries, is an important step toward this new path. It should be published and will be a valuable reading for everyone who does research on the Avars and similar topics or for anyone who is interested in the potential of DNA analysis for historical research.

Daniel Ziemann

Referee #2 (Remarks to the Author):

C: Gneccchi-Rusccone and colleagues have produced a wonderful body of work on social organisation among the Pannonian Avars. The main novelties lie in (1) the size of the pedigree reconstructed, (2) the demonstration that the society practiced levirate unions, (3) capturing a likely realignment of political power and a replacement of the main kin group at the RK cemetery. This final discovery is in my view the most significant for the field and a real testament to the exhaustive genomic and isotopic sampling strategy carried out by the authors. The discovery of patrilineality, patrilocality and female exogamy is also of interest, as is the lack of consanguinity, but less impactful as these phenomena have been observed in the ancient genomic record previously and are not surprising for this type of society.

R: *We thank the reviewer very much for appreciating the value of our research. We agree that the realignment of political power and kin groups at the RK site is an exceptional finding and probably the most innovative and original as a topic not yet addressed by ancient DNA. We thus readily address the points below about highlighting this topic in the main text. It proves that ancestry continuity might still conceal drastic population replacements and shows that these can still be identified from genetic data with such a dense sampling of whole cemeteries and proper archaeological contextualization. The reviewer is right that investigating kinship through genetic and archaeological data is not a new topic, but we would like to highlight that the extent of the pedigrees that we were able to reconstruct for these communities are unprecedented and provide a level of detail and extent of recurrent patterns that allow us to infer the patterns of patrilineality, patrilocality and female exogamy at a level un-observed so far. Furthermore, the value of observing these patterns across the four sites provides a strong evidence of social practices, stronger than any previous study before. The levirate union is one example of a recurrent pattern of a specific social practice that has never been described for an ancient community. Few, very recent genomic studies have analyzed pedigrees reconstructed from ancient DNA relatedness but to our knowledge none of them have an extent such as the ones we present here nor involved European early medieval communities with steppe/East Asian descent. This is the first time that such complete pedigrees showing a clear*

descent system are identified for a non-European origin ancient population. We know very little about these medieval populations settling in Europe during the migration period, their history being told primarily by their enemies and this is the first time we are able to have a view into the kinship and social structure of such societies.

C: I have no criticism of the methods used – the authors are innovative and should be commended for getting the absolute most out of their data. The IBD analysis is particularly impressive. They've set an excellent example for ancient DNA surveys of cemetery sites.

R: *We thank the reviewer for appreciating the methodological advances.*

C: I believe these results have the potential to attract broad interest from the readers of Nature, but not in the manner they are currently presented. As it stands, the manuscript reads like a promising first draft. However, it is long-winded and overly detailed in many places, making it hard for a coherent narrative to emerge. I think a non-specialist would struggle to grasp the significance of the results presented. The authors tend to point out interesting observations (e.g. sex bias among unrelated individuals, high status individuals are hub in IBD network) but provide no further interpretation or context. Lines of reasoning are not always clear (e.g. lack of admixture in Carpathian Basin) and results are not presented in order of importance.

R: *We thank the reviewer for this constructive feedback. We have conducted an extensive restructuring and rewriting of the main text following the reviewer suggestions (see point by point responses below). For example, we do in fact provide an interpretation for sex bias among unrelated individuals but the sentence where we highlight such bias was not sequential to its interpretation. Thanks to this comment we realized this mistake and we moved this sentence (lines 162-169). Thanks to the reviewers' inputs we hope that now the draft can be considered more mature for publication. It should be noted, to be fair to all the coauthors, that this article is the result of an extensive interdisciplinary collaboration between experts of different fields (genetics, stable isotope analyses, archaeology, social and physical anthropology and history) all of whom were actively involved in the writing process. Putting together all these different writing styles and topics (some that might not seem secondary for one field, but essential for another) has been a very complicated process. For example: reviewer 1, as historian, was able to appreciate the level of historical and archaeological detail that might seem too specific for someone from another field.*

Having said so, we agree that Nature's audience is generalist and we have therefore cut or reduced as much as possible all the field-specific details and tried to be as clear as possible in all our lines of reasoning.

C: For example, it took me until line 357 to find out about the most captivating result – the replacement of the patriline at RK. This is a really exciting application of ancient DNA that, to my knowledge, has never been demonstrated before – capturing changes in the political landscape and providing quite precise dates on these with pedigree data. This is exciting stuff and the authors should be whacking the reader over the head with this upfront. Instead, their

short abstract is extremely vague, only mentioning “insights into kinship”. The abstract is not much better and doesn’t really mention this finding.

R: *We thank the reviewer for pointing this out. We agree that the patriline replacement is one of the most innovative of our results in terms of new ancient DNA applications. We now highlighted this finding more prominently in the article: it is now mentioned in the abstract (lines 52-55) and introduction (lines 87-90) and in the other parts of the text. These results however directly stem from the pedigree analysis that precede this observation. Therefore, we believe that the main chapter where this topic is discussed in detail should still succeed the pedigree chapter. But to move upwards this topic we shortened and moved the chapter describing the genomic ancestry after the community shift and local realignment chapter (see the specific comment below).*

C: The reader also has to trawl through the supplementals to find important contextual information on steppe social organisation, while the main text provides unnecessary detail on methods and sampling strategies. New and relevant findings are also relegated to the supplementary (e.g. marriage age estimates), while older findings from previous studies are revisited at length.

R: *We thank the reviewer for providing further suggestions on things to cut and other things to extend further. We have followed these suggestions and limited the details about the sampling strategy however we believe that the whole cemetery approach applied over several sites is quite unique to this study and was necessary in order to generate the findings, so mentioning it, even in short, is of crucial importance.*

Analogies to living populations on the steppe social organization need careful and extensive discussion so it requires the accompanying text that we had in the SI (pages 25-29 of SI), but we agree that these should be mentioned in the main text so we have integrated some sections about this (lines 206-214) as well as about the marriage age estimate (lines 170-174).

C: My recommendation is that the authors carry out a substantial restructuring of the manuscript to make it more concise and accessible to the Nature readership. The key results should be made clear up front and their broader significance highlighted. Extraneous information and excessive detail should be avoided, as should long reaffirmations of the results from Gneccchi-Ruscone et al. 2022. The focus should be on the novel. The authors’ should reconsider what to display in the main figures – the first two don’t contain any significant new findings. Legibility of the figures could also be improved – they are quite busy and some text is very small, preventing the most important trends from jumping out. Grammar checks are also needed throughout – there are many incoherent sentences.

I provide some more detailed comments section-by-section below. I provide multiple suggestions on how the text may be restructured, but I stress these are suggestions. The authors may have better ideas on how to streamline the manuscript. I have no real issues with interpretation, except for the section on admixture, which I also outline below.

R: *We are very grateful for these constructive feedbacks. We made a substantial restructuring of the manuscript and figures in order to make it more accessible to the broad audience of Nature. We carefully followed the reviewer's suggestions. Details are in the point by point responses to the reviewer's comments below. The text has also been proof-read by native English speakers.*

-> Introduction and Conclusions

C: I think both these sections need to readjust their focus. It is well understood that kinship does not always map to biological relatedness, but this is not a key finding of this study so it is a bit odd that both the introduction and conclusion open on this point. Ancient DNA has been used many times to study kinship, but how many times has it been used to study local realignments of power? It would be more useful to discuss how social and political structures can built upon kinship and descent systems and theories surrounding how Avar societies were organised (e.g. more general discussion of steppe societies).

Also, the list of questions given in the introduction fail to anticipate the major findings of this work. The authors' never really revisit which steppe traditions were maintained, at least not in the main text (are they implying patriliney, levirate unions and avoidance of consanguineous marriage were customs imported from the steppe?) The interactions between Avar and local Carpathian populations are not properly resolved and aren't a key selling point of this study.

R: *We thank the reviewer for pointing these out. We have substantially restructured the introduction. We agree to change the focus of these sections and put more emphasis on the issue of population change, i.e. the local power realignments (lines 249-300). The observation that the kinship system and political power in the steppe societies were deeply connected is now given more weight as well in the main text (lines 113-214). It should be emphasized, however, that comparative material on the same scale as ours is not yet known from the Eurasian steppe. Data on social and political structures and social organization are scattered, either in analyses based solely on archaeological data or in the works of historians, especially on the Turks, the Xiongnu and the Xianbei.*

We agreed and removed our opening questions. We have now more precisely stated the aim of our study to concentrate on the focal element of our research (lines 80-83).

->Interdisciplinary analysis of entire cemeteries

C: This section reads more like methods (with some introductory archaeological context), rather than results and discussion. It is overly-detailed and could be shortened considerably. There is also repetition with the final paragraph of the introduction, which already tells us that isotope and genomic data was retrieved from four fully excavated cemeteries.

R: *We thank the reviewer for suggesting this. We have substantially shortened this section and removed the repetition (lines 91-111).*

C: I think the archaeological context for each site would be better introduced when site specific

results are being discussed (e.g. in the pedigree section), rather than overloading the reader at the beginning. For example, the relevance of RK being the largest burial ground and occupied continuously across the Avar period is not clear until the nine generation pedigree is presented.

R: We thank the reviewer for suggesting this. We have thought about this extensively and came to the conclusion that including the specifics of the archaeological context of each site in the pedigree description section would break the flow of pedigree description. So, we decided to shorten this section as the reviewer suggested in the previous comment, but keep it at the beginning in this way it also matches with Figure 1 that introduces the sites.

C: This section is also very heavy on numbers. For example, it tells us exactly how many individuals from each site were sampled for s and RC dating, as well as the sampling strategy. I'm not sure this information is really necessary in the main text, especially as we have to wait another 65 lines before detailed isotope results are given. It is an impressive amount of sampling, but this will be lost on the non-specialist reader.

R: As stated in previous comment we have substantially reduced this section and now the numbers are limited to the most important ones.

C: Also, for the general reader, being introduced to nine different acronyms in the first two paragraphs is a bit excessive. I understand acronyms sometimes can't be avoided, but when both geographical regions and archaeological sites have acronyms, text becomes harder to follow.

R: We followed the reviewer's suggestions and removed the acronyms that referred to the archaeological chronology.

-> Genomic ancestry, cultural connections and mobility of Avar period communities with steppe descent

C: I think this section could also be shortened considerably and the novel results provided in a more concise and clear fashion. Main criticisms:

1. Novelty: My main issue with this section is that it reads as a follow up on a previous publication with some new data added (Gnecchi-Ruscone et al. 2022, Cell). This is not the level of novelty a reader would expect for Nature, so I'm unclear why these results are being placed front and centre. For example, the final line of the first paragraph provides the same conclusion as the 2022 paper – Current manuscript: “only more aDNA data from these undersampled steppe areas would allow to refine the sources of admixture” 2022 paper: “Instead, it rather matches the steppes north of the Caucasus, although the scarceness of comparative data from the steppe in the 1st millennium CE calls for a future investigation of possible better alternative sources.”.

R: *We are very grateful to the reviewer for this feedback. We agree with the main point made by the reviewer about length and novelty, so we have now moved down, shortened and restructured this section following these feedbacks.*

In particular, we agree this is indeed not the main topic of the article, that is the kinship, social structure and community shift/local power realignment. But we still believe that a short chapter describing the population genomic ancestry of these communities is essential for their correct contextualization to show that these communities had also steppe descent from their genomic ancestry. But to place the focus on the main topics we moved this section as the last of the results, therefore moving up all the other sections and figures as the PCA figure becomes now the last one. Furthermore, as the reviewer suggested, we mostly focused on highlighting the main novel finding respect to previous publications on the topic (Gnecchi-Ruscione et al. 2022; Maróti et al. 2022). See following point-by-point responses below.

It should be noted that following the main suggestion by the reviewer to shorten this section, most of the details about the specific points made by the reviewer below do not find space in the new version of the main text. But they were already or we have now included these aspects in the revised version of the Supplementary Information at pages 30-35.

C: 2. Context: A related issue is that it is very hard for the reader to identify what the new findings in this section actually are. You really would have to be an expert on Avar ancient genomics to understand how these new data have significantly changed our understanding of this population's migratory or admixture history. There are a lot of confirmations of Gnecchi-Ruscione et al. 2022, but I believe there are also rejections/refinements of some interpretations made in that paper? This is not made clear to the reader. For example, the final line of the section says the evidence presented "poses a challenge to the hypothesis that there were successive large-scale migrations from the steppe". However, no citation is given for this hypothesis. Is this a rejection of the interpretation from Gnecchi-Ruscione et al. 2022 that: "This non-local admixture together with the retention of a high level of eastern ancestry in the late elites and the absence of genetic inbreeding may point to continued migration from the steppes after the initial arrival of the Avars in the Carpathian Basin".

R: *We are thankful to the reviewer for this constructive feedback. We have now re-written these parts (see previous response) highlighting more explicitly what are the novelties with respect to previous studies. Regarding the example mentioned by the reviewer: the hypothesis of successive migrations from the steppe is a long-lasting hypothesis and one of the main unsolved questions in Avar archaeology and history that has been first formulated in the 1970s. We have now included the correct references in the main text (line 341). Previous genomic studies did not solve this controversy. In Gnecchi-Ruscione et al. 2022, as the reviewer noted, the topic is mentioned, but still left as an open question. From our new analyses of isotopic and genomic data we have no indication that such large-scale migrations occurred at our sites. This topic is only mentioned now in the main text and discussed in more details in the SI (pages 53-55).*

C: I think the authors' conclusion that admixture dates support most admixture occurring prior to Avar arrival in the Carpathian also departs from that of the 2022 paper?

R: Yes, the reviewer is correct. In the 2022 paper we had mostly samples from elite sites of the DTI, the majority of these individuals had the highest amount of Northeast Asian ancestry (for example the Early Avar period, EA, group that here we named DTI_EA_East). The fewer admixed individuals had variable admixture dates and did not allow us to fully capture the admixture process. With the new extensive dataset in this article we are able to capture this admixture process, that, as the reviewer correctly stated in the comment below, is continuous but the majority of it occurred before the Avar period (80% of admixture dates are pre 568 CE with a median date of 490 ± 73 CE), therefore before their arrival in the Carpathian Basin.

C: 3. Clarity and Conciseness: Again related to the above, there is no clear summary given for the current model for the formation of Avar populations in the Carpathian Basin. The title of this section is very long and vague and does not provide any overview of the main results within. The abstract also provides no one-line summary of this section. There is also no concise conclusion at the end of the section. In fact, the “in conclusion” paragraph is longer than the rest of the section and introduces new results, including strontium isotopes. This poor structuring immediately makes the paper feel inaccessible to the non-specialist reader.

R: We are very grateful to the reviewer for this constructive feedback. We have now shortened and restructured this section following the reviewer suggestions. We have changed the title of the section and added a concise conclusion sentence at the end of the section (lines 337-343) and a summary line in the abstract (lines 43-44 and 52-53).

C: 4. Interpretation: The authors seem to conclude that only a minimal amount of admixture took place within the Carpathian basin during the Avar period, but no exact estimate is given. The argument for this is not made very coherently. I’m not unconvinced, but I find the logic hard to follow.

R: We thank the reviewer for noticing this imprecision in our wording. A “minimal amount” is a misleading expression that we deleted, the admixture post-arrival in the Carpathian Basin that we estimated is ~20% (so a “minor fraction” but not “minimal”) from DATES that corresponds to the proportion from the qpAdm models that can be resolved.

C: Firstly, it is not explained how Sarmatian ancestry differs from “main European” ancestry in the pre-Avar Carpathian basin or if Sarmatian-related groups persisted into the Avar period.

R: This can be appreciated in the Extended Data Figure 8b. The Sarmatian gene pool extends along a cline that goes from central Europe towards the steppe and it can be modeled in qpAdm with a 10%-15% influx from Iron Age Scythians from across the steppe (Gneccchi-Ruscone et al. 2022). The Pre-Avar period 6th c. cloud instead lacks the steppe influx but extends along a north to south Europe cline. This was described in previous publications with published data data we use as proxies in all or qpAdm models: the Sarmatian period 2nd-5th c. CE ancestry (represented by the published reference groups from the Hungarian sites published in Gneccchi-Ruscone et al., 2022: Sarm_P_DTI_4_5c, Sarm_P_TT_4_5c) and the Pre-Avar period 6th c. CE ancestry (represented by the published reference group from the Hungarian

site of Szolad of the Longobard- period published in Amorim et al., 2018: Lon_P_south_6c, Lon_P_north_6c).

Our new data from pre-Avar RK individuals corroborates this finding as the Sarmatian period individuals from RK can almost exclusively be modeled as belonging 100% to the Sarmatian period genepool while the Gepid period (6th century) RK as well as the un-admixed Avar period RK individuals can be mostly modeled as belonging on the north-south axis of variation previously observed in the 6th century population in Szolad (West Hungary, Longobard period; Amorim et al 2018). Nevertheless, the two clouds are in part overlapping (suggesting also a partial continuity between the time periods, partial continuity that extends also in the Avar period). This partial overlap makes it more difficult to distinguish between these two components in the ancestry models of the Avar period individuals detailed below.

We added this description in the Supplementary Information (pages 32-33) as there is no space to go into these details in the main text.

C: For example, in Fig. 2B it is not obvious what population is represented by the gray “Pre-Avar Carpathian Basin” and the piecharts in Extended Data Fig 1C don’t help matters as their relationship to the 3-way qpAdm is unclear. I also don’t follow lines 164-169. Do 20% of individuals in RK have some “main European” ancestry? If so, which source populations and how much?

R: We thank the reviewer for these questions that made us realize that the way we simplified the visualization of the results was misleading and we therefore changed the visualization. The gray color was used in an attempt to simplify for the reader the complexity of models that were explored, all reported in (Supplementary Table 7) and included both the Sarmatian period 2nd-5th c. CE proxies and the Pre-Avar period 6th c. CE proxies described above. We have now restricted the summary qpAdm barplot only in the Extended Data Fig. 8c. since we are not able to go into the details of the qpAdm models in the main text.

In an attempt to solve the misleading visualization, the “gray” component in the plot is now representing all the West Eurasian sources (including also North_Caucasus_7-11 c CE). This because the “competing” qpAdm model strategy used (below more details) involves all West Eurasian sources tested (therefore also including the North_Caucasus_7-11 c CE). So, by coloring all these sources in gray we reduce the admixture models to the most simplified and formally correct (all working models) East + West Eurasian admixture. This is also the simplified model tested with DATES for admixture dating (see Methods). Now, these qpAdm models are formally correct but do not help to distinguish between these West Eurasian sources that could inform about what ancient population proxies most resemble this West Eurasian components that could in turn suggest where this admixture could have happened (e.g. in the steppe or after their arrival in the Carpathian Basin?) and provide independent evidence from admixture Dating.

In the attempt to gain more insights into the composition of these West Eurasian components we applied the qpAdm model framework described in the method section, the same applied in Gnechchi-Ruscone et al. 2022 that is based on guidelines, considerations and strategies described in Harney et al. 2021 and recently evaluated also in Yüncü et al. 2023, were we make the different West Eurasian sources “compete” in a 3-way model (always with the East

Eurasian ancestry proxy present) when the simple 2-way models are unresolved (which is the majority of the cases). This approach helped us to resolve the majority of the models, although not all of them (e.g. in RK we still have a 22% unresolved 3 way mixtures, where different sources produce equally fitting models; answering the question about the 3-way mixtures). We declare these models unsolved (the mixture therefore remains an unspecified mix of West Eurasian sources).

Nevertheless, combining the results of the resolved models with the independent evidence of admixture dating from Dates, we reach the same proportion of individuals that carry admixture dating post-Avar period (post 568 CE; 20% of dated individuals, overall and in RK) and with a West Eurasian source matching only the Pre-Avar period 6th c. (20% of admixture models working only with Szolad/Lon_P proxies in RK).

We report under the bar plots of Extended Data Fig. 8c as pie charts, the summary of the qpAdm models retained after the competing approach for the two most numerous groups (KFJ and RK). The individual specific qpAdm models for all the individuals can be visualized in the Supplementary Fig. 12 to 14 and Supplementary Table 5 & 7.

C: It's also not clear how many mothers you would be expecting to be unadmixed to give rise to this 20% estimate, especially as the authors point out later they may not have sampled the earliest generation of migrants.

R: *We thank the reviewer for this comment. We realized that this sentence is in fact simplistic and misleading. We agree with the reviewer that it is not straightforward to estimate how many “European” mothers one would expect to give rise to 20% of admixture. Therefore, we removed this sentence from the new version of the main text.*

C: Fig 2B shows most individuals are majority gray ancestry, so it seems the conclusion that minimal admixture took place in the Carpathian Basin rests solely on admixture dates and a hypothetical Sarmatian-related population further east (rather than “multiple lines of evidence”)?

R: *We thank the reviewer for this comment. We hope that with the modifications detailed above, it is now clear what the “gray” color represents (i.e. all the West Eurasian ancestries / proxies tested). We removed “multiple lines of evidence” as well as “minimal” (20% is not “minimal”, this was our language mistake) and we stated clearly in the text that this proportion is derived from the independent analyses of DATES and the qpAdm model results. To be fair, with “multiple lines of evidence” we originally intended also the archaeological connections between the sites we analyzed and sites in the Eurasian steppe. A short summary sentence of these parallels with archaeology is in the main text (lines 333-336). A more exhaustive description of this topic can be found in the Supplementary Information (pages 10-11 and 33-34).*

C: The authors also say that there is “regional genetic homogeneity throughout time” (Line 190), but Figure 2 shows clear temporal shifts in ancestry towards west Eurasian populations for every site where there is multi-period sampling. Admixture dates for many LA samples also

fall within the EA period (Extended Data Figure 2). The fact that LA samples have later dates than EA samples also suggests continuous admixture.

R: *We thank the reviewer for this comment. We agree with the reviewer that there is a slight trend of increasing admixture through time when this can be evaluated (in RK and KFJ). It has to be specified that this observation does not apply to KUP. In KUP the Early and Late period burials belong to very different communities, archaeologically and genetically (also don't share IBD between them; Fig. 3). So for this site it would be misleading to state that there is an increasing admixture through time, as in fact there is no continuity between these two communities separated by ~100 years from one another. We have written explicitly this observation for RK and KFJ in the Supplementary Information (page 33). Although, this is in line with a certain amount of post-arrival admixture that we estimated with our analysis so we do not think this observation is surprising nor contradicts any of our analyses.*

We instead believe that the "regional genetic homogeneity throughout time" is something striking. This homogeneity does not disprove that there is increasing admixture through time. With "regional homogeneity" we mean that the admixture gradients of the different communities remain distinguishable from each other through time. This pattern can be observed in the distribution of admixture clines between RK and KUP/KFJ overall spanning the whole Avar period (see comment below for HNJ). Assuming a "random mating" between the KUP/KFJ and RK communities we would observe a rapid increase in overlap in the admixture proportion between them. Instead they remain non-overlapping throughout the whole Avar period. We have written this at lines 325-332.

C: The arguments made from isotopic data also appear weak. The authors also concede that distant areas may be isotopically indistinguishable, but then go on to argue for mobility over short distances. They presume they have not managed to sample the first generation of migrants, even though sites were sampled exhaustively. Is it a possibility that the first generation migrants have indeed been sampled, but come from a distant but isotopically indistinguishable area?

R: *We thank the reviewer for this comment. We break down this comment in the two conclusions drawn from the co-analyses of isotopes and genetic structure.*

- The first conclusion is about the first generation of migrants:

Unfortunately, it is not possible to definitely exclude that there were migrations between distant but isotopically indistinguishable areas. But the extremely homogeneous isotopic ranges within all the three sites that we analyzed would imply that these migrations always occurred between two sites with the same isotopic range, which seems less likely. This led us to presume that we might have not sampled in most cases, the very first generation of migrants. Although one early Avar-period isotopic outlier could in fact be a first-generation immigrant, unfortunately this was one of the few samples that did not have enough DNA preservation so we don't know its ancestry nor its eventual relatedness to the others. This may suggest that we can find sporadically the immigrant generation. Just for clarity, we didn't sample the entire cemeteries for isotope analyses but we did have a big sample size overall. Although, the explanation for this apparent contradiction would be that the first generation of migrants did

not use these cemeteries or at least not many of them. This is in fact an interpretation based on broad archaeological (scattered graves in early generations) and historical (high mobility of Avars in early period) evidence on the Avar period.

- The second conclusion regards the presence/absence of signs of successive migrations from the steppe during the Avar period:

This question is mostly based on the evidence from RK which is the only site spanning continuously the whole ~250 years of the Avar period and the three chronological phases, early middle and late. Following the reasoning above it seems even less likely that there were successive migrations that again should have been from distant regions with the same isotopic signal. Therefore, the lack of isotopic heterogeneity/discontinuity together with the lack of major genetic discontinuity during the course of the Avar period in RK suggest that there was no distant migration from the steppe, but rather local mobility that went undetected by the isotopic data. We state in the text that the successive migration hypothesis is “challenged” and not “disproved”, because for a definitive answer we think it would be necessary to investigate this on a larger number of sites.

C: I'm also not convinced that various steppe descent groups remain genetically distinguishable in different parts of the Carpathian Basin, given the heterogeneity seen in each site (lines 169-172). I don't think I could confidently differentiate between a KFJ and HNJ sample on the basis of ancestry.

R: *We thank the reviewer for this comment. We agree with the reviewer that KFJ and HNJ have admixture distributions partially overlapping (although they do not overlap for the most part, see Figure 4b). HNJ is the smallest site that we analyzed and it is only dated to the transitional middle Avar period and therefore we cannot evaluate its changes through time. We have now specified in the main text that we base our interpretation mostly on the distinction between the distribution of RK and KUP/KFJ that do not overlap throughout the whole Avar period.*

-> Kinship sections

C:

1. Pedigree and Kinship patterns
2. Polygyny, levirate reproductive unions and consanguinity
3. Exogamous females and connections between communities
4. Shifts in local communities from the Early to the Middle/Late Avar periods

These four sections contain the most interesting results, but their structure could be much improved. Related sets of results are scattered throughout multiple sections (which I'll label 1-4 for convenience). For example, we're told in line 255 that only one Y lineage (N1a) is found in both KFJ and KUP, but have to wait until line 344 to find out these males cluster

together in IBD analysis.

R: We agree with the reviewer that this section contains some of the most important results. We have therefore moved this whole section up. as the first Result section. See below for further adjustments.

C: RK is the main case study. There is a huge amount going on at this site and it would be easier to digest if most of the site-specific results from RK were presented together in a single section, after the more general trends across sites were highlighted. As it stands, the reader is drip-fed key findings: In section 1 we find out two Y lineages dominate RK, associated with certain pedigrees; in section 2 we find many females at RK have multiple reproductive partners; in section 3 find these females have a key role connecting these different patriline and in section 4 we're told one patriline appears to replace other, the connection being between two maternal half-brothers. This is a very long-winded way of telling a truly fascinating story.

R: Indeed, RK is in a way the main case study. It is the largest cemetery, the largest pedigree and the one that yielded the greatest number of different pieces of evidence. It is for example the site of the local community realignment finding, and in that chapter only RK is discussed. However, in the pedigree chapter we believe that the strength of our study is that the patterns that we observe can be found across multiple sites: this allows us to speak about the Avar society more broadly. In this chapter RK is still the most discussed, as it is the largest pedigree and site but we believe that presenting the evidences of patrilineality and levirate of RK together with the ones found in KFJ (which is still an extensive pedigree of ~50 individuals which is alone among the largest ever published for an ancient community) and the other sites, gives much more support to our findings.

C: Currently, sections 1 and 4 are focussed mainly on RK, while sections 2 and 3 are more evenly split between RK and more general trends seen across sites. My suggestion is that the authors try put the more general findings first, followed by a deep dive into RK and the incredible details they have gleaned from that site. For example:

1. Connections between communities: patrilocality and female exogamy
2. Pedigree Reconstruction: Strict Patriline and Levirate customs within sites
3. Rákóczfalva: capturing shifts in local power structures

The initial case for patrilocality and female exogamy can be made clearly with patterns of IBD sharing among adult males and females, alongside Y chromosome and mtDNA diversity. The IBD sharing data also gives the reader a nice first overview of the pedigrees/kin groupings present at different sites. Pedigree data from across sites can then be used to further demonstrate patriline, female exogamy as well as polygyny/levirate unions/inbreeding. The stage is then set for the RK case study, including integration of spatial and grave good data, refinements of site chronology, isotopes, all of which is needed to fully contextualise the replacement of one patriline with the other.

R: We thank the reviewer for this in-depth suggestions that stimulated us to rethink the structure and titles of the sections. We have reordered the part into three sections and used the suggested titles (with a small change). We have attempted the order of these sections as suggested but for our interdisciplinary co-authors it was not so straightforward to see the case for patrilocality and female exogamy easily on IBD data alone: the arguments based on pedigrees were more natural. Hence, we have them in the order of pedigree reconstruction (section 2 in the reviewer's numbering), connections between communities (section 1) and RK section (section 3). As suggested, we have also shifted the integration of site chronology, isotopes and archaeological data to section 3 (from the pedigree section; lines 250-255).

Small comments:

C: 1. The Y chromosome haplogroups given in the main text are very basal (e.g. J1a and J2b). It could be made clearer that the haplotype resolution is much higher for most individuals and still is consistent with a small number of dominant patriline (e.g. J1a2a1a2d2b2b2 and J2b2a1a1a1a1a1e).

R: We agree with the reviewer, we added the terminal SNP instead of the full y-haplotype resolution as it is the most common practice in literature and once declared the first time in the text we use the abbreviations for brevity reasons (lines 149-153).

C: Some grammatical issues:

Lines 110-111 two “ands” in a row without comma, makes it hard to know if the core region is “Danube-Tisza Interfluve and Hajdúnánás” or the two cemeteries from the neighbouring region are “Hajdúnánás and Rákóczifalva”.

R: After the extensive restructuring suggested by the reviewer this sentence was removed.

C: Lines 129-130: This is hard to understand. Unclear if there are cultural/social differences between different regions on the same side of the Tisza (and we see this on both sides of Tisza).

R: After the extensive restructuring suggested by Reviewer 2 and better representation of archaeology asked by Reviewer 1 this sentence was removed and cultural differences between DTI and TT clarified.

C: Lines 152-14. Again too many “ands” makes the sentence harder to follow. Could be made more concise.

R: After the extensive restructuring and shortening of this section suggested by the reviewer this sentence was revised and changed.

C: Lines 157-159. This sentence is hard to understand. There shouldn't be a comma before “is very limited”.

R: After the extensive restructuring and shortening of this section suggested by the reviewer this sentence was removed.

C: Lines 157-159. To simplify and keep consistent with terms used higher up in the paragraph, I would change “working with groups representing the main European genomic profile found in the 5th-6th CE and during the course of the Avar period in the Carpathian basin” to “working with western Eurasian ancestries associated with the Carpathian basin in the Avar and preceding periods”.

R: After the extensive restructuring and shortening of this section suggested by the reviewer this sentence was removed.

C: Lines 162-163: “would allow to refine the sources of admixture” -> “will allow the sources of admixture to be better refined”

R: After the extensive restructuring and shortening of this section suggested by the reviewer this sentence was removed.

C: Lines 186-188. “On the one side” is not an easily understood phrase. It implies these two conclusions are meant to be contradictory, which they aren't. The first generation of migrants may not have been sampled AND there may have been high regional continuity across the period.

R: We agree and rephrase as the reviewer suggested.

C: Line 206-207: “was closely related” -> “were closely related”

R: We corrected this grammar mistake

C: Line 255-256: “is found between” -> “is found in both”

R: We corrected this grammar mistake

C: Line 270-272: Hard to understand. In the vast majority of cases?

R: We changed this sentence after this and a comment from Reviewer 3

C: 340-343: This sentence doesn't make sense and is incorrectly capitalised.

R: We corrected this sentence, broke it apart to make it more comprehensible.

C: Line 359-361: Hard to follow – needs to be better phrased. IBDs -> IBD segments.

R: *We corrected this sentence to make it more comprehensible.*

C: Line 378: “the burial with horse harness” -> “such as burial with a horse harness”

R: *We corrected this grammar mistake.*

C: Line 391-393: Hard to follow. “while overall ancestry” ->“while the overall ancestry”

R: *We corrected this grammar mistake and rephrased to make it more comprehensible.*

Figure issues

C: Fig 2b. Text is different sizes and too small for final group. Why are RK individuals separated by relatedness? If there is a difference here it is not highlighted in the main text.

R: *We have now revised the figure. The previous Fig. 2b (the barplots with qpAdm models) are now in Extended Data Fig. 8c and we grouped RK altogether, and increased the sizes of the text following the reviewer's suggestions.*

C: Fig 3. Timeline needs date intervals marked. Why are the Y lineages for unsampled males not shown, when they can be inferred from their sampled sons? In pedigree 4 it looks like two brothers are also 3rd degree relatives?

R: *The chronology based on the archaeology of the burials, the 14C dates and the generation of the pedigrees overall match quite well (See Supplementary information pages 71-81) but adding the date intervals to this timeline would be misleading as it is not possible to directly translate the generation of the pedigrees in precise calendar years. These might also not overlap in many cases (e.g. a young infant of generation 3 might also be chronologically earlier than its old grandfather of generation 1). The timeline here reflects the overall continuous chronology from the early to the late Avar period and the color is just a graphical representation of this continuity: a color gradient going from red (used in the other plots for the early period) to yellow (used in other plots for the late period).*

The reviewer is correct, the unsampled fathers should have the same y-haplogroup than the sampled sons. But we do not think it is necessary to color them in the plot, the figure is already quite content-rich, we prefer to leave all the unsampled individuals white as we in fact do not have any real data on them. We thank the reviewer for pointing out the ambiguous drawing. The two brothers of pedigree 4 are first degree related siblings. Both of them are third degree related to the 6 brothers of pedigree 5, that is the reason for this intermediate step with the dotted third degree line. We changed the third degree line to make it clearer that it is not meant to connect the two brothers directly but just to avoid doubling the numbers of lines.

Referee #3 (Remarks to the Author):

C: Gneccchi-Ruscone, Rácz et al. present results of an analysis involving the reconstruction of multigenerational pedigrees using genomes from Avar-period cemeteries. This analysis extends recent work by Gneccchi-Ruscone et al. Cell 2022 to investigate cultural interactions following the formation of the Avar groups. The paper is interesting and well written, although several parts of it are outside of my expertise. This review focuses on statistical and genealogical aspects of this work, rather than on the archeological and anthropological findings, which I hope others will be better positioned to comment on.

Major comments:

- Genetic maps: My understanding is that the authors are using the same genetic map to determine the length of IBD segments for all pairs of individuals (although I could not find details on this). This genetic map is likely computed as an average between male and female recombination rates, which vary substantially (see e.g., Kong et al. 2010 Nature, Caballero et al 2019 Plos Genetics). This affects the calculation of IBD segment lengths in cM, used in several steps of the analysis. For instance, $\langle k \rangle$ and $\langle w \rangle$ observed in females are lower but this may reflect the fact that females transmit segments that appear to be shorter if a sex-averaged map is used.

We thank the reviewer for this comment, giving us the opportunity to clarify this important aspect. First of all, yes, we use the HapMap averaged genetic maps for imputation and IBD analyses. We added a sentence in the Methods clarifying this (lines 765-766). The reviewer is correct that recombination rates between males and females vary as females produce 1.6 times more crossovers and that this has been investigated as the reviewer noted, most recently in Caballero et al. 2019. This article shows that the difference between using sex-specific (SS) vs sex-averaged (SA) genetic maps in the IBD sharing pattern is not drastic but it is indeed quantifiable at the close genetic relative level (siblings and half-siblings and other second degrees). At higher degrees of relatedness, the effect of different recombination rates between males and females is averaged out and there is not a substantial difference when using SS or SA (see Figure 1, Fig. S4 and Fig. S15 from Caballero et al. 2019). Furthermore, this difference in recombination affects the number and length distribution but not the total length of IBD (The x axis of the plot below and the measure used for estimating the $\langle w \rangle$ statistics from the network, see also response to point below on this matter). Therefore, the $\langle w \rangle$ statistics should not be affected by the different recombination rates. The $\langle k \rangle$ statistics only assesses if two individuals are connected (have at least one IBD segment regardless of the length). So, at lower degrees/closer relatives (1st and 2nd, 3rd...) individuals (males or females) anyway share many IBD segments so the recombination differences would not affect the $\langle k \rangle$ distribution between close relatives. At higher degrees / distant relatives, as said above, the recombination difference averages out.

C: Two questions arise: (1) are the analyses based on IBD segments reported here, for instance on male vs female IBD graph connectivity, robust to the use of a sex-averaged map? This could

be assessed using simulations where sex-specific maps are used to generate the data, but sex-averaged maps are assumed during analysis.

This is a good suggestion. In fact, KIN, which is the main method used to infer relatedness used to construct pedigrees in this study, has been already tested on simulations based on sex-specific maps and was found to be robust to this effect (Popli et al. 2023; see also comment below). Sparked by the reviewer suggestions, we are now including the simulations done for ancIBD in the Ringbauer et al. 2023 article (discussed in the next points), these were run with sex-averaged maps. The results from KIN and ancIBD extremely match (see newly added plots discussed in the following points) and since one is not affected by using sex-averaged recombination maps, it stands to reason that the other is not affected either. And since the IBDs detected do not differ, nor should other analysis based on them (like our network and female and male connectivity). Additionally, Caballero et al 2019 in Fig.3 shows that the differences in the presence and length distribution of IBD segments between SA and SS does not differ on the level of 3rd cousins and onwards and our connections between the archaeological sites are of a higher degree than that (see also the discussion in the previous point).

C: It would also be useful for this manuscript to have a supplementary figure showing the distribution of number, length, and sum of IBD segments as a function of relationships and number of males/females in the genealogical path between two individuals. (2) Can the authors take this into account to refine their estimated IBD segment lengths and downstream analyses? (This would be nice if achievable, but a discussion would be sufficient.)

We thank the reviewer for this idea. After this and the following comment from the reviewer we agree that we should report more about the IBD results and how they compare with the relatedness from KIN and the one finally built in the reconstructed pedigrees. So we have now produced plots in line with the reviewer's suggestions with the sum of IBD vs the number of IBD (>12cM, see also response to point below). We added these plots in the response below and in the Supplementary Information (Supplementary Fig. 19-22)

So, we compared the sum of IBD vs the number of IBD of our data to the simulated pedigree data (run with PedSim) on which ancIBD was run with the same parameters (simulations were taken from Ringbauer et al. 2023). Caballero et al. 2019 estimates that successful detection of IBDs can extend up to 6th degree which is the maximum degree that was simulated in Ringbauer et al. 2023. From the relationships built in the pedigrees we reported even up to 8th degree since it was possible to identify such distant relatives from our extended pedigrees. From this comparison it is possible to appreciate a striking match between the inferred relations from the pedigrees and the expected distribution of IBDs from the simulations. It is also possible to observe that there is no enrichment for low numbers of IBDs in the real data compared to the simulations also for higher degree relatives (3rd to 8th). We also looked, as mentioned in the previous response, at the correspondence between KIN relatedness assignments and ancIBD (plot below). From this plot, it is possible to appreciate the striking correspondence between KIN and ancIBD.

Importantly, thanks to the reviewer's suggestion to trace down the relatedness path via paternal and maternal line, we were able to annotate for all our half-siblings if they were through maternal or paternal line. We chose half siblings as from Caballero et al 2019 they result

among the most affected by the different recombination rate. Our results are in line with what Caballero et al 2019 obtained and could also serve as a further way to validate half-siblings in future studies.

C: - Phasing, imputation, and IBD detection: L726: “Obtaining such data with ancient genomes has been proven reliable...” This seems a bit too strong. Surely switch error and genotyping error rates remain quite high, so “reliable” entirely depends on specific applications. Relevant to this work, accurate IBD detection requires high quality phasing and genotyping and the ancIBD approach that underpins several analyses in this work is still under review. I would therefore recommend that the authors perform independent validation of the precision and recall of ancIBD using simulated pedigree data to rule out potential biases (also see my other comment on more realistic simulations, which would also apply to these simulations).

R: We thank the reviewer for this comment. We realized the expression we used is too strong as a general statement and we agree on the fact that the “reliability” of imputations depends on the specific applications. Therefore, we specified that phasing and imputation is reliable in our specific application as it was shown reliable in other similar contexts (early Medieval Europe & Avar contexts) and applications (ancIBD analyzes), e.g. Gneccchi-Ruscione et al. 2022, Rivollat et al. 2023, Penske et al. 2023) and a recent study shows that in fact among the ancient setting tested, for European historical time periods imputation works best (Sousa de Mota et al. 2023). We agree with the reviewer about comparing our IBD results with the ones from simulated pedigrees, we did this as discussed in the previous comment though we note that the results agree with relatedness detected with other methods in our application and also in the application in the recent study (Rivollat et al. 2023).

C: - Pedigree simulations: the simulations of Supp Fig 16 and 17 are a bit too simplistic. Most importantly, they use 22 chromosomes of 100 cM each. This may be fine if the goal is to observe general trends, but my understanding is that these simulations are used to establish a null model, which is then used to reject relationships in real data, which has a different total cM length. Unless I misunderstood, this analysis needs to be revised. More broadly, I would recommend making the simulations of L702 more realistic by (1) using the correct chromosome sizes; (2) using a genetic map (possibly sex-specific); (3) matching the genotype density of real data; (4) modeling low coverage data, genotyping errors, phasing errors. (It may be also relevant to consider varying the demographic model, as homozygosity in the pedigree founders due to a small effective population size may affect IBD detection accuracy.)

R: *We thank the reviewer for this comment. We have now implemented the suggestions in our simulations. (1) We simulate realistic chromosome sizes (as per GRCh38.p14 genome). (2) We now use Ped-sim (Caballero et al., 2019) to get recombination points using sex-specific genetic maps. (3) Our data is captured with 1240K sites, and so we now use around 1400K sites in our simulations to make IBD inferences. (4) We match the coverage in the simulations to that of average coverage of our samples (~2.6x). Our libraries are captured with the 1240K array, so the genotyping error is lower (because the alleles of the 1240K SNPs are known and errors can be detected more easily) and we do not need to phase our data to get IBD estimates using KIN. We do not consider cases with different amounts of inbreeding and/or different population sizes, since KIN explicitly models the effect of ROH on IBD detection, and has been shown to perform well even in the presence of ROH (Popli et al., 2023). We have now changed lines 732-737 in the main test to describe the simulations. We would like to note that for the majority of the second degrees that we built in the pedigrees we were able to distinguish the type of relation (avuncular, grand-parent/child, half-siblings) thanks to some connecting first-degrees around them (i.e. two individuals classified as 2nd degrees because we have the child/parent in between that connects them can be classified as grand-parent/grand-child). Furthermore, in*

general the relatedness estimates for first and second degrees are highly concordant between the different methods applied: KIN, the main one we used to build the pedigrees, BREAD (an R package that estimates pairwise mismatch rate, the “classical” relatedness measure used in ancient DNA studies) and ancIBD (Supplementary Information pages 46-52).

So, these figures (now Supplementary Fig. 17 and 18) are only showcasing a few specific cases (that would not in fact have major impact on the overall pedigree structure and its interpretation) where we wanted to resolve, or at least point to the most plausible type second degree relation in situations where there aren't first degree connections around them.

C: - Admixture analyses: Please clarify statements such as “resulted in a significant rejection” (L623) and “did not significantly reject” (L625). What constitutes a significant rejection? L620 reports 0.05 as a significance threshold. Were significance levels properly adjusted for multiple testing? This also applies to other parts of the manuscript, e.g., (I believe, but see my comment on Supp Table labeling) Supp Table 4.

R: *We thank the reviewer for this comment. We clarified these sentences in the text (lines 647-649). We have added more references as the reviewer suggested below about the qpWave/qpAdm approach (see lines 630 and 643-644). Our aim was to use the method as previously established and repeatedly and widely applied in the field of ancient human population genomics, hence we have opted not to go into the details of how the method works. The core statistics underlying this method are the set of f_4 -statistics between left and right populations (i.e. the target/sources and the outgroups). It was in fact previously discussed that it is not straightforward (and practically not possible) to do multiple testing correction as one cannot simply adjust by number of tests because all the f_4 -statistics are highly dependent between each other as are the different individuals tested as target of admixture. Therefore, by consensus the correction for multiple testing is never done in qpWave/qpAdm modeling. For original development of the method see Lazaridis et al. 2014 for qpWave; Haak et al. 2015 for qpAdm; a recent discussion on multiple testing can be found here <https://rdrr.io/github/uqrmaie1/admixtools/f/vignettes/qpadm.Rmd>; and an extensive testing of the method, how best to use it and different possible model strategies, can be found at Harney et al. 2021. The limits of qpAdm framework has been under discussion (Maier et al. 2023), not for reasons of multiple testing, but we are not using the approaches criticized (e.g. rotating approach with proximal sources).*

C: Additional comments:

- I don't understand the choice of only filtering out pairs of individuals sharing a single segment within the range of 12 to 16 cM. Why not apply this to >16 cM? Close relationships should also result in multiple shared segments. The authors should provide more support for this choice, which appears arbitrary.

R: We thank the reviewer for this comment. It is correct, ancIBD paper discourages trusting segments $< 8\text{cM}$. We decided to keep an even higher threshold and filter out IBDs $< 12\text{cM}$. This for two reasons: first, because we are more interested in “closer” relatedness patterns between the communities rather than background population ancestry relatedness; second, because ancIBD still has some small false positive rate between 8cM and 12cM (4% to 1%, see Figure 2b in Ringbauer et al. 2023). So we considered only pairs with $\text{IBD} > 12\text{cM}$. Then for IBDs between 12cM and 16cM we considered pairs that had at least one other IBD segment $>12\text{cM}$ and not just a single one. The reviewer is correct that this last filtering is somewhat an arbitrary cut-off, but the reason we did this was to be on an even more conservative side of reducing further false positives cases as of single “shorter IBD” are more prone to error even if is it around 1% or lower for segments between 12cM and 16cM , there are still some “bumps” in the trend (Figure 2b in Ringbauer et al. 2023). Therefore, we are aware that with this filtering we also remove some real IBD links but given the abundance of the data we opted for a more conservative filtering. Also, we agree with the reviewer that this affects only very distant relatives. Closer relatives (1st-6th degree relatives) share indeed multiple IBD segments of various lengths and not just one (see Supplementary Table 5 and plots above). In order to test if this additional filtering had any effect on our downstream analyses, we have now replicated all our statistics on a network where we keep all the pairs with IBDs $> 12\text{cM}$ even if in single copy. The results are highly concordant and don’t change the observed patterns and their interpretations. (We have added this further control analysis in the Supplementary Information pages 86-88 Supplementary Fig. 49-50).

C: Also, ancIBD recommends only considering segments $>8\text{cM}$. The authors first state they considered segments $>5\text{ cM}$ (L754) and then $>12\text{ cM}$ (L756). Is the $>5\text{ cM}$ filter not superseded by the $>12\text{ cM}$ filter? Please clarify.

R: We thank the reviewer for pointing out this discordance. In fact the “ $>5\text{ cM}$ ” was a typo as ancIBD considers only segments $>8\text{cM}$ and with our additional filtering we kept only $>12\text{ cM}$ segments as stated in the previous comment. We corrected this typo.

C: - “Maximum IBD value” in L771 is unclear. But based on the values reported in the caption of Figure 4 I suspect it is the largest segment found in the pair. I do not understand the choice of this statistic as it is more noisy than other natural choices such as the sum of the lengths of all segments found in the pair (or equivalently the fraction of genome shared). The authors should provide support for what appears to be an odd choice or consider using the sum instead.

R: We thank the reviewer for this comment. The reviewer is correct that the “Maximum IBD” corresponds to the largest segment found in the pair. The reviewer is also correct in saying that it is a noisier statistic but we only use this for the visualization purposes, as the IBD networks shown in the figures are all unweighted (the network is only informed about presence/absence of a link). The “Maximum IBD” is therefore only used as a graphical metrics to weight the thickness and color of the edges of the network plots. We use “Maximum IBD” rather than the Sum of IBDs only for the graphical reason that the Maximum IBD range is much smaller as it only can vary between ~ 12 and $\sim 283\text{cM}$ which is the maximum length of the longest chromosome (which is the case of all the parent-child relationships), while the maximum Sum of IBDs varies between 24 and $\sim 3000\text{cM}$. We have however calculated in the Supplementary Information the strength statistic using the “Maximum IBD” as weight. We have now done the same using the “Sum of IBDs $>12\text{cM}$ ” to test if there were any differences using this less noisy statistic. The results are qualitatively the same. We replace the plot in the Supplementary with the strength statistic calculated on the “Sum of IBDs $>12\text{cM}$ ” as per the reviewer’s suggestion.

C: - Admixture analyses: I am generally a bit uneasy about analyses that compare several parametric models where the space of possible combinations is extremely large, which are affected by the choice of models being compared and the availability of appropriate reference populations. The authors do a good job at describing the rationale of their choices, but I would recommend they add more methodological context by providing references to other papers that used model selection strategies like the one adopted here.

R: *We thank the reviewer for appreciating our effort in describing the model strategy. We have added references for our selection strategy (lines 630 and 643-644). We would like to stress once again that qpAdm modeling should be seen as a “model rejection” approach rather than an approach aiming to find the best fitting model. It is true that the space of possible combinations is large and we are naturally limited by what reference populations we have available from published genomic data. This is why we do not aim to find the best-fitting model but rather aim to have enough resolution to be able to test models that we can reject compared to others. We would still have many alternative fitting models but we focus the interpretation on the ones we have the resolution to reject based on the available data. (e.g. we can reject that the admixture model matches early medieval “Central European” genomic profiles but we cannot reject that it matches early medieval Caucasus sources, so we keep the latter model because we rejected the first). We explained this further in the Supplementary Information (page 31), where we discuss qpWave/qpAdm results in more detail.*

C: - L164: How was the 20% admixture with European populations estimated? Please clarify the methodology and make admixing populations more explicit.

R: *This section has now been revised after the comments from reviewer 2 and 3 and it is now written explicitly from where we derived this estimate. Mostly from the admixture dating results that date 20% of the admixture, post-arrival in Europe (post Avar period: 567 CE). Also, this*

percentage matches quite well with the proportion of qpAdm models for which we can reject a non-European Pontic-Steppe and Sarmatian-related mixture while we cannot reject the 6th c CE local European admixture.

C: - “The majority of these are older men (35-59)”, clarify “majority”.

R: *We clarified what we mean with majority (i.e. 85%)*

C: - “Pedigree building strategy”: additional support (e.g., simulations) for the approach used to resolve inconsistencies in the case of levirate unions would be good.

R: *We thank the reviewer for this comment. It is a very good idea. We have actually plans to develop the methodology further in future studies but it would require extensive testing with simulations. Especially since we are not sure that a case of levirate could be distinguished based only on genetic relatedness pattern given the biological variability of genetic inheritance. Furthermore, we identified all but one of our levirate cases directly by the relationships built around them in the pedigree (as described in the text) so for the only case we are not sure if it is a levirate directly from the pedigree structure (the case at the KUP site Extended Data Fig. 2), we would prefer to leave it as a possible scenario that indirectly fits best with the other second degree relations to other individuals (as described in the Supplementary Information).*

C: - Supp Fig 16 and 17. It is hard to tell circles and squares apart.

R: *We have increased the visibility of the symbols of now Supplementary Fig. 18 and 19.*

C: - L144-145 should this be fig 2b? Also, it took me a few readings to understand what highest/lowest median refers to.

R: *Yes the reviewer is correct should have been Figure 2b, now moved to Extended Data Fig. 8c. We corrected the reference to the figure accordingly and we rephrased the sentence to make more explicit what we mean.*

C: - I found it hard to locate supplementary tables as tables in the excel files do not always have a caption.

R: *We have included captions for the Excel Supplementary Tables.*

C: - EDF7 caption “Aver” → “Avar”?

R: *We corrected this typo.*

C: - “Population genetic theory says“ → “suggests”?

R: *We changed as the reviewer suggests.*

C: - L192: The numbers in parentheses are not defined in the caption. Is it the Explained Variance Ratio?

R: *We explained the percentages in the caption. Yes, it is the explained variance ratio, expected to be very low on PCA calculated on genome-wide data (~600,000 markers).*

C: - L768 : (typo) if only the adults (are?) selected

R: *We corrected this typo.*

Reviewer Reports on the First Revision:

Referees' comments:

Referee #1:

Remarks to the Author:

I find it challenging to add significantly to my previous exceptionally positive comments on this article. The revisions made to the manuscript enhance its clarity, particularly for readers from diverse backgrounds, given the expected multidisciplinary audience.

A. This contribution exemplifies innovative research, presenting critical findings poised to leave a substantial impact on future research endeavors. The main results have been masterfully summarized in a concise and comprehensible manner.

B. The research presented here is undeniably groundbreaking and novel. Its potential impact extends across various disciplines, including history and archaeology. The longstanding tradition of Avar research stands to benefit significantly from this novel analytical approach, which not only substantiates existing assumptions but also paves the way for entirely new perspectives.

C. From a humanities standpoint, I may not offer an extensive evaluation of the approach's validity and data quality. Nevertheless, I must commend the exceptionally high quality of the presentation. It remains accessible and intelligible even to readers from different disciplines. The work is presented in a manner that enables anyone to effortlessly navigate each step.

D. The deployment of statistics in this work is both lucid and persuasive. It admirably addresses uncertainties and makes them readily discernible.

E. The conclusions have been cautiously formulated, staying within the confines of the specific research scope. Hypotheses are clearly delineated as such. The study conveys an impression of substantial reliability, engendering trust in the validity of the results obtained.

F. I find no suggestions for improvement to offer; the manuscript is remarkably well-crafted.

G. While it is always conceivable to extend the references, a judicious line must be drawn to determine what to include. Given the extensive and multilingual body of literature on the Avars, this study does not delve deeply into it, a decision that appears reasonable for the present purpose.

H. The abstract, introduction, and conclusion are all composed in a clear and comprehensible manner, in line with the rest of the article.

Referee #2:

Remarks to the Author:

I thank the authors for addressing my comments and improving the structure of the paper. I think it reads more clearly now and is more appropriate for a general readership. I'm also happy the authors have addressed my concerns regarding results interpretation and removed less novel results to the supplemental.

However, I'm concerned that the main text is still a little unwieldy in parts, with key findings getting lost in overly detailed descriptions of results (particularly the pedigree section). For a journal of this calibre, a bit more work might be needed to tighten things up prior to final publication. I provide some suggestions below. Other than that, all I can do is congratulate the authors on an excellent and highly impactful body of research.

Suggestions:

1. Perhaps break the pedigree section into "patriline" and "levirate" sections, as it is more than twice the length of other sections

2. Where possible, state the headline finding at the start of each section and then provide more detailed descriptions of the supporting pieces of evidence in turn.

For example, in "Pedigrees: strict patriline and levirate unions with sites", despite the section title, it takes almost a page before patriline or levirate unions are mentioned. Neither phenomenon is mentioned in the previous introductory section either. Because of this, the significance of the detailed breakdowns of male and female individuals into different categories is lost on the non-expert reader. It would be much easier for the general reader if the first line of the section borrowed from lines 150-151 and read something like:

"We discovered a total of 373 pairs of first degree and >500 second degree relatives using KIN (Methods; Supplementary Table 4). Such a large number of close relatives allowed us to reconstruct a total of 31 pedigrees of varying sizes, ranging from 2 to 146 individuals (Fig. 2a; Extended 118 Data Fig. 1-3; Supplementary Fig. 6, 10 and 14). We find these genealogies provided compelling evidence for a patrilineal descent system, patrilocality and female exogamy.

First, we found a striking difference in Y chromosome and mtDNA diversity at all cemetery sites...

Second, no cases of multiple female-female transmissions across generations. In contrast, all fathers descend through male-line from founder males ...

Third, sex-bias among adults in pedigree (fewer females and adult females have no parents) ...

Fourth, sex-bias among unrelated individuals (more adult females) ... "

3. As I did in my prior review, I would recommend giving information on uniparental markers first (currently it is lines 144-151) as it is needed to understand concepts such as "lineage male" (line

140). Without information on there being two dominant Y chromosome lineages at RK, it is also unclear why the emphasis is placed on “founding males” rather than their female spouses (line 121).

4. Remove extraneous detail from results where possible.

For example, a lot of unnecessary numbers are given. This can work to distract rather than aid the reader. For example (lines 119), in RK there are 202 individuals with close relatives and 64 with no relatives. Of the related individuals, 146 form a “macro” pedigree and “the remaining 22” form 9 smaller ones. But $146+22=168$ not 202. I’m sure there is a reason these numbers don’t add up, but it is unnecessary detail and confusing for the reader as the exact numbers are not that relevant. Perhaps a table could be used to summarise the numbers of males and females in different demographic categories if needed.

5. Provide some introductory context to patriliney and female exogamy at the beginning, as these are two main findings of the research. In the introduction, the authors mention the Avar’s maintained “Central Asian character of the political structure”, but it is unclear what this structure is. Does this refer to patrilineal descent groups? Much later, in lines 199-207, the authors note that patrilineal social organisation is the norm for Eurasian pastoralist steppe people. This information might be more useful earlier on.

6. Strengthen the opening to the conclusion (Line 336). “Although we sought to distinguish between biological and social relatedness” is an odd start as this was not the main aim of the study and was not an aim stated previously. I would rephrase. If I am wrong and this was a main aim of the study, why did the authors expect there to be a distinction between biological and social relatedness in Avar communities? Is this expectation based on data from Eurasian steppe populations, as my impression is that the best comparative communities placed strong emphasis on biological descent lines to male founders, which formed the basis of social hierarchies (lines 200-207). Perhaps an example of a comparative steppe community distinguishing between biological and social relatedness would be useful (e.g. in the intro).

Small suggested edits:

1. I don’t think the term “population replacement” is intuitive when it comes to describing the political realignment at RK, as it suggests a broader population genetic phenomenon. As you say, the shift is only detectable with respect to recent biological relatedness. I think “community shift” or “community replacement” would work better, as is used multiple times in the main text.

2. Lines 301-302: a process of continuous admixture BETWEEN WESTERN AND EASTERN EURASIAN SOURCES.

3. Lines 331: “Despite some admixture” with who? Local Carpathian populations?

4. Up to the editor’s discretion, but I would replace “~” with “approximately”

5. Line 80 -> 298 closely related PAIRS of individuals?

6. Line 119 -> Had at least one close relative at THE site
7. Line 159-161 -> This is very hard to follow. Split into two sentences.
8. Line 317 -> "Thanks to our dense sampling" is informal language -> "The density of sampling allowed us to uncover"
9. Line 330: "genetically and culturally different" -> "genetically and culturally distinct"
10. Line 47 (remove repetition) "The absence of consanguinity indicates this society maintained a detailed memory of ancestry over generations."
11. Capitalise Bayesian

Referee #3:

Remarks to the Author:

Gnecchi-Ruscone, Rácz et al. present results of an analysis involving the reconstruction of multigenerational pedigrees using genomes from Avar-period cemeteries. I like the new structure of the revised text and find it better underlies the main results of the work. I appreciate the efforts that the authors put into looking into the different IBD analyses, comparing the output of different methods, and studying the impact of sex-specific maps on different results. Most of my concerns were addressed.

I still have the following minor concern on the classification of some 2nd degree relationships. (As the authors suggest in their answer, this part of the analysis only concerns a few pairs and do not affect the overall pedigree structure.)

Supplementary Figures 17 and 18 compare the distribution of simulated and real data to classify some of the 2nd degree relationships. The authors now use more realistic simulations (sex-specific recombination and more accurate genome lengths) which improve the analysis. However, the new supplementary figures raise more concerns about this procedure:

- There seems to be a 10% difference in the number of detected IBD segments in the simulated (Supp. Figure 19) and real analyses (Supp. Figure 20). I therefore wonder if such a difference could also be present in Supp. Figure 18, and I am not sure if classifying individuals based on their location on these plots is robust. For example, is there correlation between the values on both axis with the coverage of the individuals?

- Supp. Figure 22 shows how different the distributions can be for 2nd degree related individuals. Wouldn't having an excessive number of males in the pedigree significantly shift the expected distribution of the simulated data towards smaller values on the y-axis?

Based on these two points, I wonder whether the following statement should be trusted: "The distribution of the IBD values of the pairs involving these individuals and RKF080 confirm the directionality as the pair RKF076_RKF080 falls within the distribution of grandparent-grandchild and the other relation falling to the extreme of the avuncular relation distribution thus excluding the grandparent-grandchild possibility".

Other comments

- 125 "via more than 12cM long shared identity-by-descent segments (IBDs) along the genome (Fig. 3"

If I understand correctly, Figure 3 is obtained by using anclBD, whereas this section starts by mentioning KIN, which seems to also output information about IBD segments. I was confused about

whether the length referred to in this sentence is obtained from KIN or anclBD.

- 790 "significantly different between males and females (Figure 4c, Supplementary Fig. 47 and 48).
In the"

Figure 3c?

- 131: "Only one adult daughter (RKC024) has offspring buried in the cemetery, and the missing male partner is closely related to other members of the pedigree and therefore distantly (6th degree) to her."

- 196: "distant biological consanguinity. It is intriguing that the only case we detected of reproductive partners being related was to the 6th degree (which would still be consistent with such rules) and involves the only non-exogamous female. This further suggests the uniqueness of this single case."

I could not find how the degree of the missing male partner was determined. If I missed it in the Supplementary Note, maybe adding the term RKC024 in the paragraph would help finding the information?

Referee #4:

Remarks to the Author:

Gnecchi-Ruscione, Rácz et al. present results of an analysis involving the reconstruction of multigenerational pedigrees using genomes from Avar-period cemeteries. I like the new structure of the revised text and find it better underlies the main results of the work. I appreciate the efforts that the authors put into looking into the different IBD analyses, comparing the output of different methods, and studying the impact of sex-specific maps on different results. Most of my concerns were addressed.

I still have the following minor concern on the classification of some 2nd degree relationships. (As the authors suggest in their answer, this part of the analysis only concerns a few pairs and does not affect the overall pedigree structure.)

Supplementary Figures 17 and 18 compare the distribution of simulated and real data to classify some of the 2nd degree relationships. The authors now use more realistic simulations (sex-specific recombination and more accurate genome lengths) which improve the analysis. However, the new supplementary figures raise more concerns about this procedure:

- There seems to be a 10% difference in the number of detected IBD segments in the simulated (Supp. Figure 19) and real analyses (Supp. Figure 20). I therefore wonder if such a difference could also be present in Supp. Figure 18, and I am not sure if classifying individuals based on their location on these plots is robust. For example, is there a correlation between the values on both axes and the coverage of the individuals?

- Supp. Figure 22 shows how different the distributions can be for 2nd-degree related individuals. Wouldn't having an excessive number of males in the pedigree significantly shift the expected distribution of the simulated data towards smaller values on the y-axis?

Based on these two points, I wonder whether the following statement should be trusted: "The distribution of the IBD values of the pairs involving these individuals and RKF080 confirm the directionality as the pair RKF076_RKF080 falls within the distribution of grandparent-grandchild and the other relation falling to the extreme of the avuncular relation distribution thus excluding the grandparent-grandchild possibility".

Other comments

- 125 "via more than 12cM long shared identity-by-descent segments (IBDs) along the genome (Fig. 3"

If I understand correctly, Figure 3 is obtained using anclBD, whereas this section starts by mentioning KIN, which seems to also output information about IBD segments. I was confused about

whether the length referred to in this sentence is obtained from KIN or anclBD.

- 790 "significantly different between males and females (Figure 4c, Supplementary Fig. 47 and 48).
In the"

Figure 3c?

- 131: "Only one adult daughter (RKC024) has offspring buried in the cemetery, and the missing male partner is closely related to other members of the pedigree and therefore distantly (6th degree) to her."

- 196: "distant biological consanguinity. It is intriguing that the only case we detected of reproductive partners being related was to the 6th degree (which would still be consistent with such rules) and involves the only non-exogamous female. This further suggests the uniqueness of this single case."

I could not find how the degree of the missing male partner was determined. If I missed it in the Supplementary Note, maybe adding the term RKC024 in the paragraph would help find the information?

Author Rebuttals to First Revision:

We thank the reviewers for the constructive feedback. We have addressed all the reviewers' comments, a detailed point-by-point response to these comments can be found below (marked "R"). All the changes, new additions and re-formatting adjustments are marked in red in the manuscript and supplementary information for the convenience of the peer review process.

Referees' comments:

Referee #1 (Remarks to the Author):

I find it challenging to add significantly to my previous exceptionally positive comments on this article. The revisions made to the manuscript enhance its clarity, particularly for readers from diverse backgrounds, given the expected multidisciplinary audience.

A. This contribution exemplifies innovative research, presenting critical findings poised to leave a substantial impact on future research endeavors. The main results have been masterfully summarized in a concise and comprehensible manner.

B. The research presented here is undeniably groundbreaking and novel. Its potential impact extends across various disciplines, including history and archaeology. The longstanding tradition of Avar research stands to benefit significantly from this novel analytical approach, which not only substantiates existing assumptions but also paves the way for entirely new perspectives.

C. From a humanities standpoint, I may not offer an extensive evaluation of the approach's validity and data quality. Nevertheless, I must commend the exceptionally high quality of the presentation. It remains accessible and intelligible even to readers from different disciplines. The work is presented in a manner that enables anyone to effortlessly navigate each step.

D. The deployment of statistics in this work is both lucid and persuasive. It admirably addresses uncertainties and makes them readily discernible.

E. The conclusions have been cautiously formulated, staying within the confines of the specific research scope. Hypotheses are clearly delineated as such. The study conveys an impression of substantial reliability, engendering trust in the validity of the results obtained.

F. I find no suggestions for improvement to offer; the manuscript is remarkably well-crafted.

G. While it is always conceivable to extend the references, a judicious line must be drawn to determine what to include. Given the extensive and multilingual body of literature on the Avars, this study does not delve deeply into it, a decision that appears reasonable for the present purpose.

H. The abstract, introduction, and conclusion are all composed in a clear and comprehensible manner, in line with the rest of the article.

R: We greatly appreciate the evaluation, and we are especially pleased that the reviewer believes the changes to the document have made it clearer for readers from diverse backgrounds.

Referee #2 (Remarks to the Author):

I thank the authors for addressing my comments and improving the structure of the paper. I think it reads more clearly now and is more appropriate for a general readership. I'm also happy the authors have addressed my concerns regarding results interpretation and removed less novel results to the supplemental.

However, I'm concerned that the main text is still a little unwieldy in parts, with key findings getting lost in overly detailed descriptions of results (particularly the pedigree section). For a journal of this calibre, a bit more work might be needed to tighten things up prior to final publication. I provide some suggestions below. Other than that, all I can do is congratulate the authors on an excellent and highly impactful body of research.

Suggestions:

1. Perhaps break the pedigree section into “patriliney” and “levirate” sections, as it is more than twice the length of other sections

R: We thank the reviewer for this feedback, it is indeed quite a long section and we splitted it in two separate sections as the reviewer suggested (line 175).

2. Where possible, state the headline finding at the start of each section and then provide more detailed descriptions of the supporting pieces of evidence in turn.

R: We thank the reviewer for this editing advice. We checked all the sections and when not already present, we added a summary sentence describing the main finding of each section. When possible/it made sense as the very first sentence, otherwise shortly after.

For example, in “Pedigrees: strict patriliney and levirate unions with sites”, despite the section title, it takes almost a page before patriliney or levirate unions are mentioned. Neither phenomenon is mentioned in the previous introductory section either. Because of this, the significance of the detailed breakdowns of male and female individuals into different categories is lost on the non-expert reader. It would be much easier for the general reader if the first line of the section borrowed from lines 150-151 and read something like:

“We discovered a total of 373 pairs of first degree and >500 second degree relatives using KIN (Methods; Supplementary Table 4). Such a large number of close relatives allowed us to reconstruct a total of 31 pedigrees of varying sizes, ranging from 2 to 146 individuals (Fig. 2a; Extended 118 Data Fig. 1-3; Supplementary Fig. 6, 10 and 14). We find these genealogies provided compelling evidence for a patrilineal descent system, patrilocality and female exogamy.

R: We thank the reviewer for this suggestion which we found very good and implemented it in the new version of the text (lines 122-125)

First, we found a striking difference in Y chromosome and mtDNA diversity at all cemetery sites...

Second, no cases of multiple female-female transmissions across generations. In contrast, all fathers descend through male-line from founder males ...

Third, sex-bias among adults in pedigree (fewer females and adult females have no parents) ...

Fourth, sex-bias among unrelated individuals (more adult females) ... ”

3. As I did in my prior review, I would recommend giving information on uniparental markers first (currently it is lines 144-151) as it is needed to understand concepts such as “lineage male” (line 140). Without information on there being two dominant Y chromosome lineages at RK, it is also unclear why the emphasis is placed on “founding males” rather than their female spouses (line 121).

R: We thank the reviewer for this recommendation. We now introduce the uniparental haplogroups in the summary sentence (lines 124-125). The emphasis is placed on the “founding males” and “lineage males” rather than the female “spouses” because of the strict patrilineal pedigree structure: all the males are descendants of another male in the pedigree except for the “founding males” because the pedigrees have to “start” somewhere so we don’t have the previous generation for these males. The female spouses are instead all (almost) “external” regardless if they are at the top or at any other generation of the genealogies.

4. Remove extraneous detail from results where possible.

For example, a lot of unnecessary numbers are given. This can work to distract rather than aid the reader. For example (lines 119), in RK there are 202 individuals with close relatives and 64 with no relatives. Of the related individuals, 146 form a “macro” pedigree and “the remaining 22” form 9 smaller ones. But $146+22=168$ not 202. I’m sure there is a reason these numbers don’t add up, but it is unnecessary detail and confusing for the reader as the exact numbers are not that relevant. Perhaps a table could be used to summarise the numbers of males and females in different demographic categories if needed.

R: We thank the reviewer for this feedback. The numbers don’t add up because it is missing 34 individuals mentioned now at line 129) that make the sum add up to 202. We now simplified this sentence as the “remaining 22 individuals forming 9 pedigrees” are not mentioned in the main text and figures we have now removed the detail of these numbers from the main text. We

have also included a table (Table 1) as the reviewer suggested with the numbers of individuals according to sex and age at death (adult vs subadults)

5. Provide some introductory context to patriliney and female exogamy at the beginning, as these are two main findings of the research. In the introduction, the authors mention the Avar's maintained "Central Asian character of the political structure", but it is unclear what this structure is. Does this refer to patrilineal descent groups? Much later, in lines 199-207, the authors note that patrilineal social organisation is the norm for Eurasian pastoralist steppe people. This information might be more useful earlier on.

***R:** We thank the reviewer for this feedback, we have now added more introductory context on the topic in the introduction as the reviewer suggested (lines 76-80). For the "Central Asian character of the political structure" we can conclude from written sources on the Avars that document titles of rank also used in the Turkic Khaganate (Pohl 2018, 352-369). To be more precise, we added the names of some of these titles to the text. In terms of the broader social organization, the Eurasian steppe character refers to an interplay of various elements for which we have no space to go into detail, but we refer to the literature and the supplementary information for a more exhaustive explanation. But following the reviewer's suggestion we now clarify the main aspect that concerns the topic of our study (i.e. patrilineality; lines 78-80).*

6. Strengthen the opening to the conclusion (Line 336). "Although we sought to distinguish between biological and social relatedness" is an odd start as this was not the main aim of the study and was not an aim stated previously. I would rephrase. If I am wrong and this was a main aim of the study, why did the authors expect there to be a distinction between biological and social relatedness in Avar communities? Is this expectation based on data from Eurasian steppe populations, as my impression is that the best comparative communities placed strong emphasis on biological descent lines to male founders, which formed the basis of social hierarchies (lines 200-207). Perhaps an example of a comparative steppe community distinguishing between biological and social relatedness would be useful (e.g. in the intro).

***R:** We thank the reviewer for this feedback, we have rephrased this sentence and also moved it a few lines down so it is not the opening anymore (lines 346-348).*

Small suggested edits:

1. I don't think the term "population replacement" is intuitive when it comes to describing the political realignment at RK, as it suggests a broader population genetic phenomenon. As you say, the shift is only detectable with respect to recent biological relatedness. I think "community shift" or "community replacement" would work better, as is used multiple times in the main text.

R: We changed as the reviewer suggested

2. Lines 301-302: a process of continuous admixture BETWEEN WESTERN AND EASTERN EURASIAN SOURCES.

R: We changed as the reviewer suggested

3. Lines 331: "Despite some admixture" with who? Local Carpathian populations?

R: We changed as the reviewer suggested

4. Up to the editor's discretion, but I would replace "~" with "approximately"

R: We changed as the reviewer suggested

5. Line 80 -> 298 closely related PAIRS of individuals?

R: No, the number of pairs is much more since the same individuals are in multiple relationships with one another (this can be counted in Supplementary Table 4, no need to add extra numbers). 298 is the total number of related individuals which is more comparable with the total number of individuals analyzed.

6. Line 119 -> Had at least one close relative at THE site

R: We changed as the reviewer suggested

7. Line 159-161 -> This is very hard to follow. Split into two sentences.

R: We changed as the reviewer suggested

8. Line 317 -> “Thanks to our dense sampling” is informal language -> “The density of sampling allowed us to uncover”

R: We changed as the reviewer suggested

9. Line 330: “genetically and culturally different” -> “genetically and culturally distinct”

R: We changed as the reviewer suggested

10. Line 47 (remove repetition) “The absence of consanguinity indicates this society maintained a detailed memory of ancestry over generations.”

R: We removed repetition as the reviewer suggested

11. Capitalise Bayesian

R: We changed as the reviewer suggested

Referee #3 (Remarks to the Author) & Referee #4 (Remarks to the Author):

Gnecchi-Ruscione, Rácz et al. present results of an analysis involving the reconstruction of multigenerational pedigrees using genomes from Avar-period cemeteries. I like the new structure of the revised text and find it better underlies the main results of the work. I appreciate the efforts that the authors put into looking into the different IBD analyses, comparing the output of different methods, and studying the impact of sex-specific maps on different results. Most of my concerns were addressed.

I still have the following minor concern on the classification of some 2nd degree relationships. (As the authors suggest in their answer, this part of the analysis only concerns a few pairs and do not affect the overall pedigree structure.)

Supplementary Figures 17 and 18 compare the distribution of simulated and real data to classify some of the 2nd degree relationships. The authors now use more realistic simulations (sex-specific recombination and more accurate genome lengths) which improve the analysis. However, the new supplementary figures raise more concerns about this procedure:

- There seems to be a 10% difference in the number of detected IBD segments in the simulated (Supp. Figure 19) and real analyses (Supp. Figure 20). I therefore wonder if such a difference could also be present in Supp. Figure 18, and I am not sure if classifying individuals based on their location on these plots is robust. For example, is there correlation between the values on both axis with the coverage of the individuals?

***R:** We thank the reviewer for noticing this discrepancy between simulated and real data. This is due to the fact that in the previous plots of the simulated data, ancIBD was run without filtering out genomic blocks with less than 220 SNPs per centimorgan as it is done for the real data (line 782-784 in Methods) which is what the developers of ancIBD suggest for 1240K data. In the new plots this filtering was implemented also on the simulated data and the 10% difference noticed by the reviewer from the previous plot is now lost (Supplementary Figure. 19 and below). We now report the results from this new run that also corresponds to the one in the revised and accepted version of Ringbauer et al. article that is scheduled for publication in the upcoming weeks.*

ancIBD on PEDSIM simulations as in Ringbauer et al. 2023

- Supp. Figure 22 shows how different the distributions can be for 2nd degree related individuals. Wouldn't having an excessive number of males in the pedigree significantly shift the expected distribution of the simulated data towards smaller values on the y-axis? Based on these two points, I wonder whether the following statement should be trusted: "The distribution of the IBD values of the pairs involving these individuals and RKF080 confirm the directionality as the pair RKF076_RKF080 falls within the distribution of grandparent-grandchild and the other relation falling to the extreme of the avuncular relation distribution thus excluding the grandparent-grandchild possibility".

R: *We thank the reviewer for this careful assessment. The reviewer is correct that we cannot completely exclude a grandparent-grandchild relation for the other pairs mentioned. So we cannot rule out the alternative pedigree configuration for this specific case that involves a section of the small pedigree 7. As the reviewer also notes above this change would not affect the overall pedigree structure and thus the findings derived from it remain the same. To reflect the concerns raised, we rephrase this sentence of Supplementary Information acknowledging this by simply observing that the very low position of the pair RKF076_RKF080, in the Supplementary Fig. 18, makes the alternative scenario less plausible. The new sentence in the Supplementary Information (page 42) now reads “The distribution of the IBD values of the pairs involving these individuals and RKF080 confirm the directionality as the pair RKF076_RKF080 falls within the distribution of grandparent-grandchild making the alternative scenario less plausible (Supplementary Fig. 18).”*

Other comments

- 125 “via more than 12cM long shared identity-by-descent segments (IBDs) along the genome (Fig. 3”

If I understand correctly, Figure 3 is obtained by using ancIBD, whereas this section starts by mentioning KIN, which seems to also output information about IBD segments. I was confused about whether the length referred to in this sentence is obtained from KIN or ancIBD.

R: *We thank the reviewer for noticing this. Indeed, IBD networks were done with ancIBD but KIN also produces IBD segments, so for avoiding confusion we specified in the text (line 223) and in the figure legend (line 469) that the networks are based on ancIBD results.*

- 790 "significantly different between males and females (Figure 4c, Supplementary Fig. 47 and 48). In the"

Figure 3c?

R: *Yes indeed we mean Figure 3c, that was a typo, we corrected it.*

- 131: "Only one adult daughter (RKC024) has offspring buried in the cemetery, and the missing male partner is closely related to other members of the pedigree and therefore distantly (6th degree) to her."

- 196: "distant biological consanguinity. It is intriguing that the only case we detected of reproductive partners being related was to the 6th degree (which would still be consistent with such rules) and involves the only non-exogamous female. This further suggests the uniqueness of this single case."

I could not find how the degree of the missing male partner was determined. If I missed it in the Supplementary Note, maybe adding the term RKC024 in the paragraph would help finding the information?

R: We thank the reviewer for this comment. We in fact skipped one logical step in the explanation and we understand that this creates confusion. We have of course no information on the missing male partner of RKC024 but on their son (which is RKC012). RKC012 is the one to be 2nd degree related to other members of the pedigree via the paternal line. Therefore the male partner of RKC024 (father of RKC012) is distantly related to RKC024, 6th degree counting the steps of the pedigree that separate RKC024 to the related individual of RKC012. We now rephrased the first sentence to clarify this (lines 136-139).

Reviewer Reports on the Second Revision:

Referees' comments:

Referee #3 (Remarks to the Author):

n/a

Referee #4 (Remarks to the Author):

The revisions to the manuscript further enhance clarity and highlight the novelty of the findings, facilitating comprehension for readers from diverse backgrounds.

The issue regarding the 6th-degree relative has been addressed.

The revised analysis of IBD in simulated data addresses the discrepancy that was present in the previous version.

Minor comment

[Typo] 782: We used the HapBLOCK functions  function